# Chromatin compartmentalization regulates the response to DNA damage

Coline Arnould[1,2,3,8], Vincent Rocher[1,8], Florian Saur[1], Aldo S. Bader[4], Fernando Muzzopappa[1], Sarah Collins[1], Emma Lesage[1], Benjamin Le Bozec[1], Nadine Puget[1], Thomas Clouaire[1], Thomas Mangeat[5], Raphael Mourad[1], Nadav Ahituv[2,3], Daan Noordermeer[6], Fabian Erdel[1], Martin Bushell[2,7], Aline Marnef[1] & Gaëlle Legube[1✉]

The DNA damage response is essential to safeguard genome integrity. Although the contribution of chromatin in DNA repair has been investigated[1,2], the contribution of chromosome folding to these processes remains unclear[3]. Here we report that, after the production of double-stranded breaks (DSBs) in mammalian cells, ATM drives the formation of a new chromatin compartment (D compartment) through the clustering of damaged topologically associating domains, decorated with γH2AX and 53BP1. This compartment forms by a mechanism that is consistent with polymer–polymer phase separation rather than liquid–liquid phase separation. The D compartment arises mostly in G1 phase, is independent of cohesin and is enhanced after pharmacological inhibition of DNA-dependent protein kinase (DNA-PK) or R-loop accumulation. Importantly, R-loop-enriched DNA-damage-responsive genes physically localize to the D compartment, and this contributes to their optimal activation, providing a function for DSB clustering in the DNA damage response. However, DSB-induced chromosome reorganization comes at the expense of an increased rate of translocations, also observed in cancer genomes. Overall, we characterize how DSB-induced compartmentalization orchestrates the DNA damage response and highlight the critical impact of chromosome architecture in genomic instability.

DNA DSBs are highly toxic lesions that can trigger translocations or gross chromosomal rearrangements, thereby severely challenging genome integrity and cell homeostasis. Chromatin has a pivotal function during DNA repair, which is achieved by either non-homologous end joining or homologous recombination pathways[1]. Yet, little is known about how chromosome architecture contributes to these processes. DSBs activate the DNA damage response (DDR) that largely relies on phosphatidylinositol 3-kinase (PI3K)-like protein kinases, including ATM and DNA-PK, and on the establishment of megabase-sized, γH2AX-decorated chromatin domains that act as seeds for subsequent signalling events, such as 53BP1 recruitment and DDR focus formation[4,5]. Importantly, γH2AX spreading is influenced by the pre-existing chromosome conformation[6–8] and loop extrusion, which leads to the formation of topologically associating domains (TADs), is instrumental for γH2AX establishment and DDR focus assembly[7]. Moreover, irradiation reinforces TADs genome wide[9]. At a larger scale, DSBs display the ability to cluster within the nuclear space (that is, fuse) forming large microscopically visible structures that are composed of several individual repair foci[10–12]. DSB clustering depends on the actin network, the LINC (a nuclear envelope embedded complex)[11,13–15], as well as on the phase-separation properties of 53BP1[16–18]. The function of DSB

clustering has remained enigmatic given that juxtaposition of several DSBs can elicit translocation (that is, illegitimate rejoining of two DNA ends)[12], questioning the selective advantage of DSB clustering/repair focus fusion.

## ATM drives an acute reinforcement of damaged TADs

To systematically characterize chromosome behaviour after DSBs, we analysed 3D genome organization using Hi-C data generated in the human DIvA cell line, in which multiple DSBs are induced at annotated positions after addition of hydroxytamoxifen (OHT)[19]. Our previous analyses using γH2AX chromatin immunoprecipitation followed by sequencing (ChIP–seq) and direct DSB mapping by BLESS identified 80 robustly induced DSBs in this cell line[5]. Using differential Hi-C maps, we found that intra-TAD contact frequencies were strongly increased within TADs that experience a DSB (that is, damaged TADs; Fig. 1a (right, red square)) compared to undamaged TADs, while contacts with neighbouring adjacent domains were significantly decreased (Fig. 1a (right blue square) and 1b). Notably, in some instances, the DSB itself displayed a particularly strong depletion of contact frequency with adjacent chromatin (Fig. 1c (black arrow)), indicating that the

[1]MCD, Centre de Biologie Intégrative (CBI), CNRS, Université de Toulouse, UT3, Toulouse, France. [2]Department of Bioengineering and Therapeutic Sciences, University of California, San Francisco, San Francisco, CA, USA. [3]Institute for Human Genetics, University of California, San Francisco, San Francisco, CA, USA. [4]Cancer Research UK Beatson Institute, Glasgow, UK. [5]LITC Core Facility, Centre de Biologie Integrative, Université de Toulouse, CNRS, UPS, Toulouse, France. [6]Université Paris-Saclay, CEA, CNRS, Institute for Integrative Biology of the Cell (I2BC), Gif-sur-Yvette, France. [7]Institute of Cancer Sciences, University of Glasgow, Glasgow, UK. [8]These authors contributed equally: Coline Arnould, Vincent Rocher. ✉e-mail: gaelle.legube@univ-tlse3.fr

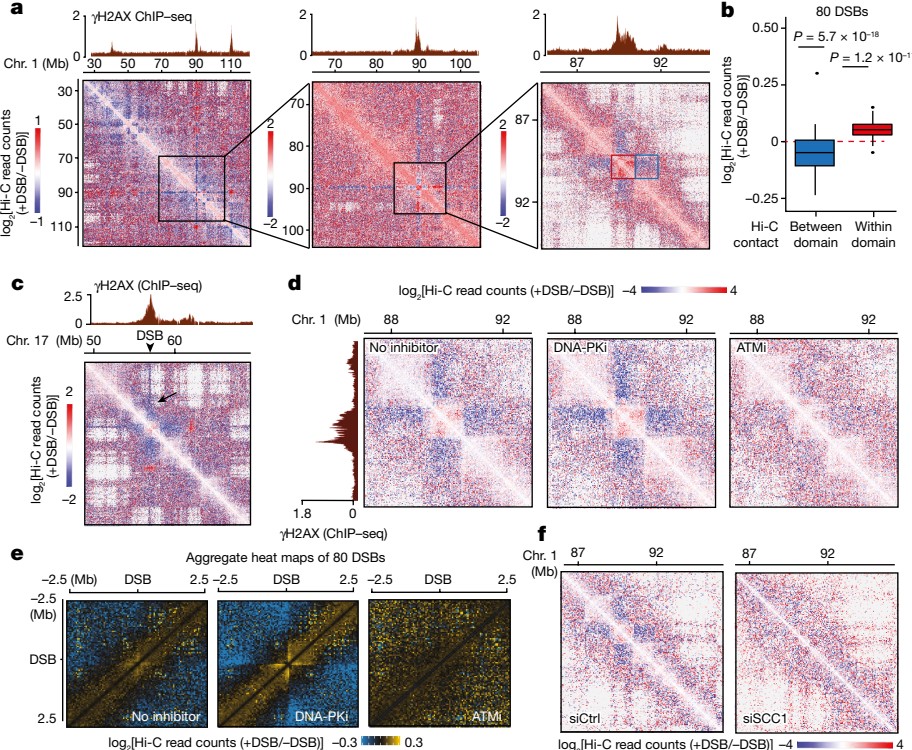

**Fig. 1 | Cohesin and ATM-dependent TAD reinforcement in response to DSBs. a**, Differential Hi-C contact matrix ($\log_2$[+DSB/−DSB]) in DIvA cells on chromosome 1 at 250 kb (left), 100 kb (middle) and 25 kb (right) resolutions. The red square shows a damaged TAD, within which *cis* interactions are enhanced. The blue square indicates a decreased interaction between the damaged TAD and its adjacent TAD. One representative experiment is shown. **b**, Differential Hi-C read counts within γH2AX domains containing the 80 best-induced DSBs (red) or between these 80 damaged domains and their adjacent chromatin domains (blue). *P* values were calculated using two-sided nonparametric Wilcoxon rank-sum tests against $\mu = 0$. **c**, Differential Hi-C contact matrix on chromosome 17. The contacts engaged by the DSB itself are indicated by a black arrow. One representative experiment is shown. **d**, Differential Hi-C contact matrix without inhibitor (left), with DNA-PK inhibitor (DNA-PKi; middle) or with ATM inhibitor (ATMi; right). **e**, Averaged differential Hi-C contact matrix without inhibitor, and after DNA-PK or ATM inhibition as indicated, centred on the 80 best-induced DSBs (50 kb resolution). **f**, Differential Hi-C contact matrix on chromosome 1 in DIvA cells transfected with control (siCtrl) or *SCC1* siRNA.

DSB is kept isolated from the surrounding environment, outside of its own TAD.

We further investigated the contribution of PI3K-like protein kinases involved in DSB response by performing Hi-C analysis in the presence of ATM and DNA-PK inhibitors, which negatively and positively impact γH2AX accumulation at DSBs, respectively (in contrast to ATR inhibition, which does not noticeably alter γH2AX focus formation in DIvA cells)[7,20]. ATM inhibition decreased intra-TAD contacts after DSB induction (Fig. 1d and Extended Data Fig. 1a). TAD structures visualized in Hi-C maps are believed to arise due to cohesin-mediated loop extrusion[21]. Our previous research indicated that a bidirectional, divergent, cohesin-dependent loop-extrusion process takes place at DSBs[7], which gives rise to a 'cross' centred on the DSB on differential Hi-C maps (Fig. 1e). Notably, ATM inhibition impaired loop extrusion, whereas DNA-PK inhibition strongly increased it (Fig. 1e). Moreover, depletion of the cohesin subunit SCC1, which abolishes DSB-induced loop extrusion[7], decreased the reinforcement of intra-TAD contacts in damaged, γH2AX-decorated, chromatin domains (Fig. 1f and Extended Data Fig. 1b).

Together, these data indicate that ATM triggers cohesin-mediated loop extrusion arising from the DSB and the insulation of the damaged TADs from the surrounding chromatin.

## Damaged TADs cluster within the nuclear space

We further analysed Hi-C data[7] with respect to long-range contacts within the nuclear space. Hi-C data revealed that DSBs cluster together (Fig. 2a), as previously observed using capture Hi-C[11] and Hi-C[15]. Notably, DSB clustering takes place between entire γH2AX/53BP1-decorated TADs and can occur between DSBs induced on the same chromosome (Extended Data Fig. 2a) as well as on different chromosomes (Extended Data Fig. 2b). To further identify if DSB clustering holds true outside of the context of artificially induced DSBs, we mapped endogenous DSB hotspots in untreated DIvA cells using ChIP−seq analysis of phosphorylated ATM (pATM)[7] (Extended Data Fig. 2c). Notably, we observed elevated *trans* chromosomal Hi-C contacts between pATM-enriched loci (endogenous DSBs) compared with those at random positions (Fig. 2b), showing that DSB clustering also occurs between physiologically induced DSBs.

Notably, our Hi-C data indicated that some γH2AX domains, assembled after DSB induction in DIvA cells, were able to interact with more than a single other γH2AX domain (Fig. 2c (black arrows)). This was also observed when performing live imaging using random illumination microscopy (RIM) in 53BP1−GFP DIvA cells (Fig. 2d and Supplementary Videos 1−3). Notably, this ability to form clusters of multiple TADs (also known as TAD cliques[22]) after DSB induction correlated with DSB-induced chromatin features that occur at the scale of an entire TAD[5], including γH2AX, 53BP1 and ubiquitin chain levels as well as the depletion of histone H1 around DSBs as detected by ChIP−seq (Fig. 2e). It also correlated with initial RNA polymerase II occupancy before DSB induction, indicating that DSBs prone to cluster and form damaged TAD cliques are those occurring at transcribed loci (Fig. 2e). In agreement, DSBs preferentially channelled to homologous-recombination repair, previously shown to be those induced in active loci[23], also displayed

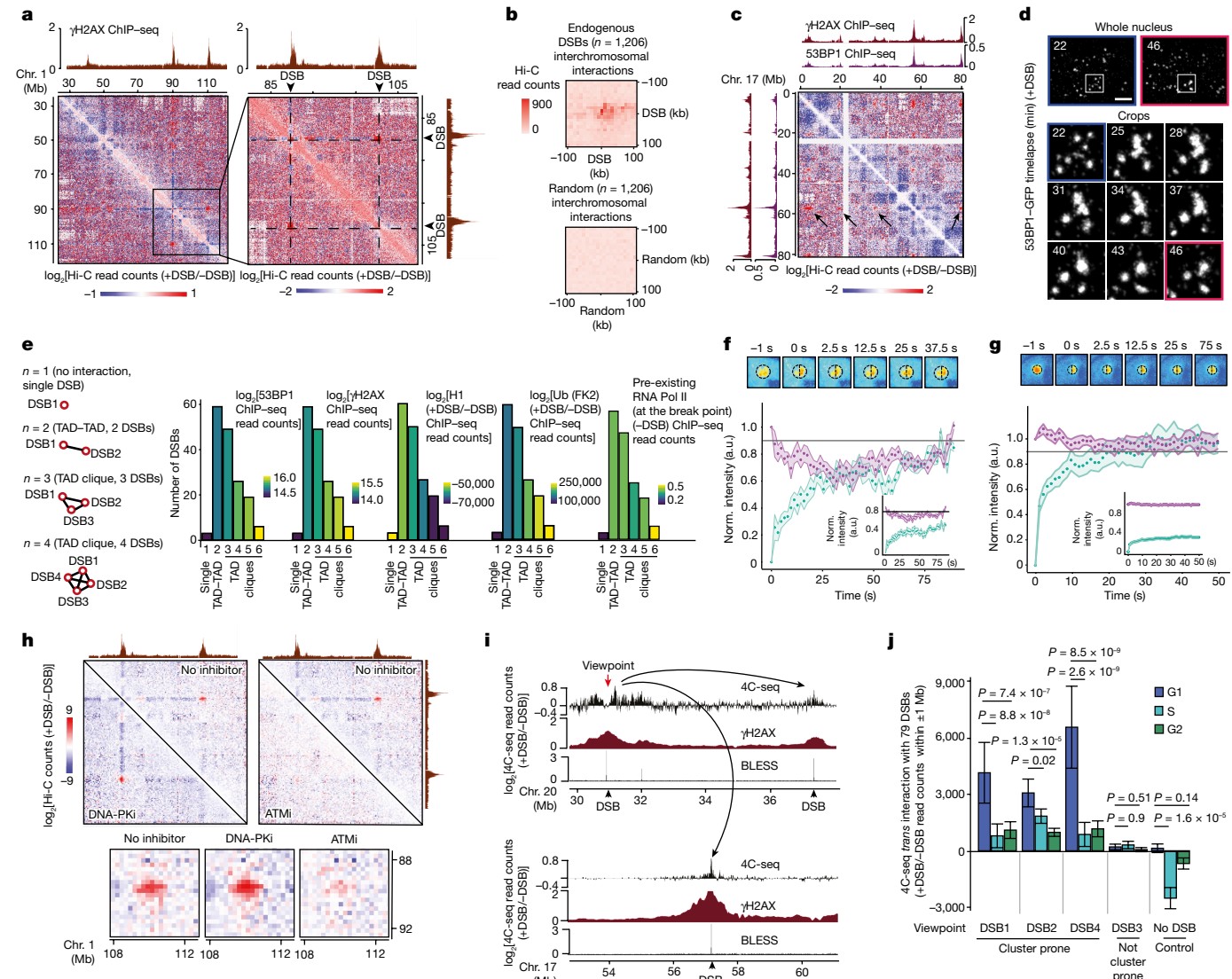

**Fig. 2 | Cell-cycle-regulated ATM-dependent, but DNA-PK-independent, clustering of damaged TADs. a**, Differential Hi-C contact matrix (log₂[+DSB/–DSB]) on chromosome 1. One representative experiment is shown. **b**, Aggregate peak analysis of *trans*-chromosomal contacts between *n* = 1,206 endogenous DSB (top) or *n* = 1,206 random locations (bottom). **c**, Hi-C contact matrix (log₂[+DSB/–DSB]) on chromosome 17. The black arrows indicate clustering of one DSB with other DSBs. One representative experiment is shown. **d**, 53BP1–GFP DIvA cells were filmed 20 min after DSB induction at 3-min intervals for 1 h. Examples of fusions of several 53BP1–GFP foci are shown (time points indicated in minutes). Scale bar, 5 μm. **e**, γH2AX domains were categorized on the basis of their interaction with one or multiple γH2AX domains as indicated. The levels of γH2AX, 53BP1, H1 and ubiquitin chains after DSB on 500 kb or local pre-existing RNA polymerase II (before DSB on 10 kb) were computed across each category. **f**, Recovery curves of the bleached half (green) and non-bleached half (magenta) normalized to the immobile fraction, or not (insets), obtained from half-FRAP analysis of 53BP1–GFP micro-irradiation sites 15 min after laser-induced damage. *n* = 9. Representative snapshots of a 53BP1–GFP foci after bleaching are shown at the top. Data are mean ± s.e.m. **g**, Half-FRAP analysis as in **f** of 53BP1–GFP foci (*n* = 22) at 4 h after DSB induction in DIvA cells. Data are mean ± s.d. **h**, Differential Hi-C contact matrix without inhibitor, with DNA-PK or ATM inhibitors as indicated. Bottom, magnification. **i**, Genomic tracks of 4C-seq (log₂[+DSB/–DSB]) using a DSB on chromosome 20 as a viewpoint (red arrow), γH2AX ChIP–seq and BLESS. The black arrows indicate interactions between the viewpoint and other DSBs. One representative experiment is shown. **j**, *Trans* interactions (log₂[ratio (+DSB/–DSB)]) between the viewpoint and the other DSBs computed from 4C-seq experiments in synchronized cells. Data are mean and s.e.m. of *n* = 79. *P* values were calculated using two-sided nonparametric paired Wilcoxon signed-rank tests. a.u., arbitrary units; norm., normalized.

increased clustering compared with DSBs repaired by non-homologous end-joining, despite equivalent cleavage as determined previously using BLESS[5] (Extended Data Fig. 2d).

To obtain additional insights into the mechanism that triggers DSB clustering, we further investigated the hypothesis that clustering may occur through liquid–liquid phase separation (LLPS) as previously proposed for radiation and etoposide-induced 53BP1 foci[16,17]. One of the characteristics of biomolecular condensates formed by LLPS is the presence of a barrier at the interface, as molecules have to break

interactions with their neighbours when leaving the condensate[24]. We therefore sought to test whether 53BP1 molecules preferentially diffuse within 53BP1 foci, which would signal the existence of such a barrier. For this, we performed half-fluorescence recovery after photobleaching (half-FRAP), which can detect the presence or absence of an interfacial barrier through a large or small intensity decrease in the non-bleached half, respectively[24,25] (Extended Data Fig. 2e). Half-FRAP experiments on early 53BP1 accumulations induced by laser micro-irradiation (15 min) showed that the mobile 53BP1–GFP

pool is constrained by an interfacial barrier, indicative of LLPS and in agreement with previous work[16,17], therefore validating our ability to detect LLPS (Fig. 2f). By contrast, half-FRAP analysis of 53BP1 foci in 53BP1–GFP DIvA cells (4 hours) showed that the decrease in intensity in the non-bleached half equalled that for freely diffusing molecules (Fig. 2g; compare magenta points to the horizontal black line), indicating that there is no significant interfacial barrier for 53BP1 under these conditions. Moreover, quantitative analysis of fusions of 53BP1 foci in DIvA cells showed that such fusions occur quite slowly and, most of the time, do not yield a spherical homogenous end product (Extended Data Fig. 2f and Supplementary Videos 3–6). We determined the capillary velocity, a quantity that characterizes the fusion kinetics[25], to be 0.003 µm s$^{-1}$ (Extended Data Fig. 2g), which is three orders of magnitude smaller than the values typically obtained for biomolecular condensates formed by LLPS[24,26]. Notably, detection of poly(ADP-ribose) (PAR) chains using macro–mKate2, indicated that micro-irradiation sites displayed elevated levels of PAR at early timepoints after DNA damage, whereas late clusters of 53BP1 foci detected in DIvA cells lacked PAR (Extended Data Fig. 2h). Taken together, these results suggest that LLPS contributes to the early accrual of 53BP1 at sites of damage, which coincides with PAR accumulation, in agreement with the previously identified properties of PAR and 53BP1[16,27]. By contrast, the lack of an interfacial barrier at later timepoints when 53BP1 foci have clustered favours a model in which this clustering is driven by self-interactions among chromatin-bound 53BP1 molecules that mediate polymer–polymer phase separation (PPPS), a mechanism that was previously proposed for the assembly of heterochromatin foci[25] and cohesin–DNA clusters[28].

## ATM and cell cycle regulation of DSB clustering

Chromatin compartmentalization was previously found to be independent of cohesin (and rather counteracted by loop extrusion[29]). In agreement with this, inspection of individual DSBs indicated that *SCC1* depletion using short interfering RNA (siRNA) did not alter clustering (Extended Data Fig. 3a). Quantification of *trans* interactions between all DSBs also indicated that *SCC1* depletion did not modify the ability of damaged TADs to physically interact together (Extended Data Fig. 3b).

If DSB clustering is mediated by the self-segregation of γH2AX/53BP1-enriched TADs, we postulated that this process should depend on ATM activity, which is required to establish γH2AX/53BP1 domains[20], but, by contrast, should be increased by impairing DSB repair (as persistent DSBs with sustained γH2AX/53BP1 will have more time to self-segregate). Indeed, we found that inhibition of ATM, which abolishes γH2AX-domain formation[20], compromised DSB clustering, whereas inhibition of DNA-PK activity, which severely impairs rejoining of AsiSI-induced DSBs without impairing γH2AX establishment[20], triggered a substantial increase in DSB clustering (Fig. 2h and Extended Data Fig. 3c). Concomitant inhibition of DNA-PK in cells treated with ATM inhibitor did not restore DSB clustering (Extended Data Fig. 3d,e), further suggesting that ATM activity is required to mediate DSB clustering, even in the presence of persistent DSBs.

We previously reported that DSBs induced at active loci, while being repaired by homologous recombination in G2, display delayed repair during G1[11], prompting us to further examine DSB clustering in synchronized cells. DSB clustering (that is, damaged TAD–TAD interaction) could be readily detected using 4C-seq when using a DSB as a viewpoint, as shown by the increase in 4C-seq signal observed on other DSBs induced on the genome (Fig. 2i). 4C-seq experiments performed before and after DSB induction in synchronized cells indicated that DSB clustering is readily detectable during G1 and is strongly reduced during the other cell cycle phases (Extended Data Fig. 3f). G1-specific DSB clustering was observed only when using 'clustering-prone' DSBs as viewpoints, but not when using an undamaged control locus or a DSB unable to cluster as viewpoints (Fig. 2j).

Taken together, our results indicate that, after DSB formation, TADs that carry DSBs and that accumulate γH2AX, 53BP1 and ubiquitin chains, are able to physically contact each other in the nuclear space (that is, cluster) in an ATM and cell cycle regulated manner. Our data are in agreement with a model whereby clustering occurs through self-segregation of TADs that are enriched in γH2AX/53BP1/ubiquitin, through PPPS. As a consequence, DSB clustering is enhanced when DSBs and, therefore, γH2AX/53BP1 persist, such as during G1 and after DNA-PK inhibition.

## Formation of the D compartment after DSB induction

Previous research identified the existence of two main, spatially distinct, self-segregated, chromatin compartments in mammalian nuclei[30]. These chromatin compartments were determined by principal component analysis (PCA) of Hi-C chromosomal contact maps in which the first principal component enabled the identification of loci that share similar interaction patterns, and that can be visualized linearly using eigenvectors. Further correlations with epigenomic features revealed that these two spatially segregated compartments correspond to active chromatin (the A compartment or euchromatin) and inactive chromatin (the B compartment or heterochromatin)[30]. A/B compartment identification using our Hi-C datasets revealed that DSB induction does not trigger major changes in genome compartmentalization into euchromatin versus heterochromatin (Extended Data Fig. 4a,b). Moreover, neither inhibition of DNA-PK nor ATM significantly modified the ability of the genome to segregate into active A and inactive B compartments (Extended Data Fig. 4b). DSB induction did not generally lead to compartment switching of the underlying chromatin domain, except in very few cases: among the 80 DSBs induced by AsiSI, 58 DSBs localized in the A compartment and all remained in the A compartment after DSB induction (Extended Data Fig. 4c (top)). Conversely, among the 22 DSBs induced in the B compartment, only 4 showed a shift from B to A (Extended Data Fig. 4c (middle and bottom)). Notably, DSB clustering was mostly detected for DSBs residing in the A compartment (Extended Data Fig. 4d).

Beyond the main classification between A/B compartments, subcompartments have since been identified using higher resolution Hi-C maps, which correspond to subsets of heterochromatin (B1–B4) and of active (A1–A2) loci[31]. These subcompartments also correspond to microscopically visible nuclear structures such as nuclear speckles (A1)[32] or polycomb bodies (B1)[31]. Given that previous studies identified large, microscopically detectable γH2AX bodies after DNA damage and that our Hi-C data revealed clustering of damaged TADs, we postulated that DSBs may also induce a subcompartment, in particular within the A compartment (that is, damaged loci in the A compartment that further segregate from the rest of the active compartment). We therefore applied PCA analysis to differential Hi-C maps (that is, contact matrices of with DSB versus without DSB (+DSB/−DSB)) on each individual chromosome. The first chromosomal eigenvector (CEV, PC1) enabled us to identify a DSB-induced chromatin compartment that was readily detectable on chromosomes displaying a large number of DSBs (chromosomes 1, 17 and X) (Fig. 3a and Extended Data Fig. 5a). Notably, a similar analysis on Hi-C maps generated after DNA-PK inhibition, which impairs repair[20] and increases DSB clustering (Fig. 2), enabled the detection of this compartment on additional chromosomes (such as chromosome 6; Extended Data Fig. 5b (bottom track)). This compartment displayed a very strong correlation with γH2AX-decorated chromatin after DSB induction (Fig. 3a and Extended Data Fig. 5a–d) and was named the D compartment, for DSB-induced compartment. Notably, further inspection revealed that the D compartment is not solely generated through the clustering of damaged chromatin. Indeed, we identified chromatin domains lacking DSBs and γH2AX that associate with the D compartment after damage (Fig. 3b (blue rectangle)). After exclusion of γH2AX-covered chromatin domains, correlation

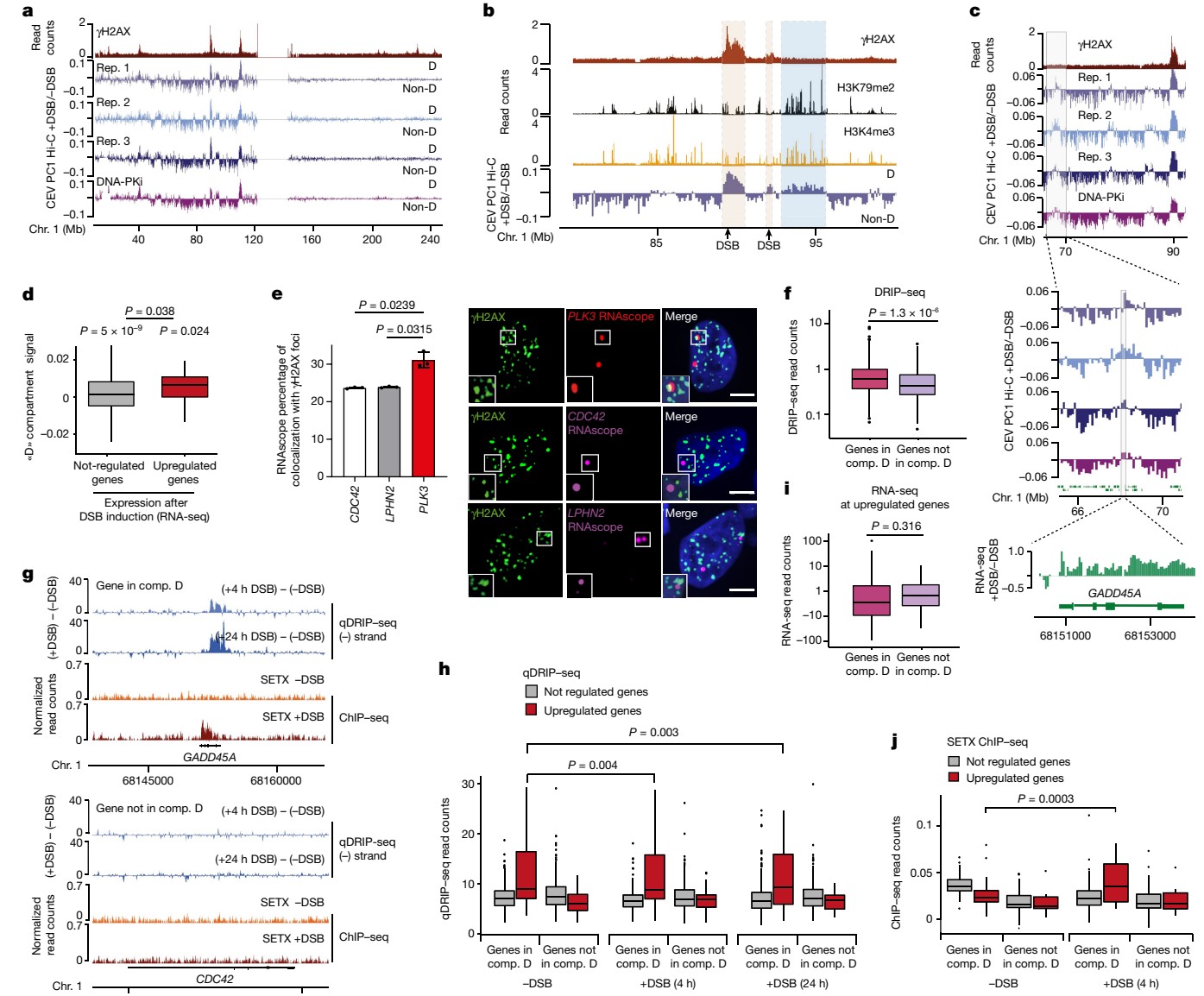

**Fig. 3 | Formation of a DSB-specific compartment (D compartment).**
**a**, γH2AX ChIP-seq and CEV PC1 from Hi-C (+DSB/−DSB) analysis of *n* = 3 biologically independent experiments without DNA-PK inhibition and *n* = 1 with DNA-PK inhibition (DNA-PKi, bottom track). **b**, CEV and ChIP-seq analysis of γH2AX, H3K79me2 and H3K4me3. The rectangles indicate genomic loci in the D compartment (comp.) carrying a DSB (brown) or lacking a DSB (blue). **c**, Magnification of an undamaged region in the D compartment. RNA-seq (log₂[+DSB/−DSB]) analysis is shown. **d**, D compartment (from *n* = 3 biological replicates) on genes from chromosomes 1, 17 and X that are unregulated (*n* = 1,839) or upregulated (*n* = 35) after DSB (Methods). *P* values for each distribution were calculated using two-sided nonparametric Wilcoxon rank-sum tests against *μ* = 0. *P* values between two distributions were calculated using one-sided unpaired nonparametric Wilcoxon rank-sum tests. **e**, Quantification of the colocalization between γH2AX and RNAscope foci after DSB for *LPHN2* and *CDC42* (non-D genes) and *PLK3* (D gene) (left). Data are mean ± s.d. of *n* = 3 biologically independent experiments. *P* values were calculated using paired two-sided *t*-tests. Right, representative examples. **f**, DRIP–seq (+DSB)

analysis of genes in the D compartment (*n* = 493) or not in the D compartment (*n* = 346). *P* values were calculated using two-sided nonparametric Wilcoxon rank-sum tests. **g**, qDRIP–seq (4 and 24 h) and SETX ChIP–seq analysis of a D gene (*GADD45A*) versus a non-D gene (*CDC42*). **h**, qDRIP–seq analysis of unregulated and upregulated genes in the D compartment (*n* = 313 peaks in unregulated genes and *n* = 83 peaks in upregulated genes) or not in the D compartment (*n* = 457 peaks in unregulated genes and *n* = 30 peaks in upregulated genes; Methods). *P* values were calculated using unpaired two-sided nonparametric Wilcoxon rank-sum tests. **i**, RNA-seq counts on upregulated genes in the D compartment (*n* = 40) or not in the D compartment (*n* = 32). *P* values were calculated using unpaired two-sided nonparametric Wilcoxon rank-sum tests. **j**, SETX ChIP–seq analysis of R-loop-enriched genes that are unregulated or upregulated after DSB in the D compartment (*n* = 154 unregulated genes and *n* = 15 upregulated genes) or not in the D compartment (*n* = 123 unregulated genes and *n* = 11 upregulated genes). *P* values were calculated using paired two-sided nonparametric Wilcoxon signed-rank tests.

analysis indicated that the non-damaged loci that segregate with the D compartment are enriched in chromatin marks associated with active transcription, such as H2AZac, H3K4me3 and H3K79me2 (Fig. 3b and Extended Data Fig. 5e). Conversely, these loci targeted to the D compartment displayed a negative correlation with repressive marks such

as H3K9me3 (Extended Data Fig. 5e). A similar trend was observed when the D compartment was computed from the Hi-C data obtained in presence of DNA-PK inhibition (Extended Data Fig. 5e (bottom)). Together, these data indicate that, after DSB induction, the damaged TADs, covered by γH2AX/53BP1, form a new chromatin compartment

that segregates from the rest of the genome. Moreover, additional undamaged loci exhibiting chromatin marks typical of active transcription can be further targeted into this new D compartment.

## DDR genes segregate with the D compartment

To decipher the nature of the active genes targeted to the D compartment, we further examined the DNA motifs that are enriched at genes recruited to the D compartment (D genes), versus 'non-D genes' that do not display targeting to the D compartment (discarding all genes directly comprised in γH2AX domains). Notably, the top enriched motifs included OSR1-, TP73-, Nkx3.1- and E2F-binding sites, which are tumour suppressors and/or are known to be involved in the DDR (Extended Data Fig. 5f), suggesting that DDR genes could be directly targeted to the D compartment. In agreement, visual inspection revealed that some known p53-target genes that are upregulated after DSB induction were associated with the D compartment, even when located as far as >20 Mb from the closest DSB (for example, *GADD45A*; Fig. 3c). To test the hypothesis that DDR genes are recruited to the D compartment, we performed RNA-seq before and after DSB induction and retrieved genes that are significantly upregulated after DSB induction. Notably, genes upregulated after DSB induction displayed a higher D-compartment signal compared with genes that were not upregulated after DSBs (Fig. 3d and Extended Data Fig. 5g). RNAscope analysis, showing transcription sites in nuclei, confirmed that, after DSB induction, *PLK3* and *GADD45A* (identified in the D compartment) significantly colocalized more frequently with γH2AX foci compared with four other non-D genes (Fig. 3e and Extended Data Fig. 5h–j). Notably, only a subset of DDR-upregulated genes was found to be targeted to the D compartment (58% of all upregulated genes). Importantly, the upregulated genes targeted to the D compartment were, on average, not closer to DSBs than the upregulated genes that were not targeted to the D compartment (Extended Data Fig. 5k), ruling out a potential bias due to the genomic distribution of AsiSI DSBs. Together, these data indicate that DSB induction triggers the formation of a chromatin compartment that comprises not only of damaged TADs decorated by γH2AX and 53BP1 but also a subset of DDR genes upregulated after DNA damage.

## Upregulated D genes display R-loops

Non-coding RNAs and R-loops—triple-stranded structures formed by the local retention of transcribed RNA within the DNA double-stranded helix—have previously been suggested to be key regulators of long-range chromosomal contacts, nuclear substructures and chromatin compartments[33–37]. Moreover, local RNA accumulation at DSBs was recently proposed to foster self-segregation of 53BP1 foci[17]. We therefore examined the interplay between D-compartment formation and R-loops. R-loop profiles—determined by S9.6 DNA:RNA hybrid immunoprecipitation followed by high-throughput sequencing (DRIP–seq) in DIvA cells after DSB induction[38]—showed some similarities to the D-compartment profile (Extended Data Fig. 6a). Importantly, at an equivalent expression level (Extended Data Fig. 6b (green track)), DDR-upregulated genes targeted to the D compartment (for example, *GADD45A*, *PLK3* and *JUN*) displayed high levels of R-loops (Extended Data Fig. 6b (blue track)) compared with genes that are not targeted to the D compartment (for example, *CDC42*, *LPHN2* and *ITGB3BP*). On average, genes with high R-loop levels showed a higher D-compartment signal compared with genes with low levels of R-loops (Extended Data Fig. 6c). By contrast, D genes displayed higher R-loop levels compared with non-D genes (Fig. 3f). To carefully evaluate whether R-loops accumulate after DSB induction on D genes, we performed quantitative and strand-specific R-loop mapping (qDRIP–seq[39]) in undamaged cells as well as 4 h and 24 h after DSB induction. As observed using our previous DRIP–seq dataset, *GADD45A* displayed R-loops (Extended Data

Fig. 6d (blue track)), which further accumulated at 4 h and 24 h after DSB induction (Fig. 3g and Extended Data Fig. 6d). This was not the case for *CDC42*, a highly expressed gene that is not targeted to the D compartment (Fig. 3g (bottom) and Extended Data Fig. 6d (right)). On average, the genes that were upregulated after DSB induction and targeted to the D compartment displayed higher R-loop levels compared with upregulated genes that were not targeted to the D compartment, or compared with genes that were not induced by DNA damage (Fig. 3h). Importantly, these upregulated D genes did not display on average elevated RNA-seq signal compared with their counterparts not targeted to the D compartment (Fig. 3i), suggesting that R-loop accumulation rather than transcriptional activity is correlated with gene localization in the D compartment. On these genes, R-loops, moderately but significantly, further accumulated 24 h after DSB induction (Fig. 3h). Notably, senataxin (SETX), a known R-loop binder in mammalian cells, also specifically accumulated at R-loop-enriched upregulated D genes after DSB induction (Fig. 3g (orange and red tracks) and 3j). Together, these data indicate that R-loop accumulation after induction of DSBs is a feature of upregulated DDR genes that are targeted to the D compartment.

## R-loops contribute to D-compartment formation

To investigate whether R-loops contribute to the formation of the D compartment, we first examined the impact of RNase H1 overexpression on DSB clustering measured by high-content microscopy (as performed previously[11]) (Extended Data Fig. 7a). RNase H1 overexpression increased the percentage of cells with a high number of small γH2AX foci, a signature of decreased DSB clustering[11] (Fig. 4a and Extended Data Fig. 7b). Notably, RNase H1 overexpression did not modify the cell cycle distribution (not shown), excluding an effect on DSB clustering through cell cycle changes. Conversely, depletion of RNase H1 increased DSB clustering of AsiSI-induced DSBs (Extended Data Fig. 7c,d). The depletion of RNase H1 also increased the clustering of etoposide-induced DSBs (Extended Data Fig. 7e), suggesting that R-loop involvement in D-compartment formation is not unique to restriction-enzyme-induced DSBs.

DRIP–seq analysis of *SETX*-deficient cells[40] revealed an accumulation of R-loops on D genes after DSB induction (Fig. 4b,c). To determine whether such R-loop accrual after *SETX* depletion modifies D-compartment formation, we performed Hi-C analysis of *SETX*-depleted cells. *SETX* depletion triggered a strong increase in DSB clustering, which was apparent on individual examples (Fig. 4d), on contact heat maps (Extended Data Fig. 7f) as well as on average when computing *trans* interactions between each damaged TAD (Fig. 4e).

Together, these data show that R-loop-forming genes display an increased capacity to be relocated in the D compartment after DSB induction and that the accumulation of R-loops increases D-compartment formation.

## The D compartment contributes to the DDR

Our above data suggest that DDR-upregulated genes that accumulate R-loops display an ability to physically localize within the newly formed D compartment after DSB induction. We postulated that such a recruitment to the D compartment may contribute to their activation after DNA damage. We therefore investigated the consequences of disrupting DSB clustering (and therefore formation of the D compartment) by depleting the SUN2 component of the LINC complex, which was previously found to be a factor in promoting DSB clustering[11,13]. *SUN2* depletion altered the transcriptional activation of upregulated D genes (*GADD45A*, *PLK3*, *RNF19B*; Fig. 4f) but did not modify the induction of other DDR genes (*PPM1D* and *SLC9A1*) that were not identified in the D compartment (Fig. 4f). We could recapitulate these findings using etoposide as another source of DSB induction (Extended Data Fig. 7g).

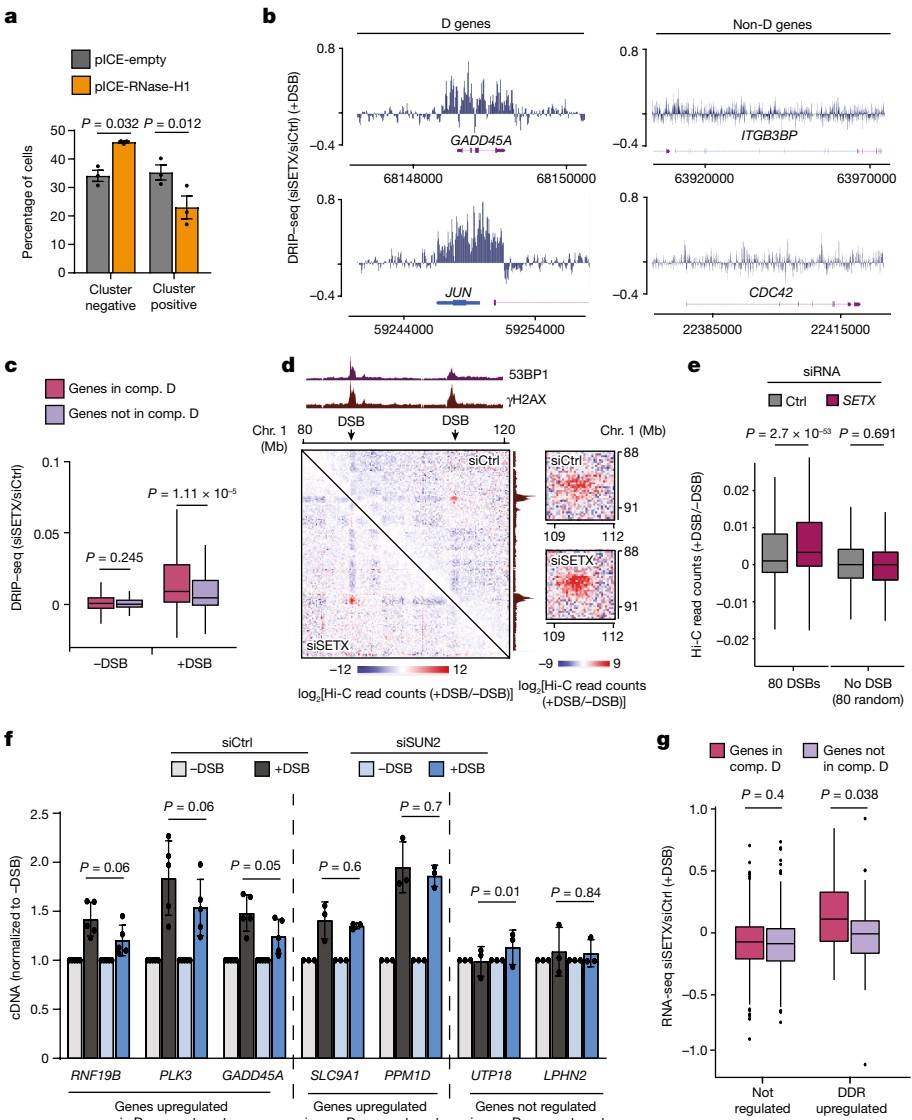

**Fig. 4 | D-compartment formation is fostered after R-loop accrual and triggers DDR-gene activation. a**, Quantification of cluster-negative versus cluster-positive cells after RNase H1 overexpression (pICE-RNase-H1, yellow) or not (pICE-empty, grey). The percentages of cells in each category are indicated. Data are mean ± s.d. of n = 3 biologically independent experiments. P values were calculated using paired two-sided t-tests. **b**, Genomic track of log$_2$[DRIP–seq (*SETX* siRNA/control siRNA)] obtained after DSB (24 h) on D genes (*GADD45A* and *JUN*; left) and non-D genes (*ITGB3BP* and *CDC42*; right). **c**, DRIP–seq log$_2$[siRNA *SETX*/siRNA control][40] after DSB (24 h) on D genes (n = 493) and non-D genes (n = 346). P values were calculated using two-sided nonparametric Wilcoxon tests. **d**, Differential Hi-C contact matrix (log$_2$[+DSB/−DSB]) in DIvA cells that were transfected with control or *SETX* siRNA (250 kb resolution). γH2AX and 53BP1 ChIP–seq (+DSB) tracks are shown. Right, magnification showing Hi-C contacts between the two γH2AX domains. **e**, Differential Hi-C read counts (log$_2$[+DSB/−DSB]) between damaged chromatin domains (DSBs, n = 80) or undamaged sites (random; n = 80) in control or *SETX*-depleted conditions. P values were calculated using two-sided paired nonparametric Wilcoxon signed-rank tests. **f**, qPCR with reverse transcription (RT–qPCR) analysis of seven genes (*RNF19B*, *PLK3*, *GADD45A*, *SLC9A1*, *PPM1D*, *UTP18* and *LPHN2*) before and after DSB induction in DIvA cells that were transfected with control or *SUN2* siRNA. Data are mean ± s.e.m. of n = 5 (*RNF19B*, *PLK3* and *GADD45A*) and n = 3 (*SLC9A1*, *PPM1D*, *UTP18* and *LPHN2*) biologically independent experiments. All genes are normalized to *RPLPO*. P values were calculated using paired two-sided t-tests. **g**, log$_2$[*SETX*/control] RNA-seq read counts in control and *SETX* siRNA transfected cells after DSB, on genes that are unregulated or upregulated after DSB induction in compartment D (n = 453 unregulated genes and n = 40 upregulated genes) or not in compartment D (n = 314 unregulated genes and n = 32 upregulated genes). P values were calculated using unpaired two-sided nonparametric Wilcoxon rank-sum tests.

Together, these data suggest that the physical targeting of a subset of DDR genes to the D compartment is required to ensure their optimal activation.

We further analysed whether, conversely, enhanced D-compartment formation could increase transcriptional activation of D genes by performing RNA-seq in *SETX*-deficient cells, as *SETX* depletion triggered increased D-compartment formation (Fig. 4d,e). On average, *SETX* depletion enhanced the transcription of DDR-upregulated genes (Extended Data Fig. 7h,i). Notably, the genes upregulated after *SETX* depletion in undamaged cells did not display an increased D-compartment signal compared with genes that were unaffected by *SETX* depletion (Extended Data Fig. 7j), suggesting that enhanced transcription in *SETX*-depleted cells is not responsible for targeting to the D compartment, in agreement with our above data showing that D genes are not more expressed than non-D genes (Fig. 3i). Importantly, *SETX* depletion selectively triggered enhanced expression of DDR-upregulated D genes compared with those not targeted to the D compartment (Fig. 4g). Combined, these data show that the

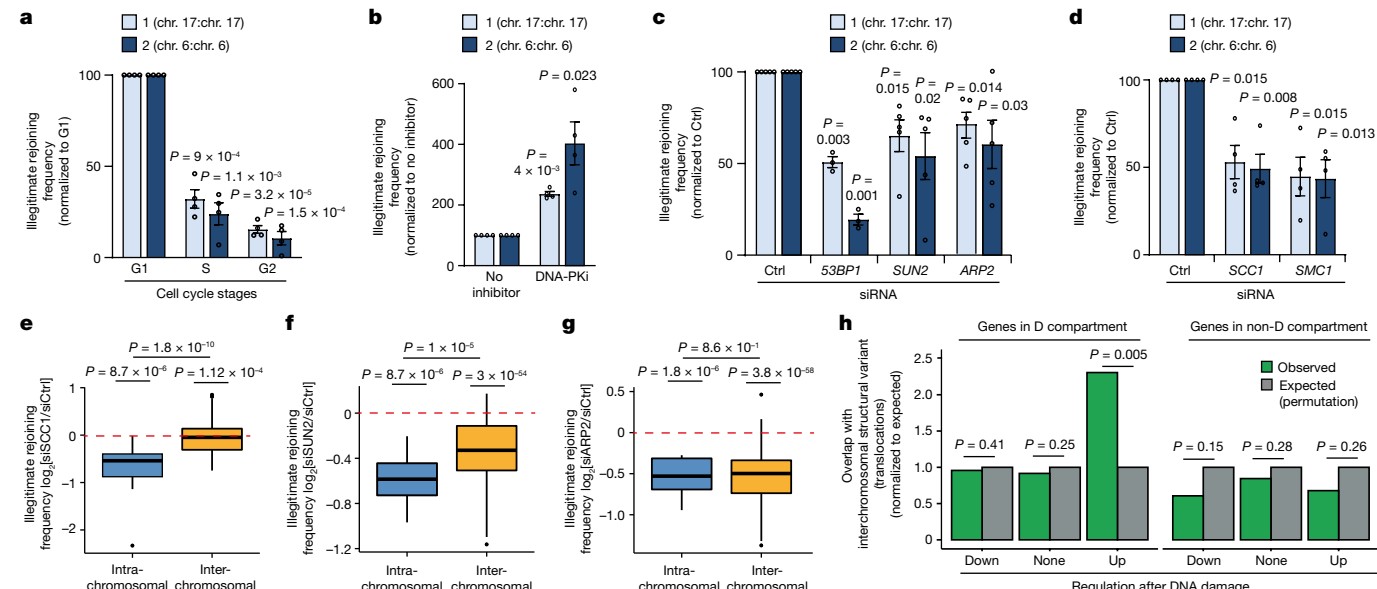

**Fig. 5 | DSB-induced loop extrusion and D-compartment formation drive translocations. a**, qPCR quantification of two genomic rearrangements after DSB induction in DIvA cells synchronized in G1, S or G2 phase. Data are mean ± s.e.m. of $n$ = 4 biologically independent experiments. $P$ values were calculated using paired two-sided $t$-tests. **b**, qPCR quantification of two genomic rearrangements after DSB induction with or without DNA-PK inhibitor. Data are mean and s.e.m. of $n$ = 4 biologically independent experiments. $P$ values were calculated using paired two-sided $t$-tests. **c**, qPCR quantification of two genomic rearrangements in control or 53BP1, SUN2 or ARP2 depleted cells. Data are mean ± s.e.m. of $n$ = 3 (53BP1) and $n$ = 5 (ARP2 and SUN2) biologically independent experiments. $P$ values were calculated using paired two-sided $t$-tests. **d**, qPCR analysis as in **c** but after control, SMC1 or SCC1 depletion. Data are mean ± s.e.m. of $n$ = 4 biologically independent experiments. $P$ values were calculated using paired two-sided $t$-tests. **e**, Intrachromosomal rearrangements (blue, $n$ = 46) or interchromosomal translocations (yellow, $n$ = 354) between 20 DSBs induced in DIvA cells after control or SCC1 depletion ($\log_2$[SCC1 siRNA/control siRNA]).

$n$ = 4 biologically independent experiments. **f**, Intrachromosomal rearrangements and interchromosomal translocations as in **e**, but the quantification was performed in SUN2-depleted cells. $n$ = 4 biologically independent experiments. **g**, Intrachromosomal rearrangements and interchromosomal translocations as in **e**, but the quantification was performed in ARP2-depleted cells. $n$ = 4 biologically independent experiments. For **e**–**g**, $P$ values were calculated using nonparametric Wilcoxon rank-sum tests (against $\mu$ = 0) and a two-sided unpaired Wilcoxon rank-sum test (intrachromosomal versus interchromosomal). **h**, The observed (green) and expected (grey, obtained through 1,000 permutations) overlap between the breakpoint positions of interchromosomal translocations identified on cancer genomes and genes targeted to the D compartment that are upregulated, downregulated or not regulated after DSB induction (as determined using RNA-seq) as indicated, compared with their counterparts that are not targeted to the D compartment. $P$ values were calculated using one-sided resampling tests (Methods).

disruption of the D compartment impairs the activation of a subset of DDR genes and that, conversely, enhanced D-compartment formation triggers the hyperactivation of DDR genes. This suggests a role for the D compartment, that is, DSB clustering, in the activation of DDR genes.

## DSB-induced chromosome folding favours translocations

Importantly, although our above data suggest a beneficial role of DSB clustering in potentiating the DNA damage response, it may also be detrimental, as bringing two DSBs into close proximity may foster translocations (illegitimate rejoining of two DSBs)[12,15]. In agreement, we showed that SETX depletion, which enhances D-compartment formation (Fig. 4d,e), increased translocation[38]. Notably, DSB illegitimate rejoining events were increased in G1-synchronized cells compared with in S/G2-synchronized cells (Fig. 5a), in agreement with the enhanced DSB clustering observed in G1 cells (Fig. 2j and Extended Data Fig. 3f). DNA-PK inhibition, which fosters D-compartment formation (Fig. 2h and Extended Data Figs. 3c and 5b), also increased illegitimate rejoining frequency (Fig. 5b). By contrast, depletion of 53BP1, SUN2 and ARP2, an actin branching factor (Extended Data Fig. 8a), all previously reported to mediate repair focus fusion and DSB clustering[11,13,14,16], decreased illegitimate rejoining (Fig. 5c). Notably, depletion of the cohesin subunits SMC1 or SCC1 also decreased the illegitimate rejoining frequency (Fig. 5d and Extended Data Fig. 8b). This was unexpected as

SCC1-depleted cells did not display clustering defects (Extended Data Fig. 3a,b).

Given that our quantitative PCR (qPCR)-based assay assesses two intrachromosomal DSB illegitimate rejoining events (that is, rejoining of two distant DSBs located on the same chromosome), we hypothesized that the illegitimate rejoining frequency at the intrachromosomal level may also be regulated by the DSB-induced loop extrusion that depends on the cohesin complex. To investigate more broadly translocation events between multiple DSBs induced in the DIvA cell line (intra- and interchromosomal), we designed a multiplexed amplification protocol followed by next-generation sequencing[41]. In control cells, we could readily detect increased translocation frequency after induction of DSBs compared with at control genomic locations (Extended Data Fig. 8c). Notably, depletion of SCC1 decreased the frequency of intrachromosomal rejoining events, while leaving interchromosomal translocations unaffected (Fig. 5e). By contrast, depletion of SUN2 and ARP2 decreased both intra- and interchromosomal rejoining events (Fig. 5f,g). Taken together, these data suggest that both the DSB-induced loop extrusion and the formation of the D compartment through clustering of damaged TADs display the potential to generate translocations.

Given our above finding that a subset of genes upregulated after DSB induction can be physically targeted to the D compartment (Fig. 3), we further hypothesized that such a physical proximity may account for some of the translocations observed on cancer genomes. We retrieved breakpoint positions of interchromosomal translocations of 1,493

individuals across 18 different cancer types (from a previous study[42]), and assessed their potential overlap with genes targeted to the D compartment. D-targeted genes were further sorted as upregulated, downregulated or not significantly altered after DSB induction, and compared to their counterparts that are not targeted to the D compartment. We found that genes that are upregulated after DSB induction and that are targeted to the D compartment displayed a significant overlap with translocation breakpoints, in contrast to genes that are not targeted to the D compartment (Fig. 5h). Together, these data indicate that the relocalization of DDR-upregulated genes in the D compartment may account for some of the translocations detected on cancer genomes. Given that DDR genes comprise a number of tumour suppressor genes, such a physical proximity of these genes with DSBs within the D compartment formed in response to DNA damage may be a key mechanism driving oncogenesis, through fostering the instability of tumour suppressor genes.

## Conclusion

Together, we show that DSB-induced changes in chromosome architecture is an integral component of the DDR, but also acts as a double-edged sword that can challenge genomic integrity through the formation of translocations.

Our data suggest that a chromatin compartment (D compartment) arises when γH2AX/53BP1-decorated domains, established by ATM-induced loop extrusion after DSB induction, self-segregate from the rest of the chromatin. This may occur through a PPPS mechanism[43], which reorganizes the folding of damaged chromosomes through bridging interactions between 53BP1-decorated domains. Similar models have been recently proposed for the formation of cohesin–DNA clusters[28] and heterochromatin foci[25]. Notably, a PPPS mechanism is also consistent with previous research reporting that 53BP1 can self-interact and that 53BP1 foci can fuse[16,17,44]. The properties of this DSB-induced D compartment, lacking PAR, may therefore differ from the properties of DDR foci at early timepoints that are formed through PAR- and FUS-dependent LLPS[27,45], within which 53BP1 accrual may indeed rely on its ability to phase separate independently of a chromatin scaffold[16] (Fig. 2g). This D compartment further recruits a subset of genes involved in the DDR, especially those prone to form R-loops, and subsequently contributes to their activation (Extended Data Fig. 8d). Additional live imaging of individual loci and high-throughput immunoDNA-FISH will further help to determine the prevalence and dynamics of DDR genes targeted to the D compartment. This model is in agreement with previous research that identified that 53BP1 is critical for p53-target gene activation[46] and checkpoint activation[16]. It also agrees with the recent finding that *AHNAK* depletion triggers an elevated p53 response due to enhanced 53BP1 chromatin binding and condensate formation[18] as does the loss of TIRR, a protein that regulates 53BP1 association to DSBs[47]. Notably, R-loop accrual fosters D-compartment formation in agreement with recent data indicating (1) that non-coding RNAs and R-loops are central regulators of chromosome architecture[33–37] and (2) that DSB-induced non-coding RNA promotes 53BP1 focus formation and fusion[17].

We propose that the physical targeting of R-loop-enriched DDR-activated genes to the D compartment enables precise fine-tuning of the magnitude of the DDR with respect to DSB load and persistency, therefore providing a function for the enigmatically large γH2AX/53BP1-decorated chromatin domains and for DSB clustering. Yet, it comes at the expense of potential translocations, as both loop extrusion and coalescence of damaged TADs are able to bring linearly distant DSBs into close physical proximity (Extended Data Fig. 8d). Importantly, we found that the genes that are upregulated in response to DSBs and relocated to the D compartment displayed significant overlap with translocation breakpoints identified by whole-genome

sequencing in cancer samples from patients. In agreement with an increased occurrence of structural variants on tumour suppressor genes[42], we propose that the physical targeting of DDR genes to the D compartment, in close spatial proximity to DSBs, may occasionally trigger deleterious rearrangements on genes involved in the control of cell proliferation and apoptosis after DNA damage, and may therefore act as a critical driver of oncogenesis by disrupting the integrity of tumour suppressor genes.

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

# Methods

## Cell culture and treatments

DIvA (AsiSI-ER-U2OS)[19], AID-DIvA (AID-AsiSI-ER-U2OS)[23] and 53BP1-GFP DIvA[20] cells were developed in U2OS (ATCC HTB-96) cells and were previously described. Authentication of the U2OS cell line was performed by the provider ATCC, which uses morphology and short tandem repeat profiling to confirm the identity of human cell lines. DIvA, AID-DIvA and 53BP1-GFP-DIvA cells derived from these U2OS cells were not further authenticated. All of the cell lines were regularly tested for absence of mycoplasma contamination using the MycoAlert Mycoplasma (Lonza). All of the cell lines were grown in Dulbecco's modified Eagle's medium (DMEM) supplemented with 10% SVF (Invitrogen), antibiotics and either 1 µg ml$^{-1}$ puromycin (DIvA cells) or 800 µg ml$^{-1}$ G418 (AID-DIvA and 53BP1-GFP AID-DIvA) at 37 °C under a humidified atmosphere with 5% $CO_2$. To induce DSBs, U2OS cells were treated with etoposide (Sigma-Aldrich, E1383) for 4 h at 500 nM and DIVA or AID-DIvA cells were treated with 300 nM OHT (Sigma-Aldrich, H7904) for 4 h (supplementary table 1 of ref. 5 for positions). For ATM or DNA-PK inhibition, cells were respectively pretreated for 1 h with 20 µM KU-55933 (Sigma-Aldrich, SML1109) or 2 µM NU-7441 (Selleckchem, S2638) and during subsequent OHT treatment. For cell synchronization, cells were incubated for 18 h with 2 mM thymidine (Sigma-Aldrich, T1895), then released during 11 h, followed by a second thymidine treatment for 18 h. S, G2 and G1 cells were then respectively treated with OHT at 0, 6 or 11 h after thymidine release and collected 4 h later. siRNA and plasmid transfections were performed using the 4D-Nucleofector kit and the SE cell line 4D-Nucleofector X L or S kit (Lonza) according to the manufacturer's instructions, and subsequent treatment(s) were performed 48 h later. siRNA transfections were performed using a control siRNA (siCtrl): CAUGUCAUGUGUCACAUCU; or siRNA targeting *SCC1* (siSCC1): GGUGAAAAUGGCAUUACGG; *SMC1* (siSMC1): UAGGCUUCCUGGAGGUCACAUUUAA; *53BP1* (si53BP1): GAACGAGGAGACGGUAAUA; *SUN2* (siSUN2): CGAGCCUAUUCAGACGUUUCA; *ARP2* (siARP2): GGCACCGGGUUUUGUGAAGU; *SETX* (siSETX): GAGAGAAUUAUUGCGUACU; and *RNASEH1* (siRNASEH1)[48]: CUGUCUUGCUGCCUGUUACU. pICE-NLS-mCherry (referred to as pICE-empty in the text; Addgene plasmid, 60364) and pICE-RNase-H1-WT-NLS-mCherry (referred to as pICE-RNase-H1 in the text, Addgene plasmid, 60365) were respectively used as control and for RNase H1 over-expression (gifts from P. Calsou and S. Britton).

## Illegitimate rejoining assay

Illegitimate rejoining assays after siRNA transfection were performed at least in triplicate in AID-DIvA cells as described previously[38]. Illegitimate rejoining assays in synchronized cells were performed in DIvA cells after an OHT treatment ($n = 4$ biological replicates). The genomic DNA was extracted from $1–5 × 10^6$ cells using the DNeasy Blood & Tissue Kit (Qiagen), according to the manufacturer instructions. Two different illegitimate rejoinings between different AsiSI sites were assessed by qPCR using the following primers: 1Fw, GACTGGCATAAGCGTCTTCG and 1Rev, TCTGAAGTCTGCGCTTTCCA; and 2Fw, GGAAGCCGCCCAGAATAAGA and 2Rev, TCCATCTGTCCCTATCCCCAA. The results were normalized using two control regions, both far from any AsiSI sites and γH2AX domains using the following primers: Ctrl_chr1_82844750_Fw, AGCACATGGGATTTTGCAGG and Ctrl_chr1_82844992_Rev, TTCCCTCCTTTGTGTCACCA; and Ctrl_chr17_9784962_Fw, ACAGTGGGAGACAGAAGAGC and Ctrl_chr17_9785135_Rev, CTCCATCATCGCACCCTTTG. Normalized illegitimate rejoining frequencies were calculated using Bio-Rad CFX Manager v.3.1.

## RT–qPCR

RNA was extracted from fresh DIvA cells before and after DSB induction using the RNeasy kit (Qiagen). RNA was then reverse transcribed to cDNA using AMV reverse transcriptase (Promega, M510F). qPCR experiments were performed to assess the levels of cDNA using primers targeting *RPLP0* (FW, GGCGACCTGGAAGTCCAACT; REV, CCATCAGCACCACAGCCTTC), *RNF19B* (FW, CATCAAGCCATGCCCACGAT; REV, GAATGTACAGCCAGAGGGGC), *PLK3* (FW, GCCTGCCGCCGGTTT; REV, GTCTGACGTCGGTAGCCCG), *GADD45A* (FW, ACGATCACTGTCGGGGTGTA; REV, CCACATCTCTGTCGTCGTCC). *PPM1D* (FW, CTTGTGAATCGAGCATTGGG; REV, AGAACATGGGGAAGGAGTCA), *SLC9A1* (FW, TGTTCCTCAGGATTTTGCGG; REV, ATGAAGCAGGCCATCGAGC), *LPHN2* (FW, CGATTTGAAGCAACGTGGGA; REV, TGATACTGGTTGGGGAAGGG), *UTP18* (FW, TCCTACTGTTGCTCGGATCTC; REV, ATGAAGCAGGCCATCGAGC), *IPO9* (FW, CACAGATGCCACTTGTTGCT; REV, TGCTGTACCACGGGAAAGAT) or *PARP1* (FW, ATTCTGGACTGGAACACTCTGC; REV, CTGTTCCAGTTTGTTGCTACCG). cDNA levels were then normalized to *RPLP0*, then expressed as the percentage of the undamaged condition.

## Immunofluorescence

DIvA cells were grown on glass coverslips and fixed with 4% paraformaldehyde during 15 min at room temperature. A permeabilization step was performed by treating cells with 0.5% Triton X-100 in PBS for 10 min, then cells were blocked with PBS-BSA 3% for 30 min. Primary antibodies against γH2AX (JBW301, Millipore Sigma, 05-636) were diluted 1:1,000 in PBS-BSA 3% and incubated with cells overnight at 4 °C. After washes in 1× PBS, cells were incubated with anti-mouse secondary antibodies (conjugated to Alexa 594 or Alexa 488, Invitrogen), diluted 1:1000 in PBS-BSA 3%, for 1 h at room temperature. After staining with DAPI, Citifluor (Citifluor, AF-1) was used for coverslip mounting. Images were acquired with the software MetaMorph, using the ×100 objective of a wide-field microscope (Leica, DM6000) equipped with a camera (DR-328G-C01-SIL-505, ANDOR Technology). The spatial analysis (Ripley function) of γH2AX spot distribution was performed as described previously[20] using the spot detector plug-in in Icy.

## High-content microscopy

Transfected DIvA and U2OS cells were plated in 96-well Cell Carrier plates (Perkin Elmer). Cells were subjected to OHT or etoposide treatments and an immunofluorescence as described above using anti-mouse Alexa 488 secondary antibodies (Invitrogen) for γH2AX detection. Cells were stained with 5 µg ml$^{-1}$ Hoechst 33342 (Invitrogen) for 30 min at room temperature. γH2AX foci were analysed using the Operetta CLS High-Content Imaging System (Perkin Elmer) and Harmony software (v.4.9). For quantitative image analysis, 81 fields per well were acquired with a ×40 objective lens to visualize around 4,000 cells per well in triplicate. Subsequent analyses were performed with Columbus software (v.2.8.2). Determination of the number (y axis) and size of γH2AX foci (mean area on the x axis) in each nucleus was been used to infer clustering (clustering leads to a low number of large foci) (>8,000 nuclei analysed per sample). Scatter plots are divided into four quadrants on the basis of the median number and size of foci and the percentages of cell populations that are cluster positive (bottom right quadrant) or cluster negative (top left quadrant) are determined.

## 3D RIM super-resolution acquisition and reconstruction

3D + t live RIM was performed on AID-DIvA cells expressing 53BP1–GFP[20] using an upgrade of the system and method described previously[49]. In brief, 60 3D images were acquired during 47 min using an inverted microscope (TEi Nikon) equipped with a ×100 magnification, 1.49 NA objective (CFI SR APO 100XH ON 1.49 NIKON) and SCMOS camera (ORCA-Fusion, Hamamatsu). The temporal resolution of all of the Supplementary Videos is 40 s. A commercial acquisition software (INSCOPER SA) enables a whole-cell single-timepoint 3DRIM acquisition in only 6 s under a low-photobleaching regime (1 W cm$^{-2}$). Fast diode lasers (Oxxius) with the wavelengths centred at 488 nm (LBX-488-200-CSB) were used to produce a TEM00 2.2-mm-diameter beam. The polarization beam was rotated with an angle of 5° before hitting a X4 Beam Expander beam (GBE04-A) and produced a 8.8 mm TEM00 beam.

A fast spatial light phase binary modulator (QXGA fourth dimensions) was conjugated to the image plane to create 48 random illumination by each plane as described previously[49]. 3D image reconstruction was then performed as described previously[49] and at GitHub (https://github.com/teamRIM/tutoRIM).

## 3D film editing

Bleaching correction was performed after RIM reconstruction using open-source FIJI software (https://imagej.net/software/fiji/) based on exponential FIT from the background signal. The 3D drift correction FIJI plugin was performed for 3D registration (https://github.com/fiji/Correct_3D_Drift). 3D + *t* video rendering was generated using the VTK library implemented in ICY software (https://icy.bioimageanalysis.org/) from a 3D crop of the area of interest.

## FRAP measurements

FRAP experiments were performed on 53BP1–GFP foci using a Zeiss LSM 710 confocal light scanning microscope (Carl Zeiss) equipped with a ×63/1.2 NA oil-immersion objective. Typically, images were acquired at 512 × 256 px at a scan speed corresponding to 200 ms per image, and 300 images were acquired over 2 min, with an interval of 500 ms between subsequent images. Before photobleaching half of the foci, three images were recorded.

For each experiment, a custom R script was used to segment the image, track the bleached foci and retrieve the average intensity of the bleached half ($I_B$), the non-bleached half ($I_{NB}$), the background ($I_{BG}$), and the rest of the nucleus ($I_{REF}$) at each frame. These intensity values were used to calculate FRAP curves for the bleached half ($FRAP_B$) and the non-bleached half ($FRAP_{NB}$), according to ref. 24:

$$FRAP^I_{B/NB}(t) = \frac{I_{B/NB}(t) - I_{BG}(t)}{I_{REF}(t) - I_{BG}(t)} + A$$

Here, $A$ represents unwanted bleaching in the non-bleached half. $FRAP_B$ and $FRAP_{NB}$ were multiplied by the size of their respective regions of interest (ROIs) ($N_B$ and $N_{NB}$, respectively) to obtain curves that are proportional to the number of particles in each half

$$FRAP^{II}_{B/NB}(t) = FRAP^I_{B/NB}(t) \frac{N_{B/NB}}{N_B + N_{NB}}$$

The curves were then normalized with respect to the number of bleached molecules:

$$FRAP^{III}_{B/NB}(t) = \frac{FRAP^{II}_{B/NB}(t) - FRAP^{II}_{B/NB}(t_{bleach})}{FRAP^{II}_B(t_{pre}) - FRAP^{II}_B(t_{bleach})}$$

Here, $t_{pre}$ and $t_{bleach}$ are the acquisition times of the last frame before the bleach and the first frame after the bleach, respectively. The resulting FRAP curves are proportional to the ROI sizes and double-normalized. Finally, an additive offset was applied to the signal in the non-bleached half to normalize to unity before the bleach.

The resulting curves reflect the change in the number of labelled molecules in each half. In the presence of an immobile fraction of molecules that do not move during the course of the experiment because they tightly bind to immobile binding sites, the signal in both halves will not recover to the same level, but there will be an offset between them that corresponds to the immobile fraction $X_{immobile}$. To correct for those immobile molecules, the FRAP curves are modified according to:

$$FRAP^{IV}_{NB}(t) = 1 + \frac{FRAP^{III}_{NB}(t)}{1 - X_{immobile}}, \ FRAP^{IV}_B(t) = \frac{FRAP^{III}_B(t)}{1 - X_{immobile}}$$

Finally, to determine the presence or absence of an interfacial barrier in the foci, the curves were compared to half-FRAP experiments performed in a solution of freely diffusing poly-lysine-fluorescein (0.1 mg ml⁻¹) and to half-FRAP experiments of 53BP1-GFP molecules diffusing in the nucleoplasm (adjacent to foci). In this case, the maximum decrease in normalized fluorescence down to a value of 0.90 ± 0.03 indicates the absence of a barrier (horizontal black lines in Fig. 2f,g) while larger decreases indicate the presence of an interfacial barrier.

## DNA damage induction by laser irradiation

DIvA cells were transfected with a plasmid encoding the macrodomain of macroH2A1.1 fused to mKate2 (macro–mKate2)[50] and treated with OHT for 4 h. For microirradiation, a circular spot with a radius of 1 μm was selected in the nucleoplasm where 53BP1–GFP foci were absent. Then, continuous illumination with a 405 nm laser at maximum power was applied for 1 min. The recruitment of 53BP1–GFP and macro–mKate2 was observed during the next 10 min. Half-FRAP experiments at these laser-induced 53BP1–GFP foci were performed as described above after 5 min of their formation.

## 53BP1 focus fusion quantification

Fusion events were selected from the 3D images acquired by RIM. During the fusion, the foci were segmented and the normalized aspect ratio, or eccentricity, was calculated as $AR_{norm} = (R_{max} - R_{min})/(R_{max} + R_{min})$, where $R_{max}$ and $R_{min}$ are the longest and shortest radius, respectively. The normalized aspect ratio over time was then fitted with an exponential function with an additive offset to obtain the relaxation constant ($t_R$) and the aspect ratio of the end product after fusion. $t_R$ values obtained from five different experiments were plotted versus the size of the respective foci after the fusion and fitted with a linear function, the slope of which is the inverse capillary velocity.

## RNAscope

RNAscope enables the visualization of transcription sites[51,52]. DIvA cells were grown on Chambered Cell Culture Slides (Corning Falcon, 08-774-25) and fixed with 4% paraformaldehyde during 15 min at room temperature. The RNAscope assay was performed using the RNAscope Multiplex Fluorescent Kit v2 kit (ACDBio, 323100) according to the manufacturer's instructions. In brief, fixed cells were pretreated with RNAscope Hydrogen Peroxide for 10 min, then permeabilized with RNAscope Protease III (1:15 diluted) for 10 min. Cells were then incubated with the probes RNAscope Probe-Hs-GADD45A-C1 (ACDBio, 477511) and RNAscope Probe-Hs-CCL2-C2 (ACDBio, 423811-C2) or RNAscope Positive Control Probe-Hs-PPIB-C2 (ACDBio, 313901-C2), or with the intronic probes RNAscope Probe-Hs-PLK3-intron-C1 (ACDBio #1263411) and RNAscope Probe-Hs-CDC42-intron-C2 (ACDBio, 1263101) or RNAscope Probe-Hs-ADGRL2-intron-C2 (ACDBio, 1263111) in a HybEZTM Oven at 40 °C for 2 h. Signal-amplification steps were performed, followed by the development of the HRP-C1 and HRP-C2 signals, using Opal dye 620 and 690 (Akoya Biosciences) diluted to 1:750. Finally, immunofluorescence was performed as described above, without an additional permeabilization step and using γH2AX (JBW301, Millipore) and DAPI staining. Images were acquired with the software Micro-Manager, using the ×40 or ×60 objective of a spinning-disk/high-speed widefield CSU-W1 microscope, equipped with an Andor Zyla sCMOS camera. The colocalization between γH2AX foci and RNA foci was measured using Cell Profiler. Two foci were considered to be colocalizing when part of their areas was overlapping and if their respective centroids were separated by less than 1 μm.

## Western blot

AID-DIvA cells were incubated in RIPA buffer (50 mM Tris at pH 8, 150 mM NaCl, 0.5% deoxycholate, 1% NP-40, 0.1% SDS) on ice for 20 min and centrifuged at 13,000 rpm for 10 min. The supernatant, containing soluble protein extracts, was then mixed with SDS loading buffer and reducing agent, resolved on 3–8% NuPAGE Tris-acetate gels (Invitrogen) and transferred onto PVDF membranes (Invitrogen)

according to the manufacturer's instructions. For RNase H1 expression, $0.5 \times 10^6$ DIvA cells were incubated 15 min at room temperature with 625 U of GENIUS Nuclease (Santa Cruz Biotechnology, sc-202391) in SDS loading buffer. After adding a reducing agent, the samples were heated at 95 °C for 5 min and loaded onto a NuPAGE 4–12% Bis-Tris gel in MOPS SDS running buffer and transferred onto a PVDF membrane (Invitrogen) with the Trans-Blot Turbo Transfer System according to the manufacturer's instructions (Bio-Rad). PVDF membranes were incubated in TBS containing 0.1% Tween-20 (Sigma-Aldrich, P1379) and 3% non-fat dry milk for 1 h for blocking, followed by overnight incubation at 4 °C using primary antibodies targeting SUN2 (Abcam, ab124916, 1:1,000), ARP2 (Abcam, ab128934, 1:1,000), 53BP1 (Novus Biologicals, NB100-305, 1:1,000), RNase H1 (Invitrogen, PA5-30974, 1:1,000), SCC1 (Abcam, ab992, 1:500), SMC1 (Abcam, ab75819, 1:1,000), myosin I-β (Sigma-Aldrich, M3567, 1:1,000), GAPDH (Sigma-Aldrich, MAB374, 1:10,000) or α-tubulin (Sigma-Aldrich, T6199, 1:10,000). The corresponding mouse or rabbit horseradish-peroxidase-coupled secondary antibodies were used at 1:10,000 to reveal the proteins (Sigma-Aldrich, A2554 and A0545), using a luminol-based enhanced chemiluminescence HRP substrate (Super Signal West Dura Extended Duration Substrate, Thermo Fisher Scientific). Picture acquisition of the membranes was performed using the ChemiDoc Touch Imaging System and pictures were visualized using Image Lab Touch software.

### RNA-seq
RNA-seq in DIvA cells was performed as described previously[38]. Raw sequencing data were mapped in paired-end to a custom human genome (hg19 merged with ERCC92) using STAR. Count matrices were extracted using htseq-count with union as resolution-mode and reverse-strand mode. Differential expression analysis was performed on the count matrix using edgeR with two replicates per condition (with or without 4 h OHT treatment) and differential genes were determined using log-ratio test. Whole-genome coverage was computed using the bamCoverage command form deeptools to generate bigwig from BAM files (without PCR duplicate suppression). Differential coverage between two conditions was performed using BamCompare from deeptools with setting the binsize parameter at 50 bp. The $\log_2$-transformed fold change was calculated using edgeR in differential expression analysis. Using a cut-off adjusted $P$ value of 0.1 and a $\log_2$-transformed fold change of 0.5 (~41% increase/decrease of expression), we were able to determine 286 upregulated and 125 downregulated genes with 11 of them directly damaged by a DSB. On chromosomes 1, X and 17, $n = 35$ genes were found to be upregulated and $n = 1,839$ not regulated (Fig. 3d). On chromosomes 1, 2, 6, 9, 13, 17, 18, 20 and X, $n = 86$ genes were found upregulated and $n = 3,829$ not upregulated (Extended Data Fig. 5g). For further classification in the D and non-D compartment, to analyse enough genes, the cut-off for upregulation was set at a $\log_2$-transformed fold change of 0.3 (see below).

### qDRIP–seq
qDRIP–seq was adapted from a previous study[39]. In brief, $2.5 \times 10^6$ of trypsinized DIvA cells were mixed with $1.67 \times 10^6$ *Drosophila* S2 cells and lysed overnight at 37 °C in TE buffer containing 0.5% SDS and 800 µg proteinase K (Roche, 03115828001). DNA was extracted by phenol–chloroform extraction using phase-lock tubes (Qiagen, MaxTract, 129065) followed by ethanol precipitation. DNA was resuspended on ice in 130 µl TE buffer before sonication in the Covaris S220 system (microtubes, PN520045) to obtain ~300 bp DNA fragments (Covaris S220, 140 W peak incident power, 10% duty factor, 200 cycles per burst for 80 s). Immunoprecipitation was performed in triplicate by incubating 4 µg of sonicated DNA with 10 µg of S9.6 antibody (Antibodies Incorporation) in 1× binding buffer (10 mM NaPO₄, 140 mM NaCl, 0.05% Triton X-100) overnight and at 4 °C. Agarose A/G beads (Thermo Fisher Scientific, 20421) were added to samples for 2 h at 4 °C. Beads were washed four times in binding buffer followed by incubation with

elution buffer (50 mM Tris pH 8, 10 mM EDTA, 0.5% SDS, 0.3 µg µl⁻¹ proteinase K) for 45 min at 55 °C. The samples were subjected to phenol–chloroform extraction and ethanol precipitation, and were resuspended in low EDTA TE. Sequencing libraries were prepared using the Swift ACELL-NGS 1S Plus kit according to the manufacturer's instructions using 12 PCR cycles. Libraries were pooled at equimolar concentrations and sequenced using the Illumina NextSeq 500 system with 75 paired-end reads.

qDRIP–seq data were processed using a custom pipeline taking into account stranded and spike-in library preparation. In brief, reads were trimmed using Trimmomatic (v.0.39)[53] to remove remaining primers from the library. BWA-MEM was used for mapping reads to a custom reference genome merging hg19 and dm6 (spike-in) chromosomes. Samtools was used to generate BAM files with reads based on their mapping location (hg19 or dm6). Strand-specific data were generated using Samtools view and merge with flags filters: 80;160 for reverse fragments, and 96;144 for forward fragments. BAM files were then sorted, indexed and duplicates were removed. Bigwig files were generated on these data, normalized to total read counts (counts per million) or by the number or reads mapped on dm6 (spike-in). Differential coverage between two conditions was performed using BigWigCompare from deeptools with the substract setting[54] and with setting bin size parameter at 50 bp. Narrow peaks were detected using macs3 callpeak algorithm[55] on qDRIP bams using -q 0.1, and by taking only good quality peaks with a score (fold-change at peak summit) at least superior to 100.

### DRIP–qPCR
DRIP–qPCR was performed as described previously[38] using primers for *RPL13A* (FW, AATGTGGCATTTCCTTCTCG; REV, CCAATTCGGCCAA GACTCTA) and *PLK3* (FW, CGGAGCAGAGGAAGAAGTGA; REV, CATGCAT GAACAGCCCATCA).

### Amplicon–seq
AID-DIvA cells were treated with or without 300 nM OHT for 4 h followed by treatment with indole-3-acetic acid for 14 h to degrade AsiSI[23]. The genomic DNA was extracted from $5 \times 10^6$ cells using the DNeasy Blood & Tissue Kit (Qiagen) according to the manufacturer's instructions. Genomic DNA was then used in a multiplex PCR reaction that amplified 25 target sites: 20 AsiSI cut sites and 5 uncut control sites (Extended Data Table 1). Amplicons were size-selected using SPRIselect beads (Beckman, B23318) and processed for DNA library preparation using the NEBNext Ultra II kit (NEB, E7645L). Libraries were pooled at equimolar concentrations and sequenced using the Illumina NextSeq 500 system with paired-end 150 cycles. The data were analysed using our custom tool mProfile, available at GitHub (https://github.com/aldob/mProfile). This identified the genomic primers used in the original genomic PCR reaction to amplify each read in the pair. Translocated reads were therefore identified as those where each read in a pair was amplified by a different primer set, and this was normalized to the total reads that were correctly amplified by these primer sets. The heat map of illegitimate rejoining/translocation events was made between 20 DSBs by computing the ratio between the +DSB and −DSB sample for each pair of DSBs and comparing the $\log_2$-transformed ratio distribution between each condition and control. Significance was computed using nonparametric Wilcoxon tests.

### 4C-seq
4C-seq experiments performed in synchronized cells, before and after DSB induction, were performed as described previously[7]. In brief, $10-15 \times 10^6$ DIvA cells per condition were cross-linked, lysed and digested with MboI (New England Biolabs). DNA ligation was performed using the T4 DNA ligase (HC) (Promega), and ligated DNA was digested again using NlaIII (New England Biolabs). Digested DNA was religated with the T4 DNA ligase (HC) (Promega) before proceeding to 4C-seq library preparation. A total of 16 individual PCR reactions was

performed to amplify around 800 ng of 4C-seq template, using inverse primers including the Illumina adaptor sequences and a unique index for each condition (Extended Data Table 2). Libraries were pooled and sent to a NextSeq 500 platform at the I2BC Next Generation Sequencing Core Facility (Gif-sur-Yvette).

4C-seq data were processed as described previously[7]. In brief, BWA-MEM was used for mapping and Samtools was used for sorting and indexing. A custom R script (https://github.com/bbcf/bbcfutils/blob/master/R/smoothData.R) was used to build the coverage file in bedGraph format, to normalize using the average coverage and to exclude the nearest region from each viewpoint. Differential 4C-seq data were computed using BamCompare from deeptools with binsize=50 bp. The average of total *trans* interactions between viewpoints and DSBs was then computed using a 1 Mb window around the breaks (80 best) and after exclusion of viewpoint–viewpoint (*cis*) interactions.

## Hi-C

Hi-C data obtained before and after DSB induction and after control or *SCC1* depletion in DIvA cells were retrieved from a previous study[7]. Hi-C experiments with or without DSB induction and after ATM or DNA-PK inhibition, or after transfection with control or *SETX* siRNAs were performed in DIvA cells as described previously[7]. In brief, $10^6$ cells were used per condition. Hi-C libraries were generated using the Arima Hi-C kit (Arima Genomics) according to the manufacturer's instructions. DNA was sheared to an average fragment size of 350–400 bp using the Covaris S220 system and sequencing libraries were prepared on beads using the NEB Next Ultra II DNA Library Prep Kit for Illumina and NEBNext Multiplex Oligos for Illumina (New England Biolabs) according to the instructions of the Arima Hi-C kit.

## Hi-C data analyses

**Hi-C heat maps.** Hi-C reads were mapped to the hg19 genome and processed using Juicer with the default settings (https://github.com/aidenlab/juicer). Hi-C count matrices were generated using Juicer at multiple resolutions: 100 kb, 50 kb, 25 kb, 10 kb and 5 kb. Hi-C heat map screenshots were generated using Juicebox (https://github.com/aidenlab/Juicebox/wiki/Download). Aggregate heat maps were computed on a set of submatrices extracted from the originally observed Hi-C matrices at 50 kb resolution or 100 kb resolution. The region of 5 Mb around DSBs (80 best) was extracted and then averaged. The $\log_2$-transformed ratio was then computed using Hi-C counts (+DSB/−DSB) and plotted as heat maps.

***Cis* contact quantification.** For *cis* contact quantification, interactions within γH2AX domains (−0.5/+0.5 Mb around the 80 best DSBs) were extracted from the observed Hi-C matrix at 100 kb resolution, and the $\log_2$-transformed ratio was computed on damaged versus undamaged Hi-C counts (+DSB/−DSB). Adjacent windows (−1.5 Mb−0.5 Mb and +0.5 Mb–1.5 Mb around 80 best DSBs) were retrieved to quantify interactions between damaged domains and adjacent undamaged domains.

***Trans* contact quantification.** To determine interaction changes in *trans* (interchromosomal), we built the whole-genome Hi-C matrix for each experiment by merging together all chromosome–chromosome interaction matrices using Juicer and R. The result is a genome matrix with 33,000 × 33,000 bin interactions for 100 kb resolution. Interactions between bins inside damaged TADs (240 × 240 for 80 DSBs) were extracted and counted for each condition, the $\log_2$-transformed ratio was calculated on normalized counts (counts per million), and plotted as box plots or heat maps. For the box plots, the centre line shows the median; the box limits show the first and third quartiles; the whiskers show the maximum and minimum values without outliers; and the points show the outliers. For the heat maps, each tile corresponds to $\log_2$[+DSB/−DSB] between DSB (100 kb bins within

±1 Mb regions were averaged). They were further sorted on the basis of the 53BP1 ChIP–seq level (Extended Data Fig. 7f), according to previously determined homologous-recombination-prone and non-homologous-end-joining-prone DSBs[5] (Extended Data Fig. 2d), or to the level of PC1 determined by applying PCA in Hi-C DIvA to identify A/B compartments (Extended Data Fig. 4d).

**APA on endogenous breaks.** Endogenous breaks were identified by calling peaks on pATM ChIP–seq data without OHT treatment[7] using macs2 with -q 0.01, giving 1,206 narrow peaks. Of these peaks, only interchromosomal pairs (*trans*) were retained using a BEDPE format file. Random positions were generated using the gkmsvm package and pairs of interactions were built according to the same procedure. APA was then performed on these *trans* pairs using juicertools.

**TAD cliques.** TAD cliques were computed using the igraph R package on an undirected graph representing DSB clustering. This graph was computed on the differential Hi-C matrix (+DSB/−DSB) counts, at 500 kb resolution, considering a change of around 86% of interaction (0.9 in $\log_2$) between two DSBs as a node on the graph. Averaged ChIP–seq signal values (53BP1/γH2AX/H1/Ubiquitin FK2) were then computed for each category of cliques using 500 kb windows around DSBs. For prior RNA polymerase II occupancy, the signal was computed on 10 kb around DSBs.

**A/B compartment.** To identify the two mains chromosomal compartments (A/B), the extraction of the first eigenvector of the correlation matrix (PC1) was performed on the observed/expected matrix at 500 kb resolution using the juicer eigenvector command. The resulting values were then correlated with the ATAC-seq signal to attribute positive and negative values to the A and B compartment, respectively, on each chromosome. The observed/expected bins were arranged on the basis of the PC1 values and aggregated into 21 percentiles, to visualize A–B interactions on our experiments (saddle plots).

**D compartment.** To identify the D compartment, we retrieved the first component (PC1) of a PCA made on the differential observed Hi-C matrix $\log_2\left(\frac{\text{damaged}}{\text{undamaged}}\right)$ at 100 kb resolution. Each matrix was extracted from the .hic files using Juicer and the ratio was computed bin per bin. Pearson correlation matrices were then computed for each chromosome and PCA was applied on each matrix. The first component of each PCA was then extracted and correlated with the positions of DSB. A PC1 showing a positive correlation with DSB was then called the D compartment and PC1 showing a negative correlation with DSBs were multiplied by −1. We were able to extract the D compartment on chromosomes 1, 17 and X for +DSB/−DSB and chromosomes 1, 2, 6, 9, 13, 17, 18, 20 and X for +DSB/−DSB in the DNA-PKi condition. The D compartment (first component of the PCA) was converted into a coverage file using the rtracklayer R package. Using the same package, D compartment value were computed around DSBs and genes at 100 kb resolution.

**Determination of D genes.** First, all genes embedded in γH2AX domains (that is, −1 Mb/+1 Mb around the 80 DSBs) were removed from the gene set. Under normal conditions (without DNA-PKi), on chromosomes 1, 17 and X, $n = 493$ genes displaying a positive PC1 value on their entire length in each of the three Hi-C replicates experiments were identified as 'genes in compartment D' genes, while $n = 346$ genes displaying a negative value in each of the three Hi-C replicates experiments were labelled as 'genes not in compartment D'. In the presence of DNA-PKi, where the D compartment was identified on chromosomes 1, 2, 6, 9, 13, 17, 18, 20 and X, $n = 2,161$ were found in compartment D, while $n = 2,112$ were not. The genes were further categorized according to their upregulation after DSB induction (fold change > 0.3, unexpressed genes filtered), giving four categories. Without DNA-PKi, upregulated/compartment D ($n = 40$); upregulated/no compartment D ($n = 32$);

downregulated and not regulated/compartment D ($n = 453$); down-regulated and not regulated/no compartment D ($n = 314$).

**Transcription factor motif analysis.** Transcription-factor-binding motifs were extracted on the promoter regions (−500 bp of the transcription start site (TSS)) of genes with positive value of D compartment (2,161) versus genes with negative value (2,112) identified on chromosomes 1, 2, 6, 9, 13, 17, 18, 20 and X from the PCA analysis of Hi-C (+DSB + DNA-PKi/−DSB) using motifmatchr and TFBSTools R packages on the JASPAR2020 database. Motifs were sorted by significance using Fisher's exact test and adjusted using the Benjamini–Hochberg procedure between motifs found on gene inside the D compartment versus genes outside D compartment.

**Correlation with DRIP–seq or qDRIP datasets.** To assess a correlation between R-loop accrual and D-compartment formation, R-loop levels obtained from DRIP–seq experiments performed in DIvA cells[38] were computed on extended gene bodies (±2 kb TTS) and plotted as a box plot for categories (Figs. 3f and 4c). Conversely, to establish whether R-loop enriched genes, display higher levels of differential CEV (D compartment) signal (Extended Data Fig. 6c), R-loop levels were computed on all genes of chromosome 1, 17 and X (±1 kb of the TTS) and further categorized into 10 groups (based on percentiles). The D-compartment signal was compared between the highest ($n = 180$) and lowest ($n = 190$) groups.

For qDRIP experiments, given that the signal accumulates on narrow peaks within genes (in contrast to DRIP–seq), we used the identified locations of narrow peaks (see the 'qDRIP–seq' section) inside the genes in each category (±2 kb of the TTS) (Fig. 3h). We were able to identify 83 peaks on upregulated D genes, 313 peaks on not upregulated D genes, 30 peaks on upregulated non-D genes and 457 peaks on not-upregulated non-D genes.

**Correlation with SETX ChIP-seq data.** ChIP-seq SETX data in DIvA cells were from a previous study[38]. To assess SETX accruals at R-loops, BigWig coverage files were used to get SETX ChIP-seq coverage on gene bodies (±2 kb of the TTS) that overlap with qDRIP peaks (see above) (Fig. 3j).

**Translocation breakpoints.** For translocation breakpoints, data from a previous study[42] were retrieved, and only breakpoints for interchromosomal structural variants were selected ($n = 28,051$). Genes reproducibly enriched in the D compartment in the three biological replicates, on chromosomes 1, 17 and X as well as genes not enriched in the D compartment were retrieved. The significance of the overlap between genes and breakpoints was determined using the regioneR package[56] by using resampling test with PermTest. In brief, we selected 1,000 times a control set of genes, with the same size and on the same chromosome as our original gene set. We tested the overlap between each gene and breakpoints to determine a distribution of the number of overlaps between control set and breakpoints. We further tested whether the overlap between our gene set (D compartment or non-D compartment) and breakpoints was significant by counting the number of times more overlaps occurred in the control set than in our gene set.

GenomicRanges, plyranges, tidyverse, patchwork, ggforce, ggside and ggtext were used to read, manipulate and visualize genomic data in R and produce figures. Bedtools was used to manipulate genomic location and produce bed or bigwig files. Integrated Genome Browser was used to visualize bed and bigwig files. All of the box plots show the median (centre line), first and third quartiles (box limits), maximum and minimum without outliers (whiskers) and outliers (points).

**Reporting summary**

Further information on research design is available in the Nature Portfolio Reporting Summary linked to this article.

## Data availability

All high-throughput sequencing data (Hi-C, 4C–seq, amplicon-seq, qDRIP–seq and RNA-seq) performed in this study have been deposited at Array Express under accession number E-MTAB-10865. Other high-throughput sequencing data used in this study are available under accession numbers E-MTAB-8851 (Hi-C data before and after DSB induction and after control or *SCC1* depletion; pATM ChIP–seq); E-MTAB-5817 (BLESS before and after DSB induction; LIG4, 53BP1, γH2AX, FK2, histone H1 and H3K79me2 ChIP–seq experiments); and E-MTAB-6318 (DRIP–seq before and after DSB induction; SETX ChIP–seq). Breakpoint positions of interchromosomal translocations across 18 different cancer types were retrieved from ref. 42.

## Code availability

Source code is available at GitHub (https://github.com/LegubeDNARE-PAIR/ and https://github.com/bbcf/bbcfutils/blob/master/R/smoothData.R).

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

**Acknowledgements** We thank the members of the Genomics Core Facility of EMBL and the High-Throughput Sequencing Facility of I2BC for high-throughput sequencing as well as the staff at the Light Microscopy (LITC) Facility at the CBI; S. Britton and P. Calsou for the pICE-RNaseHI-WT-NLS-mCherry plasmid; and L. Ligat from the CRCT Cell Imaging Platform (INSERM-UMR1037) for technical assistance in high-content microscopy. RNAscope data for this study were acquired at the Center for Advanced Light Microscopy—Nikon Imaging Center at UCSF on the W1-CSU confocal system obtained using a NIH S10 Shared Instrumentation grant (1S10OD017993-01A1). M.B. and A.S.B. were supported by Cancer Research UK core funding to the Beatson Institute (A31287) and to the Bushell laboratory (A29252); A.S.B. by the National Productivity Award from the MRC (MC_ST_U17040). Funding in the G.L. laboratory was provided by grants from the European Research Council (ERC-2014-CoG 647344 and ERC-AdG-101019963), the Agence Nationale pour la Recherche (ANR-18-CE12-0015), the Association Contre le Cancer (ARC), the ITMO Cancer (RiDR), the association Robert Debré and the Fondation Bettencourt-Schueller. C.A. was a recipient of a FRM fellowship (FRM FDT201904007941) and is a recipient of EMBO postdoctoral fellowship (EMBO ALTF 585-2021). Funding in the F.E. laboratory was provided by a grant from the European Research Council (ERC-2018-StG 804023). F.S. is a recipient of a LNCC fellowship. S.C. is a recipient of a PhD fellowship from the Joint Training and Research Programme on Chromatin Dynamics & the DNA Damage Response (H2020 ITN aDDRess, grant no. 812829). T.C. and N.P. are INSERM researchers.

**Author contributions** C.A., E.L., T.C., F.S., A.M. B.L.B. and N.P. performed and analysed experiments. V.R., S.C. and R.M. performed bioinformatic analyses of all high-throughput sequencing datasets. T.M. performed RIM acquisition and analysis. C.A. performed RNAscope under the supervision of N.A. A.S.B. performed the amplicon-seq experiment under the supervision of M.B. D.N. helped to realize and to analyse 4C-seq experiments. F.M. acquired half-FRAP data under the supervision of F.E. G.L. wrote the manuscript with the help of A.M. and T.C. All of the authors commented on and edited the manuscript.

**Competing interests** N.A. is a cofounder and on the scientific advisory board of Regel Therapeutics and receives funding from BioMarin Pharmaceutical Inc. The other authors declare no competing interests.

**Additional information**
**Correspondence and requests for materials** should be addressed to Gaëlle Legube.

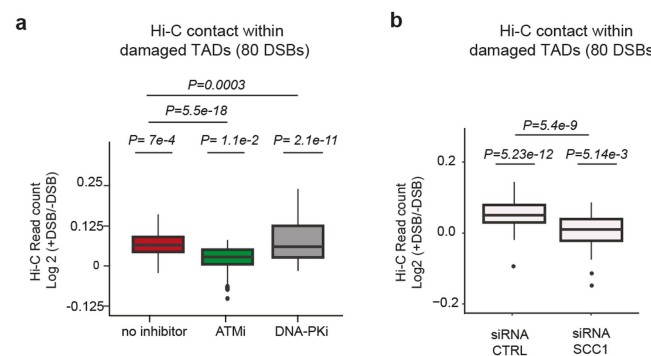

**Extended Data Fig. 1 | Related to Fig. 1. ATM and cohesin-dependent local changes within damaged TADs. a,** Boxplot showing the differential Hi-C read counts log2 (+DSB/−DSB) within −0.5/ +0.5 Mb regions containing the 80 best-induced DSBs ($n$ = 66 damaged domains, overlapping domains excluded, red) without inhibitors, upon ATM inhibition (ATMi) or upon DNA-PK inhibition (DNA-PKi). For each distribution (+vs −DSB), $P$ values were calculated using paired two-sided non-parametric Wilcoxon signed-rank tests. No inhibitor *vs* ATMi or DNA-PKi, $P$ values were calculated using two-sided unpaired Wilcoxon rank-sum tests. **b,** same as in **a** but in cells transfected with a control (CTRL) or *SCC1* siRNA. $P$ values were calculated using two-sided paired non-parametric Wilcoxon signed-rank test. si*SCC1* vs siCTRL, $P$ value was calculated using two-sided unpaired Wilcoxon rank-sum test.

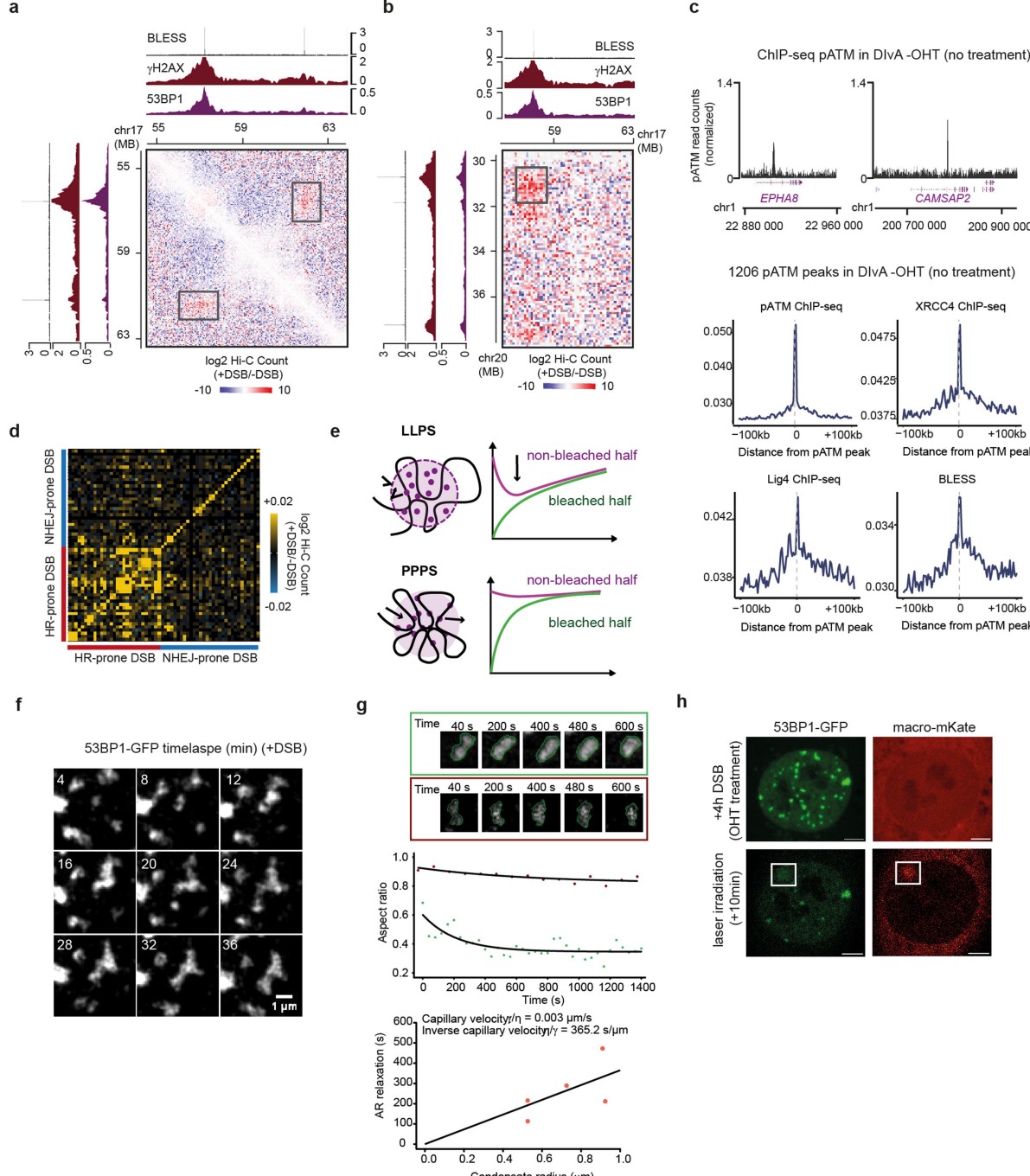

**Extended Data Fig. 2** | See next page for caption.

**Extended Data Fig. 2 | Related to Fig. 2. Late DSB clustering agrees with Polymer-Polymer phase separation. a**, Hi-C contact matrix showing the log2(+DSB/−DSB) on a region of chromosome 17 at 50 kb resolution. Genomic tracks for γH2AX, 53BP1 ChIP-seq and BLESS following DSB induction are shown on the top panel. One representative experiment is shown. **b,** Same as in **a** but showing Hi-C contacts between a region of chromosome 17 and a region of chromosome 20. **c,** Top panels: Genomic track of pATM ChIP-seq performed in DIvA cells without OHT treatment (no AsiSI-induced DSB), showing endogenous DSB hotspots. Middle and Bottom panels: pATM, XRCC4, Lig4 average ChIP-seq profiles and BLESS (as indicated) on 1206 endogenous DSBs hotspots retrieved by peak calling on pATM ChIP-seq. **d,** Differential Hi-C heatmaps (+DSB/−DSB) computed on DSBs previously identified as HR- or NHEJ- prone[5]. One representative experiment is shown. **e,** Schematic representation of a focus formed through 53BP1 LLPS (top) or PPPS (bottom). In the case of LLPS, the phase boundary will favour 53BP1 diffusion within the foci and hinder its diffusion across the boundary. This leads to preferential internal mixing that can be observed as a decrease of the fluorescence in the non-bleached half (marked by a black arrow). In the case of PPPS, the phase separation of the foci is driven by chromatin bridging-interactions. In the latter case, 53BP1 is recruited through binding without a phase boundary, and unbound 53BP1 molecules can diffuse between nucleoplasm and foci without restriction. Then, the recovery of fluorescence is primarily due to molecules entering from the nucleoplasm, and only a small intensity decrease in the non-bleached half is observed. **f,** 53BP1-GFP DIvA cells were recorded 4 h after DSB induction using 4 min intervals for 1 h. In this example the fusion does not trigger the formation of a round-shaped condensate. **g,** Quantification of fusions of 53BP1-GFP foci. Top/middle panels: The normalized aspect ratio of the 53BP1-GFP condensates was determined during fusion events imaged by RIM, and its relaxation was fitted to an exponential decay. A representative event for a condensate that relaxes to a round shape (top snapshots and green points) and for a condensate that does not relax to a round shape (bottom snapshots and red points) are shown. Bottom panel: The time constant for these decays is plotted against the size of the condensates to obtain the inverse capillary velocity (η/γ) that characterizes the fusion of these foci, $n = 5$. **h,** Confocal images of 53BP1-GFP (green channel) and macro-mKate2 (macro-domain of macroH2A1.1 fused to mKate2, PAR-sensor, red channel) accumulation at micro-irradiation sites 10 min after micro-irradiation (bottom panel) and 4 h after OHT addition (top panel). Scale bar, 5 μm. $n = 4$ biologically independent experiments.

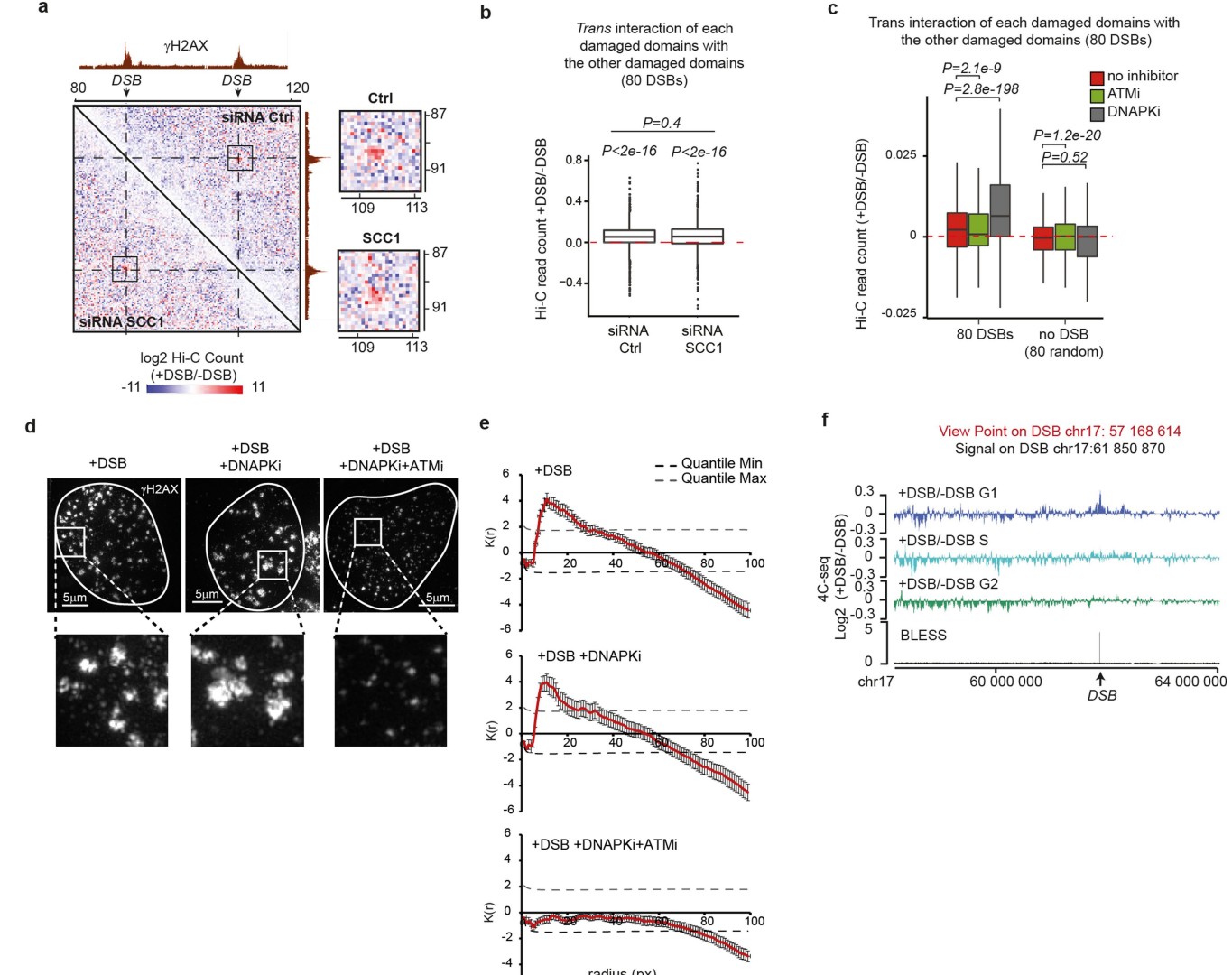

**Extended Data Fig. 3 | Related to Fig. 2. DSB clustering depends on ATM but not on DNA-PK and cohesin, and is enhanced in G1. a,** Left panel: Hi-C contact matrix of the log2(+DSB/−DSB) upon Ctrl (upper right) or SCC1 depletion (lower left). A region of chromosome 1 is shown at 250 kb resolution. The γH2AX ChIP-seq track following DSB induction is shown on the top and on the right. Right panel: magnification of the black squares, showing Hi-C contacts between the two γH2AX domains. **b,** Boxplot showing the quantification of the differential Hi-C read counts log2 (+DSB/−DSB) on 100 kb bins between the 80 most-damaged chromatin domains in control or *SCC1*-depleted conditions (*cis* contacts were excluded) (*n* = 6320). *P* values for each distribution were calculated using non-parametric Wilcoxon rank-sum tests tested against μ = 0. *P* value between two distributions was calculated using non-parametric two-sided unpaired Wilcoxon rank-sum test. **c,** Boxplot showing the quantification of the differential Hi-C read counts (log2(+DSB/−DSB)) between the 80 most-damaged chromatin domains (*n* = 6320 100 kb bins) or random undamaged sites (no DSB *n* = 6320 100 kb bins) in cells after DSB induction without inhibitor (red),

in presence of ATM inhibitor (green), or DNA-PK inhibitor (grey) (*cis* contacts were excluded). *P* values were calculated using paired two-sided non-parametric Wilcoxon signed-rank test. **d,** γH2AX staining in DIvA cells after DSB induction without inhibitor, in presence of DNA-PK inhibitor, or with ATM and DNA-PK inhibitors, acquired using Random Illumination Microscopy (RIM). Magnifications are shown in the bottom panels. Scale bar, 5 μm. *n* = 3 biologically independent experiments. **e,** Spatial analysis (Ripley) performed on γH2AX foci in DIvA cells after DSB induction without inhibitor (*n* = 18 nuclei from *n* = 3 biologically independent experiments), in presence of DNA-PK inhibitor (*n* = 22 nuclei from *n* = 3 biologically independent experiments), or with ATM and DNA-PK inhibitors (*n* = 17 nuclei from *n* = 3 biologically independent experiments). Mean and s.e.m. are shown. **f,** Genomic tracks showing the differential 4C-seq signal (log2 (+DSB/−DSB)) at the DSB located on chr17: 61 850 870 in G1 phase (blue), S phase (turquoise) and G2 phase (green) smoothed using a 10 kb span. The BLESS signal after DSB induction is also shown. 4C-seq was performed using a DSB located on chr17:57 168 614 as a viewpoint. (*n* = 1 experiment).

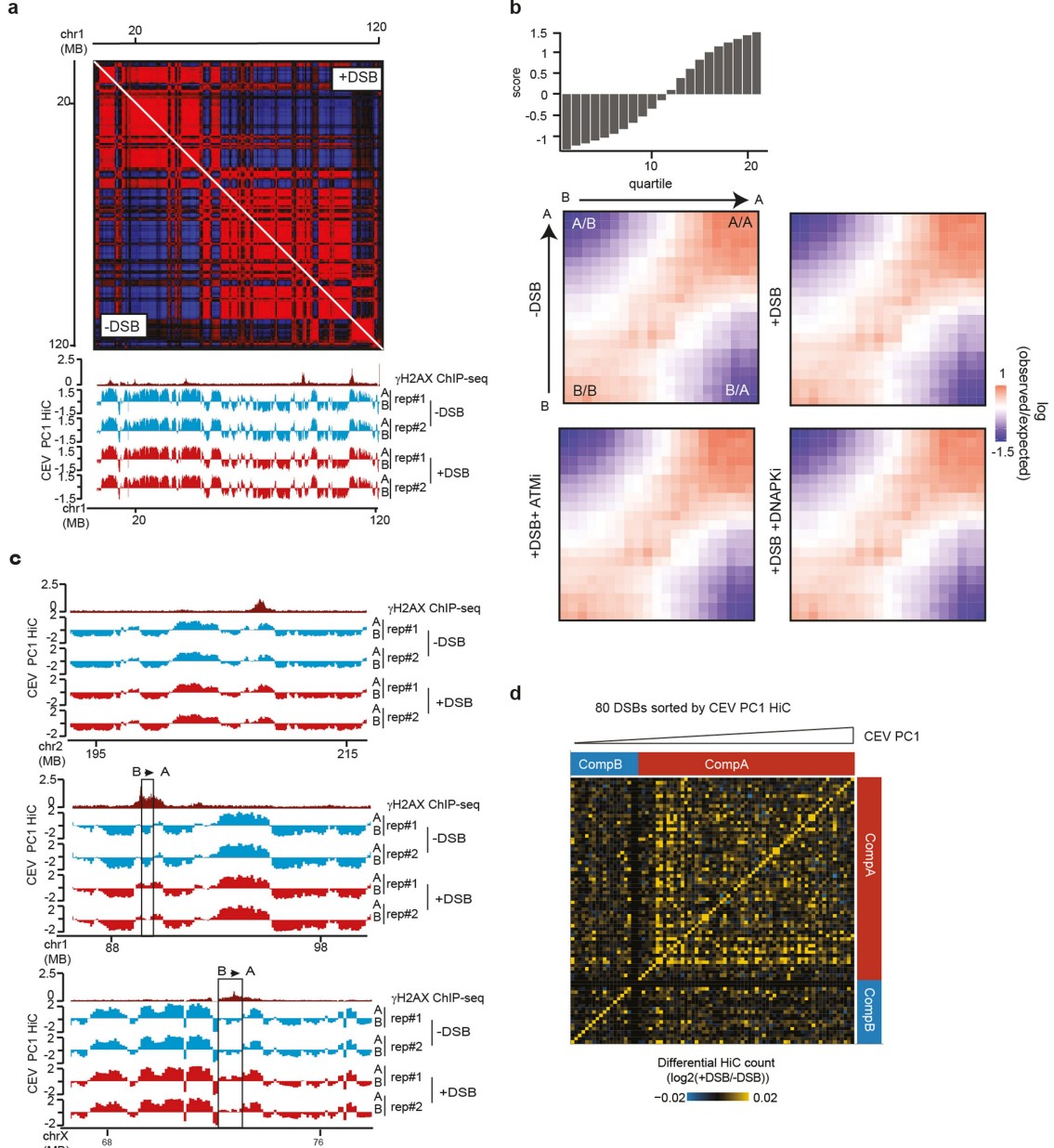

**Extended Data Fig. 4 | Related to Fig. 3. DSB induction does not trigger major changes of A/B compartmentalization. a,** Genome browser (Juicebox) screenshot showing the Pearson correlation matrix of Hi-C count on the left arm of chromosome 1, before and after DSB induction as indicated (500 kb resolution). A representative experiment is shown. The bottom panel shows the A/B compartment (PC1) retrieved from two biological replicates of Hi-C experiments before (blue) and after (red) DSB induction on the same chromosomal region. The γH2AX ChIP-seq (+DSB) track is also shown (dark red). **b,** Saddle plots of Hi-C data obtained before DSB, after DSB, and after DSB in presence of ATM inhibitor or DNA-PK inhibitor as indicated, showing interactions between pairs of 500 kb bins sorted according to their eigenvector value (PC1). **c,** Genomic tracks of the eigenvector (PC1) obtained from two

biological replicates of Hi-C experiments before (blue) and after (red) DSB induction. The γH2AX ChIP-seq (+DSB) track is shown on the top (dark red) and DSBs are indicated by arrows. The top panel shows an example of a DSB induced in the "A" compartment (positive PC1) which stays in the "A" compartment following DSB induction. The middle and bottom panels show examples of two DSBs which occur in the "B" compartment (negative PC1) and switch to the "A" compartment following DSB induction. **d,** Heatmap showing the differential (log2 +DSB/−DSB) Hi-C read count for each of the 80 DSBs (1 Mb window). DSBs were sorted according to their Eigenvector value computed on 40 kb around the DSB. DSBs located in compartment A tend to cluster more. A representative experiment is shown.

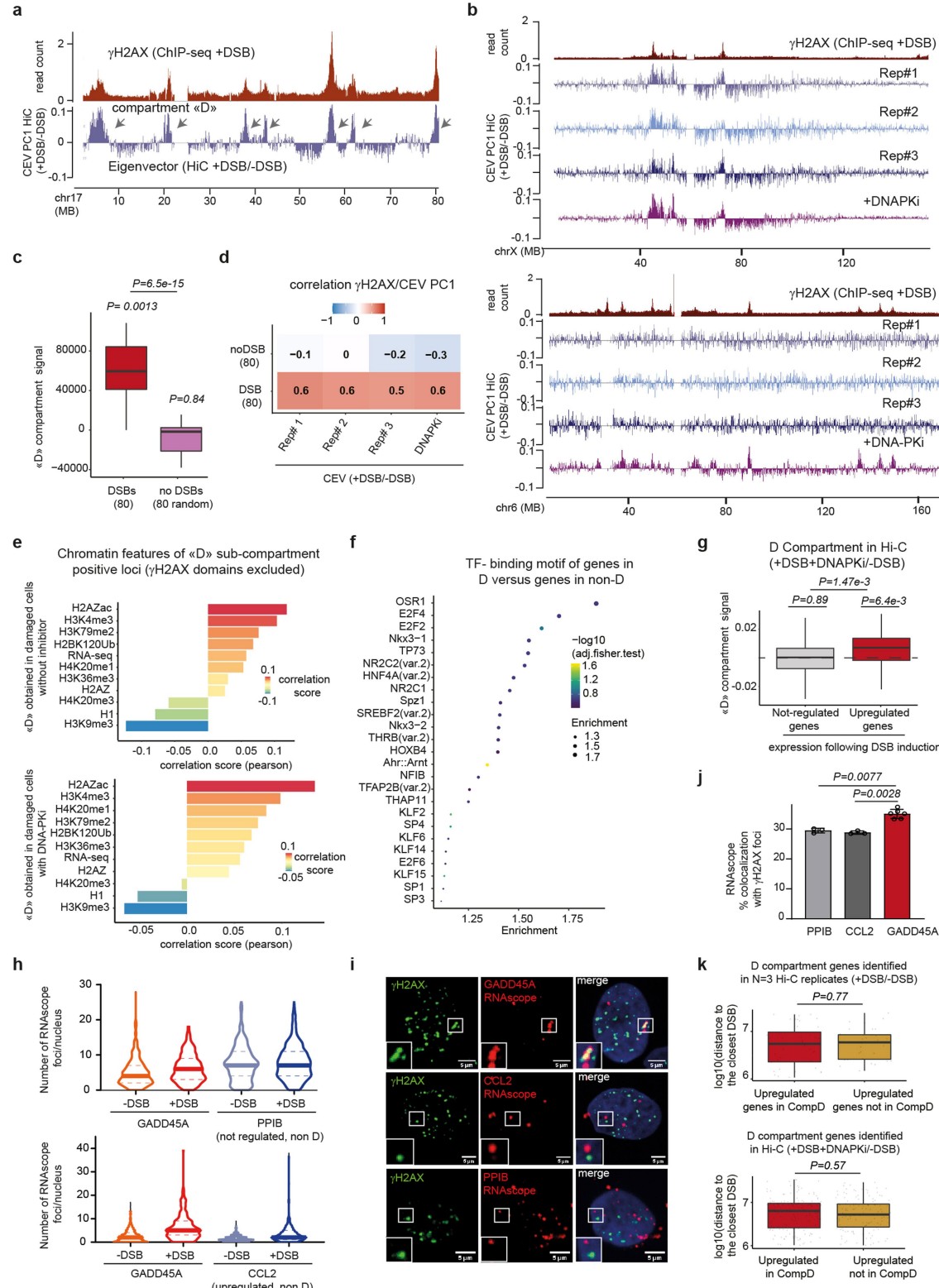

**Extended Data Fig. 5 |** See next page for caption.

**Extended Data Fig. 5 | Related to Fig. 3. A subset of DDR genes segregates with a DSB-induced compartment. a,** Genomic tracks of γH2AX ChIP-seq (+DSB) and first Chromosomal eigenvector (CEV) computed on differential (+DSB/−DSB) Hi-C matrix of chromosome 17 using a 100 kb resolution (blue). Genomic regions displaying a positive CEV signal belong to the DSB-induced "D" compartment (black arrows). **b,** Genomic tracks of γH2AX ChIP-seq (+DSB) and first Chromosomal eigenvector (CEV) computed on differential (+DSB/−DSB) Hi-C matrix on chromosome X (top panel) or chromosome 6 (bottom panel). Three biological replicate experiments are shown as well as the CEV obtained upon DNA-PK inhibition. **c,** Boxplot representing the quantification of the «D compartment» signal on a 1 Mb window around best-induced DSBs, located on Chr. 1, 17, X (red, $n = 15$) and around control undamaged regions (random, purple, $n = 10$). *P* values were calculated using two-sided unpaired non-parametric Wilcoxon rank-sum tests. **d,** Pearson correlation between γH2AX ChIP-seq and D compartment (positive CEV PC1 on differential Hi-C matrices) signals for three biological Hi-C replicates and the Hi-C performed in presence of DNA-PK inhibition as indicated. **e,** Pearson correlation score was calculated on 100 kb bins between various histone modifications ChIP-seq data or RNA-seq obtained in DIvA cells and the D compartment (CEV) signal computed from +DSB/−DSB (top panel) or upon DNA-PK inhibition (+DSB + DNA−PKi/−DSB (bottom panel)) after exclusion of γH2AX-covered chromatin domains. Correlation analysis performed on all chromosomes showing D compartmentalization (*i.e*, Chr. 1,17 and X for top panel, and Chr. 1,2,6,9,13, 17,18,20 and X in presence of DNA-PKi, bottom panel). **f,** Enrichment for transcription factor motifs was analysed for genes that displayed a positive D compartment value, as compared with genes that displayed a negative

D compartment value. **g,** Boxplot showing the quantification of the D compartment signal computed from Hi-C data (+DSB + DNA-PKi/−DSB) on genes from chromosomes 1,2,6,9,13,17,18,20, X that are not upregulated (grey, $n = 3829$) or upregulated following DSB induction (red, $n = 86$) identified by RNA-seq. *P* values for each distribution were calculated using non-parametric Wilcoxon rank-sum tests tested against $\mu = 0$. *P* value between two distributions was calculated using two-sided unpaired non-parametric Wilcoxon rank-sum test. **h,** Transcripts from three genes were analysed using the above RNAscope probes: *PPIB* (non-D gene, not DSB-regulated), *CCL2* (non-D gene, up-regulated post DSB) and *GADD45A* (D-gene, up-regulated post DSB). Quantification of the RNAscope signals (as number of spots within nuclei) for probes targeting mRNA of *GADD45A* before and after DSB induction. **i,** Representative examples of γH2AX staining and RNAscope signals in DIvA cells after DSB induction are shown. **j,** Quantification of the colocalization between γH2AX foci and RNAscope foci following DSB induction (Mean and ±s.d. from biologically independent experiments ($n = 3$ for *PPIB* and *CCL2* and $n = 6$ for *GADD45A*)). *P* values were calculated using paired two-sided t-tests. **k,** Box plots showing the distance (Log10) between each gene up-regulated (log2 FC > 0.3) following DSB induction and comprising either in the D compartment or not and the closest DSB induced in DIvA cells. Top panel, D compartment genes identified from 3 independent Hi-C replicates (+DSB/-DSB) on Chr. 1, 17, X (upregulated in CompD, $n = 40$; upregulated not in CompD $n = 32$). Bottom panel, D compartment genes identified in Hi-C performed with DSB + DNA-PKi on chromosomes 1, 2, 6, 9, 13, 17, 18, 20 and X (upregulated in CompD, $n = 155$; upregulated not in CompD $n = 135$). *P* values were calculated using two-sided unpaired Wilcoxon rank-sum test.

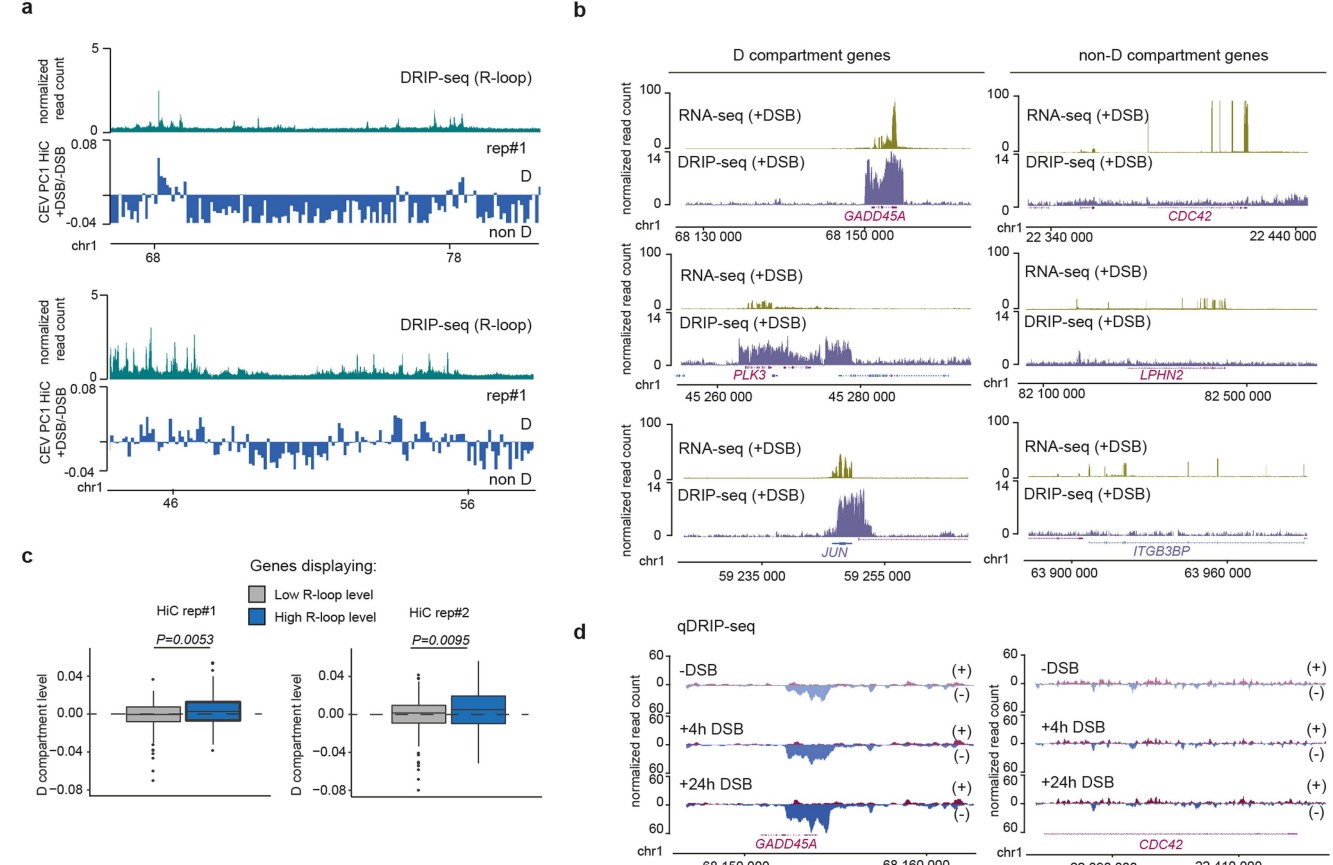

**Extended Data Fig. 6 | Related to Fig. 4: D compartment genes are enriched in R-loops. a,** Genomic tracks showing R-loops distribution analysed by DRIP-seq in DIvA cells after DSB induction (light blue), and the first Chromosomal Eigenvector computed on the differential Hi-C (+DSB/−DSB) (CEV, dark blue). A representative experiment is shown. **b,** Genomic tracks showing RNA-seq (green) and DRIP-seq (blue) read counts post DSB induction, at three genes found to be located in the D-compartment (left panels) or not (right panels). **c,** Box plot showing the quantification of the first chromosomal eigenvector (PC1) obtained from two Hi-C replicates (left and right panels) on genes that display a high R-loop level (top 10%, $n = 180$) or low R-loop level (bottom 10%, $n = 190$) as indicated. P values were calculated using two-sided unpaired non-parametric Wilcoxon rank-sum test. **d,** Genomic tracks showing qDRIP-seq data on the (+) strand (purple), and (−) strand (blue), before DSB induction, 4 h after DSB induction and 24 h after DSB induction (as indicated), on *GADD45A* (D compartment gene) and *CDC42* (non-D compartment gene).

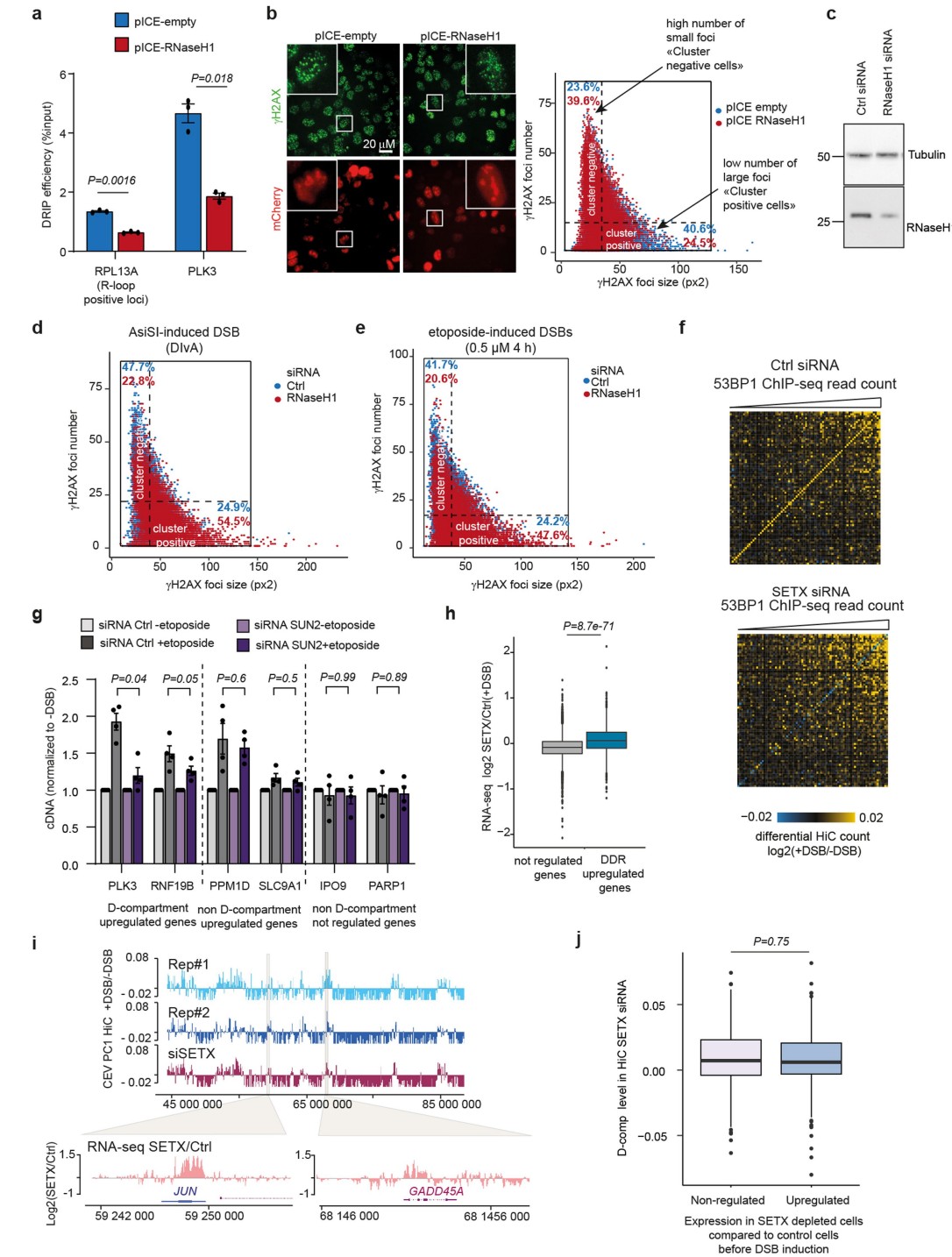

**Extended Data Fig. 7** | See next page for caption.

**Extended Data Fig. 7 | Related to Fig. 4: RNaseH1 and SETX regulate DSB clustering and D compartment genes. a,** DRIP-qPCR performed in DIvA cells overexpressing RNaseH1 (pICE-RNaseH1-NLS-mCherry, red) or not (pICE-NLS-mCherry, blue), on a known R-loop enriched locus (*RPL13*) and a D-compartment gene (*PLK3*). Mean and s.e.m. of *n* = 3 biologically independent experiments. *P* values were calculated using paired two-sided t-tests. **b,** Left panel: γH2AX staining and mcherry fluorescence signal after DSB induction in DIvA cells overexpressing RNaseH1 (pICE-RNaseH1-NLS-cherry) or not (pICE-NLS-mCherry). Scale bar 20 μm. Right panel: Scatter plot showing the number of γH2AX foci (y axis) as compared to their average size (x axis) in each DIvA cell overexpressing RNaseH1 (pICE-RNaseH1-NLS-mCherry) (*n* = 12446) or not (pICE-NLS-mCherry) (*n* = 13414) after DSB induction. The scatter plot is divided into four areas based on the median foci number and the median foci size (dotted black lines). Cluster-negative cells correspond to the upper left area of the scatter plot (cells with high number of γH2AX foci with small average size) while cluster-positive cells correspond to the lower right-side area of the scatter plot (cells with low number of γH2AX foci of large size). The percentage of cells in each category is indicated in the corresponding areas. A representative experiment is shown. **c,** Western blot of RNaseH1 and alpha-tubulin in DIvA cells transfected with control siRNA (Ctrl) and *RNaseH1* siRNA. (*n* = 2 biologically independent experiments). One representative experiment is shown. For gel source data, see Supplementary Fig. 1. **d,** As in **b** (right panel) but measured in DIvA cells after DSB induction and upon Ctrl (blue) or *RNaseH1* (red) depletion by siRNA. A representative experiment is shown. Ctrl siRNA, *n* = 13131 nuclei; *RNaseH1* siRNA, *n* = 12171 nuclei. **e,** As in **d** but measured in U2OS cells after DSB induction by Etoposide treatment. A representative experiment is shown. Ctrl siRNA, *n* = 10884 nuclei.; *RNaseH1* siRNA, *n* = 9039 nuclei. **f,** Differential Hi-C heatmaps (log2(+DSB/−DSB)) obtained in DIvA cells transfected with Ctrl (left panel) or *SETX* (right panel) siRNA, showing the contacts between each DSB (2 Mb domain resolution), sorted by their level of 53BP1. −log10(*p*) are indicated, with negative fold changes (damaged<undamaged) in blue, positive fold change (damaged>undamaged) in yellow. **g,** RT-qPCR quantification of the expression level of two up-regulated D compartment genes (*PLK3* and *RNF19B*), two upregulated non-D compartment genes *(PPM1D* and *SLC9A1*) and two unregulated and non-D compartment genes (*IPO9* and *PARP1*) before and after DSB induction by etoposide in U2OS cells transfected with Ctrl or *SUN2* siRNA. Gene expression is normalized to *RPLP0* gene expression. Mean ± s.e.m. of *n* = 4 biologically independent experiments are shown. *P* values were calculated using paired two-sided t-tests. **h,** Boxplots showing the log2 (*SETX*/Ctrl) RNA-seq read counts obtained following DSB upon Ctrl or *SETX* depletion by siRNA, computed on genes either upregulated (*n* = 991) or not (*n* = 8714) as indicated. *P* value was calculated using two-sided unpaired non-parametric Wilcoxon rank-sum test. **i,** Top panel: Genomic tracks showing the first Chromosomal eigenvector (CEV) computed on differential (+DSB/−DSB) Hi-C matrices. Two biological replicates are shown as well as the CEV obtained upon *SETX* depletion. Bottom panel: Two magnifications showing the log2 ratio of the RNA-seq obtained in *SETX* versus Ctrl siRNA transfected cells, after DSB induction. **j,** D-compartment signal obtained in *SETX*-depleted cells on genes either upregulated upon *SETX* depletion as compared to CTRL siRNA transfected undamaged cells on chromosome 1, 17, X (blue, *n* = 384), or not (white, *n* = 384, randomly sampled from *n* = 2624 total non-regulated genes). *P* value was calculated using unpaired two-sided non-parametric Wilcoxon rank-sum test.

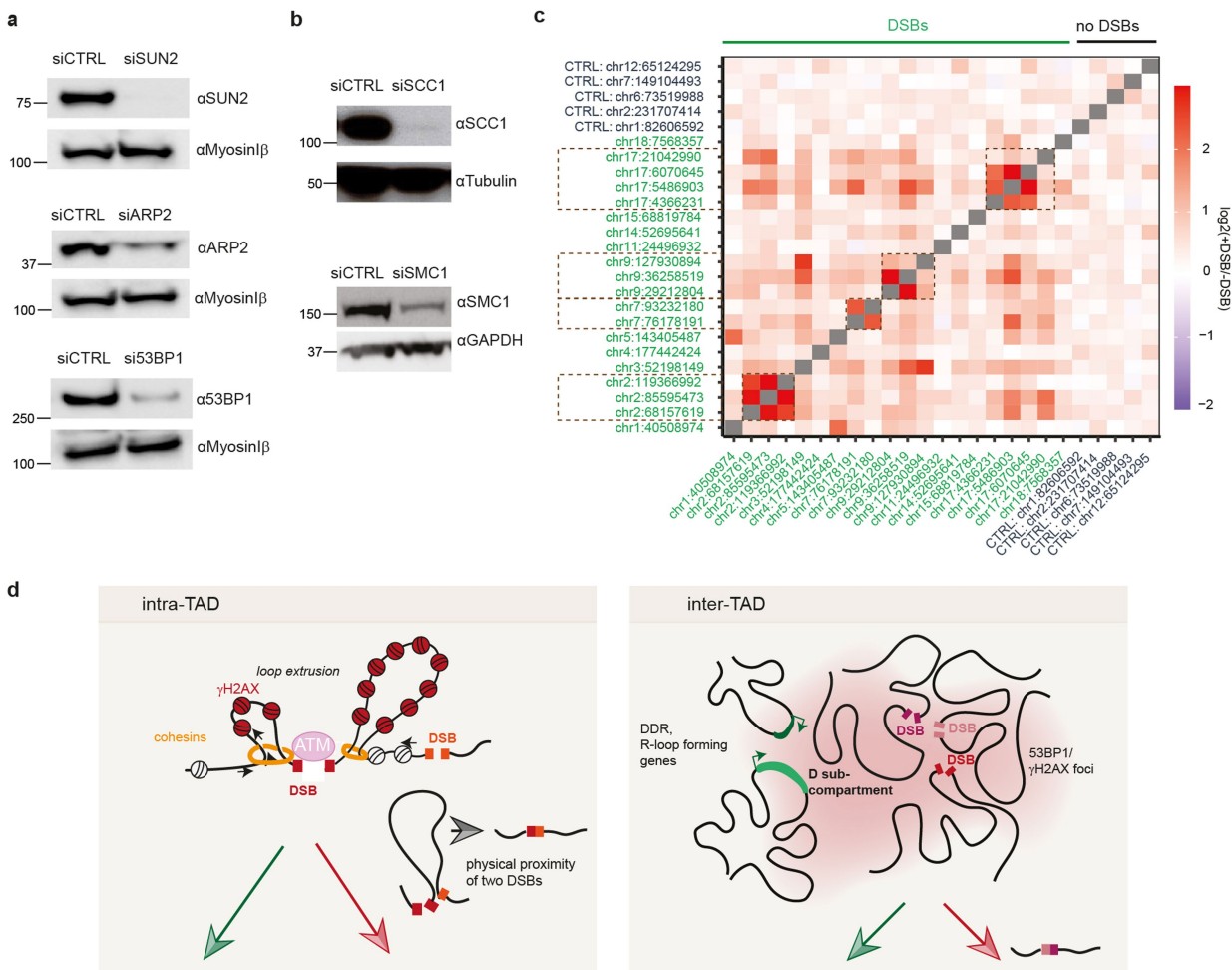

**Extended Data Fig. 8 | Related to Fig. 5: Translocations arise from loop extrusion and D compartment formation. a,** Western Blot showing the efficiency of *SUN2* (*n* = 5 biologically independent experiments), *ARP2* (*n* = 3) or *53BP1* (*n* = 2) depletion by siRNA. One representative experiment is shown. For gel source data, see Supplementary Fig. 1. **b,** Western Blot showing the efficiency of *SCC1* (*n* = 3 biologically independent experiments) or *SMC1* (*n* = 2) depletion by siRNA. One representative experiment is shown. For gel source data, see Supplementary Fig. 1. **c,** Heatmaps showing the rate of translocations sequenced between five control undamaged regions and twenty DSB sites in the AID-DIvA cell line (log2 (+DSB/−DSB)). *n* = 4 biologically independent experiments. **d,** DSBs elicit profound changes in chromosome architecture that display both beneficial and detrimental outcomes. Left panel: Cohesin-dependent loop extrusion arising at DSBs ensures ATM-dependent

phosphorylation of H2AX on an entire TAD, allowing fast establishment of a TAD-scale DDR focus. On the other hand, DSB-anchored loop extrusion also displays the potential to bring two DSBs which are located on the same chromosome into close proximity, which subsequently favour the occurrence of intra-chromosomal translocations. Right panel: Once assembled, the γH2AX/53BP1- decorated damaged TADs can further fuse together, *i.e.* self-segregate, through polymer-polymer phase separation. This creates a new nuclear, DSB-induced compartment, the D compartment, in which a subset of the DNA Damage Responsive genes that display R-loop accrual, physically relocate, in order to achieve optimal activation. On the other hand, spatial proximity induced within the D compartment also increases the frequency of translocations between DSBs and DDR responsive genes.

## Extended Data Table 1 | Primer sequences for amplicon-seq

| Name | Chr | coordinate of AsiSI site (hg19) | Primer coordinate (hg19) | sequence FW | sequence REV |
|------|-----|------|------|------|------|
| CTRL_CHR1 | chr1 | | 83072163 | GCACATGGGATTTTGCAGGAACATGTCTCTGTCACTGG | CCATCTGGCTTTAGTTCCACTTCAATCAGC |
| CTRL_CHR12 | chr12 | | 65517948 | CTAGCATATGATAAAAATTTGGAAATAATGTGCTAGGC | GGCTATGATCATGTTTTATTTGTACTATAATTTCCAGGTC |
| CTRL_EEF1A1 | chr6 | | 74229585 | CCTTTCTGGTATTAAACATACCTCAGCAGCC | GATCTTGGTTCATTCTCAAGCCTCAGACAGTGG |
| CTRL_PTMAp | chr2 | | 232572004 | CGACAGCTAAGTGCGGTGGCGATAAGGCCAGCG | CAGAGTTCCTCGGGAAAAATCATCGTCTCTGGAGCC |
| CTRL_ZNF425 | chr7 | | 148801459 | CGCCGCAGGGAAACTCCTTCTGCCTGTCGTGGAC | CGAGTGTAACAAAAGTTTCCGCCTCAAGAGAAGCCTG |
| SITE516 | chr1 | 40974643 | 40974513 | GCGTCCGGGAGCAGCTCCGAGGCCGCGGCG | GAAGGAGTCCCGGCCAACAGCACTGTGTTCCGA |
| SITE709 | chr2 | 68384748 | 68384627 | CGGCAACCTGGAAACCAGACTCCAAACATTGGC | GGTCTTGGGCTCACTCCAGCCGCGCAACTCCCAG |
| SITE716 | chr2 | 85822593 | 85822445 | TCCGGAAGCGCCATGGGCCAACGCTCGCA | CCCTCTCATTGGCTTGAACAAGTCAGGTGACC |
| SITE722 | chr2 | 120124565 | 120124434 | GCTAGCGCCGCGGCGGGGGCTGGGCACGC | GAGCCCCAACGGGCCTCCCGCCCGCTGCACC |
| SITE765 | chr3 | 52232162 | 52232026 | TCCGCAGAGCCCAAGAAGTGGGCCTCTCTGC | GCCGCACCTACCGCGCGGGCCCTCACCTGCGCTG |
| SITE871 | chr4 | 178363575 | 178363436 | TGACGACCAGGGGCAGAGGGCTGGAGCAGC | CGTGACACAACTCTCCCGCGGGCCCAGGGACG |
| SITE919 | chr5 | 142785049 | 142784920 | CGGCGGCTCCCCCTGCTCTGACATCTTGAAGACG | CCTCTGGAGGCGGCCCCCGTAGATCGTCTCCGG |
| SITE1025 | chr7 | 75807506 | 75807383 | GCGCCCAGGGCGAAGGTACTTAGACAGC | AGCCTCAGCCTCTGGCTCCCTCTGCAGGCAG |
| SITE1031 | chr7 | 92861490 | 92861341 | TCACACTCCTAGGGGACATCGCGTCAATCTGGC | GCTGGGAACTGTAGTCTTCTGGACGCCGGCGGTAGCAACG |
| SITE1123 | chr9 | 29212799 | 29212659 | TCAACCTCCACAGAGAACCCAGTCCGAAATCG | GCCTCAAGGTAAGAGGCAGCAAAGCTGCTGCTTG |
| SITE1129 | chr9 | 36258513 | 36258383 | GCCACGAAGCAGGCAGAGCGCGAGC | CATGGAAGATGGTCCGCTGGTCAGC |
| SITE1159 | chr9 | 130693170 | 130693031 | TCGCCCCCGCAGCAGCTTCCCGGGCTTTGGCCGC | CAGGCGGCCCTAGCGACCCGAGTCCCCACGCCG |
| SITE72 | chr11 | 24518475 | 24518337 | CAATTATCGGCAGATTAGGTTTCTACTCTCCGTG | CAGGAGTGATGTCACCGGCGAGGCGAGCCCTGCC |
| SITE219 | chr14 | 54955825 | 53162224 | CACGCTTGGGAGGCCAGTCCGCACGCTCGGTCG | GCGGGGCCGAGCCGGGAAGCTGGTCGGTGCG |
| SITE270 | chr15 | 69112120 | 69111977 | CTGCCGACAAGCCCAGCACTGAGAAAGACGGGC | TCCACCCCACCTTCGGGCCAGATGCTGATGTTTCT |
| SITE341 | chr17 | 4269523 | 4269398 | CGGGCGAGAACGGCTGGGCCCGGCCGGGACCG | AGCCCCGGCGCCAGCCCGGGCCCTCGGC |
| SITE343 | chr17 | 5390220 | 5390088 | CGGCCTGTGGGTCGGCCTCACCCCGGCCTCCG | CCGCCTCATCTTCTCTGCTAGACTTAGAGTTCCTG |
| SITE344 | chr17 | 5973962 | 5973831 | GTCCGCTTGCCGCGGGCGGCCGGAGACGTGC | CGCCGCGCTCCCAACCGCGTCTGCAC |
| SITE360 | chr17 | 20946300 | 20946179 | CGGCGGCGAGGGAGGCGAGGACGACGCCAG | GTGACGTAATCTCCGTCCGCGGCCGCGGCG |
| SITE396 | chr18 | 7566712 | 7568200 | GAGTCCCTGGCCGGCTGCAAAAGGAACAGCAG | AGCCTCTCCGCAGACGCTGCACGACCGCG |

## Extended Data Table 2 | Primer sequences for 4C-seq

| Name | Forward primer | Reverse primer |
|---|---|---|
| Viewpoint DSB1 (chr1, cluster -prone) | AATGATACGGCGACCACCGAGATCTACACTCTTTC CCTACACGACGCTCTTCCGATCTAACCTGGCAAC TTATGAATCAGGA | CAAGCAGAAGACGGCATACGAGAT NNNNNN GT GACTGGAGTTCAGACGTGTGCTCTTCCGATCT ATGTCAAAAGCCAAGGGGACA |
| Viewpoint DSB2 (chr17, cluster -prone) | AATGATACGGCGACCACCGAGATCTACACTCTTTC CCTACACGACGCTCTTCCGATCTTCCTTACGATTA TTTGTGAATTTTG | CAAGCAGAAGACGGCATACGAGAT NNNNNN GT GACTGGAGTTCAGACGTGTGCTCTTCCGATCT AAGCTAATTCTGAGTTACATACATT |
| Viewpoint DSB3 (chr21, not cluster -prone) | AATGATACGGCGACCACCGAGATCTACACTCTTTC CCTACACGACGCTCTTCCGATCTGATTACGTAGAA GGGTGCC | CAAGCAGAAGACGGCATACGAGAT NNNNNN GT GACTGGAGTTCAGACGTGTGCTCTTCCGATCT AAGGCAAATGATAACCCTGT |
| Viewpoint DSB4 (chr20, cluster -prone) | AATGATACGGCGACCACCGAGATCTACACTCTTTC CCTACACGACGCTCTTCCGATCTGGTTATACTAAG ATGTCAGTTCCT | CAAGCAGAAGACGGCATACGAGAT NNNNNN GT GACTGGAGTTCAGACGTGTGCTCTTCCGATCT CACGCACCTGGTTTAGATT |
| Viewpoint ctrl region (chr17) | AATGATACGGCGACCACCGAGATCTACACTCTTTC CCTACACGACGCTCTTCCGATCTTCCTCAGGTTAT CATCCCAA | CAAGCAGAAGACGGCATACGAGAT NNNNNN GT GACTGGAGTTCAGACGTGTGCTCTTCCGATCT ACCTTTACACCTCAAAACCT |

NNNNNN is the position of the optional index.

# Reporting Summary

## Statistics

For all statistical analyses, confirm that the following items are present in the figure legend, table legend, main text, or Methods section.

| n/a | Confirmed | |
|---|---|---|
| ☐ | ☒ | The exact sample size (*n*) for each experimental group/condition, given as a discrete number and unit of measurement |
| ☐ | ☒ | A statement on whether measurements were taken from distinct samples or whether the same sample was measured repeatedly |
| ☐ | ☒ | The statistical test(s) used AND whether they are one- or two-sided<br>*Only common tests should be described solely by name; describe more complex techniques in the Methods section.* |
| ☒ | ☐ | A description of all covariates tested |
| ☒ | ☐ | A description of any assumptions or corrections, such as tests of normality and adjustment for multiple comparisons |
| ☐ | ☒ | A full description of the statistical parameters including central tendency (e.g. means) or other basic estimates (e.g. regression coefficient) AND variation (e.g. standard deviation) or associated estimates of uncertainty (e.g. confidence intervals) |
| ☐ | ☒ | For null hypothesis testing, the test statistic (e.g. *F*, *t*, *r*) with confidence intervals, effect sizes, degrees of freedom and *P* value noted<br>*Give P values as exact values whenever suitable.* |
| ☒ | ☐ | For Bayesian analysis, information on the choice of priors and Markov chain Monte Carlo settings |
| ☒ | ☐ | For hierarchical and complex designs, identification of the appropriate level for tests and full reporting of outcomes |
| ☐ | ☒ | Estimates of effect sizes (e.g. Cohen's *d*, Pearson's *r*), indicating how they were calculated |

*Our web collection on statistics for biologists contains articles on many of the points above.*

## Software and code

Policy information about availability of computer code

| Data collection | Bio-Rad CFX Manager version 3.1<br>ChemiDoc™ Touch Imaging System.<br>Image Lab Touch version 1.2.0.12<br>MetaMorph version 7.1.0.0<br>Micro-Manager 2.0<br>Colombus software version 2.8.2<br>Harmony software version 4.9 |
|---|---|
| Data analysis | Bio-Rad CFX Manager version 3.1<br>Article Github (https://github.com/LegubeDNAREPAIR/ATMcompD)<br>A custom R script (https://github.com/bbcf/bbcfutils/blob/master/R/smoothData.R) was used to build the coverage file in bedGraph format<br>Juicer Tools HiCCUPS program (https://github.com/aidenlab/juicer/wiki/HiCCUPS)<br>Juicer Tools APA program (https://github.com/aidenlab/juicer/wiki/APA)<br>mProfile (https://github.com/aldob/mProfile)<br>A specific python script from FourCSeq R package (Klein, F. A. et al. FourCSeq: analysis of 4C sequencing data. Bioinformatics 31, 3085–3091 (2015)) v1.2.0<br>edgeR package 3.26.0<br>Image Lab Touch software 5.0<br>ImageJ (Fiji) v1.53c<br>Icy 2.4.1.0<br>Inscoper<br>HiTC 1.30.0<br>JASPAR v2020<br>motifmatchR 1.6.0 |

igraph R package 1.2.4.1
regioneR R package 1.15.2
GenomicRanges 1.38
plyranges 1.6.10
tidyverse 1.3.0 (including ggplot2 R package 3.3.3)
patchwork 1.1.1
ggforce 0.3.3
ggside 0.2.0
ggtext 0.1.1
Integrated Genome Browser version 9.1.6
Juicer version 1.6
Juicebox version 1.11.08
macs2 2.1.2
macs3 3.0.0a7
bedtools v2.26.0
bwa 0.7.12-r1039
samtools 1.9
deeptools 3.4.3
R 3.6.3
gkmsvm 0.81.0
CellProfiler 4.2.1
Colombus software version 2.8.2

For manuscripts utilizing custom algorithms or software that are central to the research but not yet described in published literature, software must be made available to editors and reviewers. We strongly encourage code deposition in a community repository (e.g. GitHub). See the Nature Portfolio guidelines for submitting code & software for further information.

## Data

Policy information about availability of data

All manuscripts must include a data availability statement. This statement should provide the following information, where applicable:

- Accession codes, unique identifiers, or web links for publicly available datasets
- A description of any restrictions on data availability
- For clinical datasets or third party data, please ensure that the statement adheres to our policy

New high throughput sequencing data have been deposited publicly to Array Express under the accession number: E-MTAB-10865.

Other high-throughput sequencing data used in this study are available under accession numbers: E-MTAB-8851 (Hi-C data before and after DSB induction and upon Ctrl or SCC1 depletion; pATM ChIP-seq), E-MTAB-5817 (BLESS before and after DSB induction; LIG4, 53BP1, gH2AX, FK2 and histone H1 ChIP-seq experiments) and E-MTAB-6318 (DRIP-seq before and after DSB induction; SETX ChIP-seq). Breakpoint positions of inter-chromosomal translocations across 18 different cancer types were retrieved from Zhang et al., Cell Rep, 2018 (doi: 10.1016/j.celrep.2018.06.025).

# Field-specific reporting

Please select the one below that is the best fit for your research. If you are not sure, read the appropriate sections before making your selection.

☒ Life sciences   ☐ Behavioural & social sciences   ☐ Ecological, evolutionary & environmental sciences

For a reference copy of the document with all sections, see nature.com/documents/nr-reporting-summary-flat.pdf

# Life sciences study design

All studies must disclose on these points even when the disclosure is negative.

| | |
|---|---|
| Sample size | No sample size calculation was performed. The number of samples in each experiment was determined based on standards practice in the field (N=3). This is why we did N= or > 3 independent experiments unless stated otherwise, for example some sequencing experiments, which based on our experience require less replicates (doi: 10.1038/s41586-021-03193-z ; 10.1016/j.molcel.2018.08.020 ; 10.1038/s41467-018-02894-w etc.). The number of independent experiments are indicated in the legend of each Figure. |
| Data exclusions | No data were excluded from analysis. |
| Replication | RNA-seq before and after DSB induction: 2 replicates. Replication was successful
Hi-C before and after DSB induction: 3 replicates. Replication was successful
Hi-C after DSB induction upon ATM or DNA-PK inhibition : n=1. Data were confirmed by Immunofluorescence experiments (i.e. DSB clustering)
Hi-C after DSB induction in CTRL or SETX depleted cells : n=1. Data were confirmed by Immunofluorescence experiments (i.e. DSB clustering)
4C-seq before and after DSB induction in the G1, G2 or S phase of the cell cycle: n=1, 4 DSB viewpoints. The same results were observed from the different DSB viewpoints.
Amplicon-seq upon Ctrl, SCC1, SUN2 or Arp2 depletion : 4 replicates. Replication was successful
Translocation assays (qPCR) : n=3 or 4 depending on the conditions. Replication was successful
RT-qPCR after siRNA treatments /etoposide: n=4. Replication was successful
RNAscope: n=3 for CCL2 and PPIB probes, n=6 for GADD45A probe. Replication was successful
qDRIP-seq: n=1. |

RNAseq siRNA SETX n=2. Replication was successful
High-throughput microscopy n=3. Replication was successful
DRIP-seq siSETX n=1
Half-FRAP n=1 experiment with n=22 foci analyzed. Replication was successful

| | |
|---|---|
| Randomization | Randomization is not relevant because we did not use different experimental groups in our study. |
| Blinding | Blinding was not relevant to our study since we did not have experimental group to compare. |

# Reporting for specific materials, systems and methods

We require information from authors about some types of materials, experimental systems and methods used in many studies. Here, indicate whether each material, system or method listed is relevant to your study. If you are not sure if a list item applies to your research, read the appropriate section before selecting a response.

## Materials & experimental systems

| n/a | Involved in the study |
|---|---|
| ☐ | ☒ Antibodies |
| ☐ | ☒ Eukaryotic cell lines |
| ☒ | ☐ Palaeontology and archaeology |
| ☒ | ☐ Animals and other organisms |
| ☒ | ☐ Human research participants |
| ☒ | ☐ Clinical data |
| ☒ | ☐ Dual use research of concern |

## Methods

| n/a | Involved in the study |
|---|---|
| ☒ | ☐ ChIP-seq |
| ☒ | ☐ Flow cytometry |
| ☒ | ☐ MRI-based neuroimaging |

## Antibodies

| | |
|---|---|
| Antibodies used | See methods section of the manuscript for use and dilution of antibodies

Primary antibodies:
SUN2 (Abcam ab124916, clone EPR6557)
ARP2 (Abcam ab128934, clone EPR7980)
53BP1 (Novus Biologicals NB100-305, polyclonal)
SCC1 (Abcam ab992, polyclonal)
SMC1 (Abcam ab75819, clone EP2879Y)
Myosin (Sigma M3567, polyclonal)
GAPDH (Sigma MAB374, clone 6C5)
DNA-RNA Hybrid (clone S9.6) (Antibodies Incorporation)
RnaseH1 (Invitrogen PA5-30974, polyclonal)
gH2AX (Millipore Sigma 05-636, clone JBW301)

Secondary antibodies:
Goat Anti-Mouse IgG-Peroxydase (Sigma A2554, polyclonal)
Goat Anti-Rabbit IgG-Peroxydase (Sigma A0545, polyclonal)
Goat anti-Mouse IgG (H+L) Alexa Fluor™ 488 (Invitrogen # A-11029, polyclonal)
Goat anti-Mouse IgG (H+L) Alexa Fluor™ 594 (Invitrogen # A-11032, polyclonal)
Goat anti-Rabbit IgG (H+L) Alexa Fluor™ 488 (Invitrogen # A-11034, polyclonal)
Goat anti-Rabbit IgG (H+L) Alexa Fluor™ 594 (Invitrogen # A-11037, polyclonal) |
| Validation | Primary antibodies:
SUN2 (Abcam ab124916) previously validated in human for WB as stated by the manufacturers on their website (https://www.abcam.com/sun2-antibody-epr6557-ab124916.html)
ARP2 (Abcam ab128934) previously validated in human for WB as stated by the manufacturers on their website (https://www.abcam.com/arp2-antibody-epr7980-ab128934.html)
53BP1 (Novus Biologicals NB100-305) previously validated in human for WB as stated by the manufacturers on their website (https://www.novusbio.com/products/53bp1-antibody_nb100-305)
SCC1 (Abcam ab992) previously validated in human for WB as stated by the manufacturers on their website (https://www.abcam.com/rad21-antibody-ab992.html)
SMC1 (Abcam ab75819) previously validated in human for WB as stated by the manufacturers on their website (https://www.abcam.com/smc1a-antibody-ep2879y-ab75819.html#lb)
Myosin (Sigma M3567) previously validated in human for WB as stated by the manufacturers on their website (https://www.sigmaaldrich.com/catalog/product/sigma/m3567?lang=fr®ion=FR)
GAPDH (Sigma MAB374) previously validated in human for WB as stated by the manufacturers on their website (https://www.sigmaaldrich.com/US/en/product/mm/mab374?gclid=CjwKCAjwxo6IBhBKEiwAXSYBs2mXMTrpnvmfDppcnfh3YEPY6Rdo9QRlHfhswQNLHvcDDPDRyV_wxxoCiykQAvD_BwE)
DNA-RNA Hybrid antibody S9.6 antibodies previously validated in human for DRIP experiment (Crossely et al, 2020, NAR, doi: 10.1093/nar/gkaa500)
gH2AX (Millipore Sigma 05-636) previously validated in IF as stated by the manufacturers on their website (https://www.emdmillipore.com/US/en/product/Anti-phospho-Histone-H2A.X-Ser139-Antibody-clone-JBW301,MM_NF-05-636-I?) |

ReferrerURL=https%3A%2F%2Fwww.google.com%2F)

Secondary antibodies:
Goat Anti-Mouse IgG-Peroxydase (Sigma A2554) previously validated for WB as stated by the manufacturers on their website (https://www.sigmaaldrich.com/US/en/product/sigma/a2554)
Goat Anti-Rabbit IgG-Peroxydase (Sigma A0545) previously validated for WB as stated by the manufacturers on their website (https://www.sigmaaldrich.com/US/en/product/sigma/a0545)
Goat anti-Mouse IgG (H+L) Alexa Fluor™ 488 (Invitrogen # A-11029) previously validated for IF as stated by the manufacturers on their website (https://www.thermofisher.com/antibody/product/Goat-anti-Mouse-IgG-H-L-Highly-Cross-Adsorbed-Secondary-Antibody-Polyclonal/A-11029)
Goat anti-Mouse IgG (H+L) Alexa Fluor™ 594 (Invitrogen # A-11032) previously validated in IF as stated by the manufacturers on their website (https://www.thermofisher.com/antibody/product/Goat-anti-Mouse-IgG-H-L-Highly-Cross-Adsorbed-Secondary-Antibody-Polyclonal/A-11032)
Goat anti-Rabbit IgG (H+L) Alexa Fluor™ 488 (Invitrogen # A-11034) previously validated in IF as stated by the manufacturers on their website (https://www.thermofisher.com/antibody/product/Goat-anti-Rabbit-IgG-H-L-Highly-Cross-Adsorbed-Secondary-Antibody-Polyclonal/A-11034)
Goat anti-Rabbit IgG (H+L) Alexa Fluor™ 594 (Invitrogen # A-11037) previously validated in IF as stated by the manufacturers on their website (https://www.thermofisher.com/antibody/product/Goat-anti-Rabbit-IgG-H-L-Highly-Cross-Adsorbed-Secondary-Antibody-Polyclonal/A-11037)

# Eukaryotic cell lines

Policy information about cell lines

| Cell line source(s) | Cell lines developped from U2OS cells (ATCC® HTB-96™) in the Gaelle Legube's laboratory (DIvA cell line, 53BP1-GFP-DIvA, and AID-DIvA cell line) |
|---|---|
| Authentication | Authentication of the U2OS cell line was performed by the provider ATCC which uses morphology and Short Tandem Repeat profiling to confirm the identity of human cell lines. DIvA, AID-DIvA and 53BP1-GFP-DIvA cells derived from these U2OS cells were not further authenticated. |
| Mycoplasma contamination | All cell lines (DIvA, 53BP1-GFP-DIvA and AID-DIvA) were regularly tested for absence of mycoplasma contamination by using the MycoAlert Mycoplasma (Lonza). All cell lines used in this study were tested negative for Mycoplasma. |
| Commonly misidentified lines<br>(See ICLAC register) | No commonly misidentified cell lines were used in the study. U2OS are not registered in ICLAC. |

