## [Peer Review File · Nature]

Manuscript Title: Chromatin compartmentalization regulates the response to DNA damage

Reviewer Comments & Author Rebuttals

Reviewer Reports on the Initial Version:

Referees' comments:

Referee #1:

This manuscript by Legube and colleagues addresses the role of double-strand break (DSB) clustering. Recently published work from this group has shown that loop extrusion, which is important for TAD formation, is important for H2AX spreading and DNA damage response (DDR) foci formation. It has also been shown that ionizing radiation (IR) enforces TAD formation, and that DSB clustering is dependent on the LINC complex and the phase separation properties of 53BP1. However, the function of clustering is unclear. Using their DiVA system, Legube and colleagues examined genome organization and TAD contacts around induced DSBs using Hi-C, showing essentially as they did in Arnould et al. that DSB formation increases intra-TAD contacts and triggers SCC-dependent and ATM-dependent loop extrusion. They also now show that DNA-PK inhibition enhances the process. Their studies show that multiple DSBs can cluster to form TAD cliques and that those prone to this process are in transcriptionally-active regions and have higher levels of H2AX and 53BP1. ATM is also needed for clustering (inhibition blocks clustering), while DNA-PK inhibition, by contrast, promotes clustering. Furthermore, clustering is more common/prominent in G1 cells than S or G2 cells.

The authors also show that damaged TADs, marked by H2AX and 53BP1, form a new chromatin compartment, akin to A and B compartments, that segregates from the rest of the genome and that includes genomic loci that do NOT contain a DSB, but which are areas of active transcription. They suggest that the regions of active transcription that do cluster are enriched in genes upregulated following DNA damage. They also suggest that these genes are enriched in the formation of R-loops, and the knockdown of SETX, known to increase R-loop formation, leads to increased clustering and increased expression of damage response genes. Disruption of clustering leads to less expression. Lastly, the authors show that clustering, while beneficial for the expression of certain DDR genes, can lead to increased translocations of the genes targeted to these sites and that translocations observed in cancer genomes are enriched at these genes.

Overall, this story presents several novel findings concerning the role of DSB clustering and chromatin organization and TAD structure in the DNA damage response. Although the story builds on previously published work, it extends it in new directions particularly by showing the link of clustering to gene expression. This is a relatively unexplored area, and the authors' findings open new avenues of exploration that will be of high general interest. That said, some of the conclusions are a little premature and are lacking mechanistic insight. Although ATM is proposed to regulate this pathway, how it does so is unclear, as is the role of DNA-PK in this process. The role of R-loops in the process is also somewhat tenuous, and more should be done to clarify the meaning of these observations and support the new, novel conclusions. Both are a little loosely tied to the main model at this time yet are some of the most novel data. Specific comments are below:

1) What is the relationship between ATM and DNA-PK in this work? Are the activities of these two kinases countering each other? If both drugs are added at the same time, is one effect dominant or lost? More information about the function of these kinases would be helpful, particularly as the impact of DNA-PK is somewhat surprising. Is DNA-PK's impact on clustering, etc due to a defect in DSB repair? Does loss of a downstream protein in the NHEJ pathway have the same effect as DNA-PK inhibition? Or is this some type of signaling effect?

2) In analyzing R-loops (Fig. 3e) are the authors using data from undamaged cells or damaged cells? Are R-loops increased on the genes targeted to the DDR clusters after damage? The link between R-loops and the targeting of R-loop containing genes to clusters needs further development.

3) Along these lines, the authors suggest that R-loops are important for clustering and in support of this idea, they show that perturbation of SETX, which has been reported to affect R-loops levels, leads to more clustering and gene expression. But is this really due to SETX's effects on R-loops or its effects on something else (e.g. transcription or repair)? The authors should test the impact of RNase H expression on R-loop levels, D-compartment formation/clustering, and the transcription of DDR genes. In addition, could the effects of SETX on repair be affecting the underlying cause for these effects? We know SETX loss leads to more translocations and defects in repair from the authors' previous work.

4) All of the experiments are done in the same system, with a large number of DSBs induced. If the authors induce a smaller number of DSBs using a CRISPR system, would they see the same effects? Is DSB clustering of these breaks observed, and are all of the upregulated damage response genes targeted to these clusters as well? I am wondering if somehow their observation regarding the recruitment of certain genes to the clusters is tied to the location of the DSBs induced here and the large number of DSBs induced.

5) Around line 313, the authors link R-loop accrual to non-coding RNAs, 53BP1 foci formation, etc. It would help if they elaborated on this point more as these are not really R-loops. The logic behind these statements is not entirely clear and seems to be a bit of a stretch.

6) Do DNA-PK inhibition and SETX knockdown have additive effects on these processes? Are they somehow working together? The role of DNA-PK needs further exploration.

7) Minor point: The authors note (p, 11, end of the section) that there is a role for the D compartment in activation of the DNA damage response. Saying this is the DNA damage response is a rather broad statement and should be modified to be more specific and targeted to the transcriptional activation of damage-responsive genes.

Referee #2:

Arnould et al. use the DIvA cell line for the controlled formation of DSBs at defined sites across the genome to follow and characterize 3D chromosome dynamics by using chromosome conformation capture and ChIP-seq approaches. They report that upon DSB formation, contact frequencies were increased within TADs with DSBs, while contact frequencies between neighboring TADs were decreased. In accordance with previous studies showing reinforcement of TADs upon irradiation in mammalian cells (Sanders et al., *Nat. Commun.*, 2020), the authors found that those changes in contact frequencies were also dependent on the kinase activity of ATM. This observation is in line with the requirement of ATM kinase activity for the DSB-anchored loop extrusion, and it is in stark contrast with what happens under DNA-PK inhibition, which exacerbates the increase in intra-TAD contacts, increases loop extrusion and the frequency of formed translocations.

Moreover, following their previous findings that DSBs cluster in an ATM-dependent manner in G1 phase of the cell cycle, when induced within active genes (Caron et al., *Cell Rep.*, 2015 and Aymard et al., *NSMB*, 2017) the authors show by Hi-C how clustering occurs between breaks within the same and different chromosomes. They report that clustering correlates with DSB-induced chromatin features that occur at the scale of an entire TAD; however, an important control is missing here that could explain these findings (see below).

Applying principal component analysis in Hi-C data upon DSB formation the authors report the formation of a new "D" compartment, which comprises both damaged and undamaged chromatin domains and is enriched in active chromatin marks and R-loops. Intriguingly, they also provide evidence that the non-damaged active sites that are recruited to D compartments are genes with dedicated roles in the DNA damage response, and their recruitment is required for optimal activation and coincide with breakpoint regions found rearranged in cancer.

Not surprisingly, clustering of DSBs promotes the formation of fusions, whose formation, as the authors showed, was dependent on 53BP1, phase separation (disrupted by treatment with 1,6-hexanediol), the LINC complex, ARP2, an actin-branching factor, and cohesin (SCC1 depletion, although doesn't display clustering defects).

Overall, this is a high quality study that builds upon already published findings to further describe principles of chromosome dynamics upon DSB formation. They use innovative methodologies to obtain data of high quality, which are presented with clarity. In few occasions, the conclusions drawn by the authors were not fully supported by the obtained data, and further experiments are required to support their claims (see below). Having said that, overall the novel aspects presented by this manuscript are preliminary and lack mechanistic insights for publication in Nature.

Addressing the criticism raised below will substantially improve the manuscript:

- The authors report that clustering of multiple TADs upon DSB induction correlated with several DSB-induced chromatin features that occur at the scale of an entire TAD, including γ H2AX, 53BP1, and ubiquitin chain levels as well as the depletion of histone H1 around DSB detected by ChIP-seq. I am wondering whether this is just a reflection of the cutting efficiency of the DSBs in the population. In other words is it possible that DSBs with higher cutting efficiency in the population show higher changes in DDR events (recruitment of DDR factors, histone eviction, etc) and cluster the most? Correlating clustering with cutting efficiency determined by BLESS will directly answer this.
- It is unclear to me whether the experimental description of the D compartment is correct and truly corresponds to any physiologically relevant events that are observed in vivo. The main concern is that the PCA analysis on the Hi-C map of the +DSB condition alone should have revealed the reported "new" compartment, but as the authors clearly showed that was not the case. The PCA analysis of differential Hi-C maps on the ratio of the contact matrices (+DSB/-DSB condition) is therefore not meaningful and may exacerbate differences due to its nature.
- Can the authors confirm their most important findings when inducing DSBs by other means (such as CRISPR, IR, etc)?
- I am wondering what fraction of the genes with changes in gene expression upon DSB formation is that was targeted in the D compartment? The authors show that genes that upregulated following DSB induction displayed a higher D compartment signal compared to genes that were not upregulated after DSBs (Fig. 3d, Fig. S4g). Can the authors show how many genes were differentially expressed upon DSB induction (upregulated, downregulated, show no changes) and what was the fraction of those that were targeted in the D compartment?
- The authors report that genes identified in the D compartment displayed increased R-loop levels compared to non-D genes (Fig. 3f) and vice-versa, genes exhibiting high R-loop levels showed higher D-sub compartment signal (PC1 value) when compared to genes with low level of R-loops (Fig. 3g). Do the authors think that this is a direct effect of R-loops per se, or could it be a secondary effect of transcription levels that could also affect the levels of R-loops? In the same direction, SETX depletion could also affect transcription levels (not only the levels of R-loops) and thus increased transcription could promote clustering. Can the authors identify what is causal here? For example the authors could use alternative means to increase R-loops and try to rescue

clustering by overexpression of factors that resolve R-loops (such as RNase H1).

- How does ATM inhibition influence translocation frequency?

Referee #3:

In this manuscript the authors report that upon induction of DSBs, damaged TADs cluster together in an ATM-dependent manner forming a DSB-induced compartment ("D"-compartment). They report that these changes in genomic architecture caused by DSB induction targets DDR genes to D compartments, which augments their transcription. They also propose that DSB clustering could promote translocations and genome instability since many DDR genes are tumor suppressor genes. Overall, the data is of very high quality, presented in a logical flow, and potentially represents important new insights that would be of interest to both specialists and a general readership. I believe this manuscript could be considerably strengthened with additional validation (see below).

Major comments:

1. D-compartment formation shown by fluorescence in-situ hybridization before and after DSBs would significantly strengthen this manuscript. For example, in agreement with Hi-C and 4C, immune-FISH should show that "cluster-prone" DSBs cluster together within γ -H2AX foci, as opposed to control DSB loci, and the same loci in cells without DSBs.
2. On p. 5 (line 97) the authors mention that some γ -H2AX domains are able to interact with more than one other γ -H2AX domain. They imply that their data indicates that multiple damaged TADs are capable of simultaneous clustering. However it cannot be assumed to be the case since this is population Hi-C data. This would be an ideal set of TADS for a multi-label DNA FISH experiment to show simultaneous associations of multiple TADs in a single D-compartment following DSB.
3. The authors suggest that increased clustering of DDR genes in D-compartments plays a role in their transcriptional upregulation. RNA FISH with an immuno-marker of D-compartments would greatly strengthen/validate this conclusion and rule out some downstream consequence or diffusible signal from D-compartment formation or DSBs, such as phosphorylation of a transcription factor (for example) in response to DSBs or D-compartment formation.

Minor comments:

4. It is not clear what the distribution of DSBs is across all chromosomes. The data indicates that 80 DSBs were induced by AsiSI in this study. What is the distribution of these breaks across all chromosomes? Was this similar to the DSB distribution obtained in Clouaire et al. (ref. 3 in manuscript)? A table such as "Table S1" included in Clouaire et al. (2018) will make this information more transparent to the reader.
5. The methods section indicates that the authors "were able to extract the D-compartment on chromosomes 1, 17 and X". Does this mean that D-compartment was not found on other chromosomes?
6. It is suggested that the DSB clustering "entirely dependent on ATM, exacerbated upon DNA-PK inhibition..." More validation/data presentation is required to make this more obvious to the reader. Inhibition of DNA repair (ATM and DNA-PK inhibition) may be causing these cells to undergo apoptosis/pyknosis. Do genome-wide Hi-C heat maps show other normal higher order features upon DNA-PK and ATM inhibition, for example TAD structure and A/B compartments? This is not possible to decipher from the plots included in Fig. 2.

7. Fig. 2e shows some residual contact between DSBs in the ATMi panel. Although this would suggest that DSB clustering is somewhat compromised in the absence of ATM, it is not sufficient to suggest complete dependence of DSB clustering on ATM, unless this phenomenon can be shown in ATMi-depleted cells by microscopy. Complete dependence could be indicated if DSB loci in ATMi cells would show no evidence of clustering (similar proximity as the same loci in cells without DSBs).

8. "D-compartment" and "D-sub compartment" are both used in the text. This is potential cause for confusion. Please keep nomenclature the same.

9. The legend of Fig. 3f reads "Boxplot showing the quantification of DRIP-seq read count"; however, the y-axis in the figure does not show true read count (scale trends below 0).

10. In a number of instances in the text the induced interaction between TADs with DSBs is described as "exacerbated upon DNA-PK inhibition". Exacerbated means to make worse, and it may not be clear to a non-specialist reader whether an increase or a decrease in clustering is beneficial or detrimental. I suggest replacing exacerbated with more quantitative language such as increased or decreased.

11. It is questionable as to whether the intra-chromosomal "translocations" observed are in fact translocations. According to the NCI definition of cancer terms a translocation is defined as: a genetic change in which a piece of one chromosome breaks off and attaches to another chromosome. Sometimes pieces from two different chromosomes will trade places with each other. I suggest that the authors consider changing their text to "large deletion" rather than translocation, though they could obviously suggest a role in translocations in their discussion.

Divyaa Srinivasan and Peter Fraser

Author Rebuttals to Initial Comments:

Rebuttal letter

Revision overview:

First, we would like to thank all the referees for their work and their thorough understanding of our manuscript and for providing fair and constructive comments.

In this revised version, we have now added a very large amount of new data, that we believe, really improves our manuscript:

- 1- **To address whether we can recapitulate our findings with other DSBs** (not enzymatically induced) we provide evidence that endogenous DSB also cluster (**Extended Data 2c, Fig. 2b**), that etoposide-induced DSB clustering is also regulated by R-loops (**Extended Data 7e**), and that D-comp genes also require SUN2 for optimal activation after etoposide, similarly to what we show for AsiSI-induced DSBs (**Extended Data 7g**).
- 2- **To clarify the role of R-loops in D-compartment formation**, we now show by qDRIP-seq (strand-specific high-resolution R-loop mapping) that R-loop accumulate 4h and 24h after DSB induction on D-compartment genes (and not on other, non-D, DDR upregulated genes) as does the Senataxin (SETX) by ChIP-seq (**Fig.3g-i; Extended Data 6d**). Moreover, we also found that SETX depletion triggers R-loop accrual post DSB specifically on D-compartment genes (**Fig. 4b-c**). Finally, we report that while overexpression of RNaseH1 decreases clustering (**Fig. 4a, Extended Data 7a-b**), depletion of RNaseH1 increases clustering (**Extended Data 7c-e**). Altogether, these new data strengthen the fact that R-loop forming on a subset of upregulated genes following DNA damage contribute to the formation of the D-compartment.
- 3- **We now also show D-compartment formation and D-gene targeting using an orthogonal approach.** Random Illumination microscopy allowed us to visualize multiple 53BP1 foci fusion with a high spatial (115nm*115nm*260nm) resolution (**Supplemental movies S1-6, Fig. 2d**) and we show that *GADD45A* mRNAs significantly colocalizes with γ H2AX foci (**Fig. 3e, Extended Data 5h**).
- 4- **As a mechanism for D-compartment formation, we provide evidence that it occurs through Polymer-Polymer phase separation (PPPS)** (also called bridging induced phase separation) (**Fig. 2f, Extended Data 2e-g**). We provide evidence that D-compartment formation is distinct from early DNA repair factor accrual at sites of micro-irradiation, which indeed agrees with LLPS (**Fig. 2g, Extended Data 2h**).
- 5- **We have also clarified the role of DNA-PK, and ATM in DSB clustering** (**Extended Data 3d-e**).

We would like to emphasize that the true novelty of our work is the discovery that beyond DSB clustering (which was previously reported by us and others), **DSB production triggers the formation of a novel chromatin compartment that not only comprises of the DSBs and their surrounding TAD, but also of a subset of DNA Damage Responsive genes. Both ATM-dependent 53BP1-mediated Polymer-Polymer phase separation and R-loop accrual contribute to this new DSB-induced chromatin compartment that is required for the activation of a subset of DNA Damage Responsive genes.** Beyond providing a **biological meaning for DSB clustering**, our discovery of a physical, damage-induced proximity of DDR genes with the DSBs themselves, **opens a new perspective to understand tumor suppressor gene instability.**

Referee #1:

This manuscript by Legube and colleagues addresses the role of double-strand break (DSB) clustering. Recently published work from this group has shown that loop extrusion, which is important for TAD formation, is important for H2AX spreading and DNA damage response (DDR) foci formation. It has also been shown that ionizing radiation (IR) enforces TAD formation, and that DSB clustering is dependent on the LINC complex and the phase separation properties of 53BP1. However, the function of clustering is unclear. Using their DiVA system, Legube and colleagues examined genome organization and TAD contacts around induced DSBs using Hi-C, showing essentially as they did in Arnould et al. that DSB formation increases intra-TAD contacts and triggers SCC-dependent and ATM-dependent loop extrusion. They also now show that DNA-PK inhibition enhances the process. Their studies show that multiple DSBs can cluster to form TAD cliques and that those prone to this process are in transcriptionally-active regions and have higher levels of H2AX and 53BP1. ATM is also needed for clustering (inhibition blocks clustering), while DNA-PK inhibition, by contrast, promotes clustering. Furthermore, clustering is more common/prominent in G1 cells than S or G2 cells.

The authors also show that damaged TADs, marked by H2AX and 53BP1, form a new chromatin compartment, akin to A and B compartments, that segregates from the rest of the genome and that includes genomic loci that do NOT contain a DSB, but which are areas of active transcription. They suggest that the regions of active transcription that do cluster are enriched in genes upregulated following DNA damage. They also suggest that these genes are enriched in the formation of R-loops, and the knockdown of SETX, known to increase R-loop formation, leads to increased clustering and increased expression of damage response genes. Disruption of clustering leads to less expression. Lastly, the authors show that clustering, while beneficial for the expression of certain DDR genes, can lead to increased translocations of the genes targeted to these sites and that translocations observed in cancer genomes are enriched at these genes.

Overall, this story presents several novel findings concerning the role of DSB clustering and chromatin organization and TAD structure in the DNA damage response. Although the story builds on previously published work, it extends it in new directions particularly by showing the link of clustering to gene expression. This is a relatively unexplored area, and the authors' findings open new avenues of exploration that will be of high general interest. That said, some of the conclusions are a little premature and are lacking mechanistic insight. Although ATM is proposed to regulate this pathway, how it does so is unclear, as is the role of DNA-PK in this process. The role of R-loops in the process is also somewhat tenuous, and more should be done to clarify the meaning of these observations and support the new, novel conclusions. Both are a little loosely tied to the main model at this time yet are some of the most novel data.

We would like to thank this reviewer for their comment on our study. We have now clarified the function of ATM/DNA-PK and made extensive work to support the involvement of R-loop in D-compartment formation (see summary above, and response to specific comments below).

Specific comments are below:

- 1) What is the relationship between ATM and DNA-PK in this work? Are the activities of these two kinases countering each other? If both drugs are added at the same time, is one effect dominant or lost? More information about the function of these kinases would be helpful,

particularly as the impact of DNA-PK is somewhat surprising. Is DNA-PK's impact on clustering, etc... due to a defect in DSB repair? Does loss of a downstream protein in the NHEJ pathway have the same effect as DNA-PK inhibition? Or is this some type of signaling effect?

Thanks for raising this point, that we have now tried to clarify both in the text and by adding new data. We performed immunofluorescence and Ripley analysis, which previously allowed us to identify DSB clustering in DivA cells (Caron et al, 2015). **We found that ATMi is dominant over DNA-PKi**, since clustering is lost when both inhibitors are used in combination (Extended Data 3d-e).

Altogether, our data suggests that ATM mediates DSB clustering through 53BP1 deposition on TADs, and that DNA-PK inhibition increases clustering due to the inhibition of DSB repair (and hence the persistence of 53BP1-decorated domains), for the following reasons:

- 1- Previous studies have established a role of 53BP1 in DSB clustering (Pessina et al, 2019; Kilic et al, 2019), which is in agreement with our data given that i) 53BP1 recruitment occurs on entire TADs (Clouaire et al, 2018), ii) clustering between DSBs occurs at the level of entire TADs (**Fig. 2a, Extended Data 2a-b**) and iii) clustering correlates with the level of 53BP1 accumulated around DSBs (**Fig. 2e**). To further investigate how 53BP1 could mediate chromatin self-segregation of 53BP1-decorated domains, we applied a recent methodology (Half-FRAP) to investigate fused 53BP1 foci properties *in vivo*. Our data indicate that the D-compartment forms through a segregation process named Polymer-Polymer phase separation (PPPS) or bridging induced separation, which relies on a chromatin scaffold to occur (**Fig. 2f, Extended data 2e-g**). In contrast, early 53BP1 accrual at sites of micro-irradiation agrees with Liquid-Liquid phase separation (LLPS) (**Fig. 2g., Extended data 2h**).
- 2- **ATM inhibition strongly reduces γ H2AX and 53BP1 accrual around DSB in DivA cells** without significantly affecting the repair kinetics ((Caron et al, 2015) and **Fig.1 for referees**). So **altogether these data agree with a role of ATM in clustering through 53BP1 deposition.**
- 3- Moreover, we previously reported that **DNA-PK inhibition strongly impairs repair in DivA cells without interfering with γ H2AX/53BP1 establishment** (Caron et al, 2015; see also below **Fig.1 for referees**). We also show that DSB clustering is enhanced in G1 (**Extended Data 3f, Fig. 2i**), a cell cycle stage where DSBs induced in active loci tend to persist (Aymard et al, 2017). **Hence DSB clustering coincides with delayed repair, both across the cell cycle and in presence of DNA-PKi.**
- 4- As expected, DNA-PK inhibition in presence of ATM inhibitor does not rescue DSB clustering, nor 53BP1 establishment (**Extended Data 3d-e, Fig. 1 for referees**), showing that ATM-dependent 53BP1 deposition is indeed required for clustering even in presence of persistent DSBs.

In summary we propose the following model: 1) γ H2AX/53BP1 establishment on entire TADs is mediated by ATM, 2) These γ H2AX/53BP1 covered TADs further self-segregate thanks to PPPS properties of chromatin-bound 53BP1, 3) inhibition of DNA-PK, by impairing DSB repair, triggers sustained presence of γ H2AX/53BP1 giving more time for D-compartment to form, which translates into increased clustering in our Hi-C map.

Fig.1 For referees:

Immunostaining of 53BP1 in DlvA cells after DSB induction (4h) in presence of the indicated inhibitors.

Images were acquired using RIM. ATM inhibition triggers a substantial loss of 53BP1 staining, that is not rescued by DNA-PK inhibition. We can also observe increased 53BP1 foci size in DNA-PKi treated cells in agreement with increased clustering

2) In analyzing R-loops (Fig. 3e) are the authors using data from undamaged cells or damaged cells? Are R-loops increased on the genes targeted to the DDR clusters after damage? The link between R-loops and the targeting of R-loop containing genes to clusters needs further development.

We agree with this referee that the relationship with R-loop was requiring further work. We have now added a large amount of new data to sustain this conclusion (see response to point 2 and point 3)

In our original manuscript (previous Fig. 3e), the R-loop were analyzed after DSB induction. We have now performed high-resolution, strand-specific, R-loop mapping by qDRIP-seq, both before and after DSB production (4h and 24h). We found that R-loops are indeed enriched on genes that are targeted to the D-compartment and that R-loops are further increased on these genes following DSB induction (**Extended Data 6d, Fig. 3g-h**). Of interest we found that SETX is also specifically recruited on D-compartment genes following DSB induction (**Fig. 3g, i**). Moreover, we have now also added data showing that SETX depletion triggers an increase of R-loop after DSB induction especially on D-compartment genes (**Fig. 4b-c**). Altogether **we therefore show that 1) R-loop specifically accumulate at D-compartment genes post-DSB, as does SETX, and 2) depletion of SETX increases R-loop at D-compartment genes and D-compartment formation**, further strengthening the function of R-loop on D-genes in D compartment formation.

3) Along these lines, the authors suggest that R-loops are important for clustering and in support of this idea, they show that perturbation of SETX, which has been reported to affect R-loops levels, leads to more clustering and gene expression. But is this really due to SETX's effects on R-loops or its effects on something else (e.g. transcription or repair)? The authors should test the impact of RNase H expression on R-loop levels, D-compartment formation/clustering, and the transcription of DDR genes.

In order to address this comment, we **performed RNaseH overexpression** and found that it decreases clustering (**Extended Data 7a-b; Fig. 4a**). Conversely, we found that **depletion of RNaseH1** by siRNA triggers increased clustering (**Extended Data 7c-d**). Importantly siRNA against RNaseH also increased clustering following etoposide (**Extended Data 7e**). These new data strengthen the conclusion that R-loop contribute to D-compartment formation.

In addition, could the effects of SETX on repair be affecting the underlying cause for these effects? We know SETX loss leads to more translocations and defects in repair from the authors' previous work.

We now report using ChIP-seq against SETX, that **SETX accumulates post-DSB induction on D-compartment genes (Fig. 3g, i)**, and using DRIP-seq upon SETX depletion, that **R-loop accumulate at D-compartment genes in SETX-depleted cells (Fig. 4b-c)**. Moreover, SETX depletion triggers an increase in the transcription of DDR upregulated genes, but **only for those targeted to the D-compartment (Fig. 4g)**. If the effect of SETX depletion was through deficient repair, we would expect a more general impact on the DDR (*i.e.* an increase in the transcription of all DNA Damage Responsive genes). Altogether we thus think our data rather agrees with a function of SETX in forming the D-compartment genes outside of its function at DSBs and through its ability to regulate R-loop.

4) All of the experiments are done in the same system, with a large number of DSBs induced. If the authors induce a smaller number of DSBs using a CRISPR system, would they see the same effects? Is DSB clustering of these breaks observed, and are all of the upregulated damage response genes targeted to these clusters as well? I am wondering if somehow their observation regarding the recruitment of certain genes to the clusters is tied to the location of the DSBs induced here and the large number of DSBs induced.

In order to address this point, we now provide two important new set of evidence:

- We report that **endogenous breaks**, mapped by ATM ChIP-seq in unchallenged U20S cells, also display the ability to **cluster (Extended Data 2c, Fig. 2b)**
- We report that **etoposide-induced DSBs also cluster**, in a manner regulated by R-loops (**Extended Data 7e**), SETX and SUN2 (see below **Fig. 2 For referees**). Disrupting **etoposide-induced DSBs clustering** by SUN2 depletion, also impaired activation of D-compartment genes but not non-D compartment genes (**Extended Data 7g**, see also **Fig.3 for Referees**).

Altogether this suggests that D-compartment formation and its ability to activate the DDR is not a unique feature to AsiSI-induced DSB and also hold true for endogenous as well as etoposide-induced DSBs.

Fig.2.

Fig.2 For referees:

Clustering of etoposide-induced γ H2AX foci was analyzed using high-content microscopy. siRNA against RNaseH1 and SETX (both increasing R-loops) increased clustering, while siRNA against SUN2 decreased clustering, as observed for AsiSI-induced DSB in DivA cells.

5) Around line 313, the authors link R-loop accrual to non-coding RNAs, 53BP1 foci formation, etc. It would help if they elaborated on this point more as these are not really R-loops. The logic behind these statements is not entirely clear and seems to be a bit of a stretch.

We have now added references of previous work showing the involvement of R-loops in chromosome architecture.

6) Do DNA-PK inhibition and SETX knockdown have additive effects on these processes? Are they somehow working together? The role of DNA-PK needs further exploration.

As mentioned above, our data suggest that DNA-PKi triggers increased clustering due to its potential to inhibit repair. In agreement, SUN2 depletion still decreases D-compartment gene expression even in presence of DNA-PK inhibition (see below, **Fig. 3 For referee**). Moreover, SETX depletion still increases D-compartment gene expression in presence of DNA-PKi (see below **Fig. 4 for referee**). Altogether these data agree with an effect of DNA-PK inhibition due to defective repair. We have not added these data in the manuscript (given the already consequent amount of data) but could do so if this referee thinks it is essential.

Fig.3 For referees:

cDNA level (normalized to RPL0) of *PLK3* (Up, D gene); *PPM1D* (Up, non-D gene) and *UTP18* (not Up, non-D gene) before and after etoposide with or without DNAPK-inhibitor, upon Ctrl or Sun2 depletion by siRNA (RT-qPCR N=4 biological replicates)

Fig.4 For referees:

cDNA level (normalized to RPL0) of *GADD45A* (Up, D gene); *PPM1D* (Up, non-D gene) and *UTP18* (not Up, non-D gene) following DSB induction in DlvA cells (+OHT) upon Ctrl or SETX depletion as indicated (RT-qPCR N=3 biological replicates)

7) Minor point: The authors note (p, 11, end of the section) that there is a role for the D compartment in activation of the DNA damage response. Saying this is the DNA damage response is a rather broad statement and should be modified to be more specific and targeted to the transcriptional activation of damage-responsive genes.

This has been modified.

Referee #2:

Arnould et al. use the DlvA cell line for the controlled formation of DSBs at defined sites across the genome to follow and characterize 3D chromosome dynamics by using chromosome conformation capture and ChIP-seq approaches. They report that upon DSB formation, contact frequencies were increased within TADs with DSBs, while contact frequencies between neighboring TADs were decreased. In accordance with previous studies showing reinforcement of TADs upon irradiation in mammalian cells (Sanders et al., Nat. Commun., 2020), the authors found that those changes in contact frequencies were also dependent on the kinase activity of ATM. This observation is in line with the requirement of ATM kinase

activity for the DSB-anchored loop extrusion, and it is in stark contrast with what happens under DNA-PK inhibition, which exacerbates the increase in intra-TAD contacts, increases loop extrusion and the frequency of formed translocations.

Moreover, following their previous findings that DSBs cluster in an ATM-dependent manner in G1 phase of the cell cycle, when induced within active genes (Caron et al., Cell Rep., 2015 and Aymard et al., NSMB, 2017) the authors show by Hi-C how clustering occurs between breaks within the same and different chromosomes. They report that clustering correlates with DSB-induced chromatin features that occur at the scale of an entire TAD; however, an important control is missing here that could explain these findings (see below).

Applying principal component analysis in Hi-C data upon DSB formation the authors report the formation of a new “D” compartment, which comprises both damaged and undamaged chromatin domains and is enriched in active chromatin marks and R-loops. Intriguingly, they also provide evidence that the non-damaged active sites that are recruited to D compartments are genes with dedicated roles in the DNA damage response, and their recruitment is required for optimal activation and coincide with breakpoint regions found rearranged in cancer.

Not surprisingly, clustering of DSBs promotes the formation of fusions, whose formation, as the authors showed, was dependent on 53BP1, phase separation (disrupted by treatment with 1,6-hexanediol), the LINC complex, ARP2, an actin-branching factor, and cohesin (SCC1 depletion, although doesn't display clustering defects).

Overall, this is a high-quality study that builds upon already published findings to further describe principles of chromosome dynamics upon DSB formation. They use innovative methodologies to obtain data of high quality, which are presented with clarity. In few occasions, the conclusions drawn by the authors were not fully supported by the obtained data, and further experiments are required to support their claims (see below). Having said that, overall, the novel aspects presented by this manuscript are preliminary and lack mechanistic insights for publication in Nature.

We would like to thank this referee for their comments. We have now included new data and analyses to address this referee concerns. Our new data provide additional insights on the mechanism that allows the formation of D-compartment through polymer-polymer phase separation (PPPS) (Fig. 2d, Extended Data 2e-g) and on the contribution of R-loop in this process (see below answer to specific comments).

Addressing the criticism raised below will substantially improve the manuscript:

- The authors report that clustering of multiple TADs upon DSB induction correlated with several DSB-induced chromatin features that occur at the scale of an entire TAD, including γ H2AX, 53BP1, and ubiquitin chain levels as well as the depletion of histone H1 around DSB detected by CHIP-seq. I am wondering whether this is just a reflection of the cutting efficiency of the DSBs in the population. In other words is it possible that DSBs with higher cutting efficiency in the population show higher changes in DDR events (recruitment of DDR factors, histone eviction, etc) and cluster the most? Correlating clustering with cutting efficiency determined by BLESS will directly answer this.

Thanks for raising this point which is an important point. Indeed, cutting efficiency by AsiSI, varies across the genomic locations (Iacovoni et al, 2010, Clouaire et al, 2018).

To address this point, we therefore compared clustering of categories of DSBs that we know are cleaved to an equivalent level. Indeed, we previously identified two subsets of DSBs, an

HR-prone subset, displaying increased Rad51 recruitment in G2, and an NHEJ-prone subset (low Rad51 recruitment) despite equivalent cleavage analyzed by BLESS (Clouaire et al, 2018, we reproduce some figures below **Fig. 5a-b for referees**). Importantly, the HR-prone subset comprises of DSBs induced in transcribing loci (enriched in RNAPII), and **display more 53BP1** (Clouaire et al, 2018) (**see below Fig. 5c-d. for referee**).

We show that, in agreement with Fig. 2e, **HR-prone DSB** (induced in RNAPII enriched loci and displaying high 53BP1 levels) **cluster more than NHEJ-prone DSBs** (**Extended Data 2d**).

Fig.5.

Fig.5 For referees:

These panels are a reproduction from Clouaire et al, 2018.

a. HR-prone and NHEJ-prone DSBs subsets were identified using RAD51 and XRCC4 ChIP-seq ratio. Both subsets display equivalent BLESS level in average. b. Both HR- and NHEJ-prone DSBs display equivalent BLESS read count around DSB. c. Both subsets display different 53BP1 accrual around DSB (1Mb window), 53BP1 accumulating more at HR-prone DSBs. d. Average profiles of 53BP1 around HR- and NHEJ-prone DSBs. γ H2AX profiles behave in the same manner (see Clouaire et al, 2018)

- It is unclear to me whether the experimental description of the D compartment is correct and truly corresponds to any physiologically relevant events that are observed in vivo. The main concern is that the PCA analysis on the Hi-C map of the +DSB condition alone should have revealed the reported “new” compartment, but as the authors clearly showed that was not the case. The PCA analysis of differential Hi-C maps on the ratio of the contact matrices (+DSB/-DSB condition) is therefore not meaningful and may exacerbate differences due to its nature.

Indeed, as this referee mentions we could not identify a new compartment from the Hi-C +DSB only: we think this was expected since PCA analysis can only identify two compartments at a time for each of the principal component. We know from previous studies that DSB induction does not “dissolve” heterochromatin foci, so performing a PCA analysis upon DSB induction still identifies the B and the non-B (*i.e.* A) compartments as the first PC.

It is unclear for us why this referee thinks the PCA analysis on differential matrix is not meaningful and will exacerbate differences. Mathematically it will indeed identify the genomic locations that behave in a similar manner regarding 3D interactions (which was the basis of B and A compartment discovery), and self-segregate more together after DSB than before DSB. In other words, applying the PCA on the differential matrix allows us to set free from the main chromatin compartment (heterochromatin) that is not drastically modified by DNA damage induction and will always come up as a strong component of chromatin compartmentalization. Applying PCA on the differential matrix allows us to focus on the changes between those two conditions and abolishes the steady-state chromatin compartmentalization.

This PCA analysis was efficient in retrieving γ H2AX/53BP1 domains clustering (high similarities between the PC1 and γ H2AX distribution (**Fig. 3a-b, Extended data 5a-d**) which validates the methodology used here. The main advantage of applying this methodology is that it allowed

us to identify additional loci that also physically segregate with this compartment, loci that we have further extensively characterized.

Perhaps the issue this referee wants to point out is more about naming this first PC a “chromatin compartment”? We believe this is justified given that the fact that γ H2AX/53BP1 form visible large bodies in the nucleus. However, we could also call it a sub-compartment, if this referee believes this is more appropriate.

- Can the authors confirm their most important findings when inducing DSBs by other means (such as CRISPR, IR, etc)?

This is indeed an important comment. In order to address this point, we now provide two important new set of evidence:

- We mapped **endogenous breaks** by ATM ChIP-seq in unchallenged U2OS cells, and show that they also display the ability to **cluster (Extended Data 2c, Fig. 2b)**.
- We report that **etoposide-induced DSBs also cluster**, in a manner that is regulated by R-loops (**Extended Data 7e**), SETX and SUN2 (see above **Fig. 2 For referees**). Disrupting **etoposide-induced DSBs clustering** by SUN2 depletion, also impaired activation of D-compartment genes but not non-D compartment genes (**Extended Data 7g**, see also **Fig.3 for Referees**).

Altogether this suggests that D-compartment formation and its ability to activate the DDR is not a unique feature to AsiSI-induced DSB and also holds true for endogenous as well as etoposide-induced DSBs.

- I am wondering what fraction of the genes with changes in gene expression upon DSB formation is that was targeted in the D compartment? The authors show that genes that upregulated following DSB induction displayed a higher D compartment signal compared to genes that were not upregulated after DSBs (Fig. 3d, Extended Data 4g). Can the authors show how many genes were differentially expressed upon DSB induction (upregulated, downregulated, show no changes) and what was the fraction of those that were targeted in the D compartment?

Numbers of genes in each category are indicated in the material and methods section, and we now mentioned in the results section the % of upregulated genes that are targeted to the D compartment. On the chromosomes where the D compartment was identified (carrying enough DSBs) we had 77 upregulated genes post DSB induction. 45 of them were actually comprised in the D-compartment, so 58% of upregulated genes are targeted in the D-comp (*NB. The definition of a “D” gene was very stringent here as it has to be retrieved in each of our 3 Hi-C biological replicates*).

In presence of DNA-PK inhibitor, we could identify the D compartment on more chromosomes (since repair is inhibited and clustering increased). We had 358 genes upregulated in response to DSBs on these chromosomes and among those genes, 223 (62%) were identified in the D-compartment (*NB. Here the definition of a D-gene is much less stringent as we had only one Hi-C dataset with DNA-PK inhibitor*).

- The authors report that genes identified in the D compartment displayed increased R-loop levels compared to non-D genes (Fig. 3f) and vice-versa, genes exhibiting high R-loop levels

showed higher D-sub compartment signal (PC1 value) when compared to genes with low level of R-loops (Fig. 3g). Do the authors think that this is a direct effect of R-loops per se, or could it be a secondary effect of transcription levels that could also affect the levels of R-loops? In the same direction, SETX depletion could also affect transcription levels (not only the levels of R-loops) and thus increased transcription could promote clustering. Can the authors identify what is causal here? For example, the authors could use alternative means to increase R-loops and try to rescue clustering by overexpression of factors that resolve R-loops (such as RNase H1).

Indeed, we agree with this referee that disentangling the effect of transcription and R-loop is an important point that we have tried to address in the revised manuscript.

As mentioned in the summary and in response to referee 1 point 2, we have now added a large number of additional data to further strengthen the link between R-loop and D-compartment genes targeting.

- We performed both **depletion and overexpression of RNaseH**, and found opposite effect on clustering (**Extended Data 7a-e, Fig. 4a**).
- We performed high-resolution strand-specific mapping of R-loop (qDRIP-seq) before DSB and at 4h and 24h after DSB. We show that **R-loop accumulate specifically at D-compartment genes and not at other non-D genes following DSB induction (Extended Data 6d, Fig. 3g-h)**
- We show by ChIP-seq that **SETX is recruited at D-compartment genes post DSB induction and not at other non-D genes (Fig. 3g, i)**.
- We show by DRIP-seq that **SETX depletion triggers specific R-loop accrual at D-comp genes but not at non-D genes (Fig. 4b-c)**.

Moreover, we show that, **for equivalent transcription levels, D-compartment genes display elevated R-loop compared to non-D compartment genes (Extended Data 6b, see green tracks)**. Actually, upregulated D-comp genes displayed rather less RNA-seq signal than upregulated non-D genes (**see below Fig. for referee 6b, bottom panel**).

We also provide below few examples where TT-seq was used instead of RNA-seq to measure nascent transcription rather than steady state levels (**Fig. 6a for referee**). Again, upregulated D-comp genes displayed rather less TT-seq signal than upregulated non-D genes (**see below Fig. for referee 6b, top panel**).

Taken altogether these data suggest that it is R-loop accumulation rather than transcription activity itself that contributes to D-compartment targeting.

Fig. 6.

Fig.6 For referees:

a. Genomic tracks of DRIP-seq and TT-seq obtained after DSB induction, for *PLK3* (D-compartment gene), and *SERBP1* and *CDC42* (two non-D genes). Despite higher level of nascent transcription than *PLK3*, these two genes are not targeted to the D compartment.

b. Quantification of TT-seq +DSB (top panel) and RNA-seq +DSB (bottom panel) at genes upregulated after DSB induction and either comprised in D-compartment (pink) or not (purple).

- How does ATM inhibition influence translocation frequency?

We performed qPCR analysis of intra-chromosomal and inter-chromosomal illegitimate rejoining events. We found that, unlike as expected given the function of ATM in both loop extrusion and DSB clustering, ATM inhibition increases all types of translocations (both intra and inter chromosomal) (**Fig. 7a For Referees**). We also reanalyzed data that were produced using Amplicon-seq in presence of both ATM and ATR inhibitors (Bader et al., *in revision*, also BioRxiv, <https://doi.org/10.1101/2021.12.08.471781>), and similarly found that ATM/ATR inhibition triggers an increase of translocations. We believe that the consequences of ATM inhibition on DSB repair events (such as decreasing resection) likely promote translocations through NHEJ mechanisms and bypass the effect of loop extrusion and clustering.

Fig. 7.

Fig.7 For referees:

a. qPCR analysis of 4 illegitimate rejoining events (TR3 and TR6=intra chromosomal; TR13 and TR14 inter-chromosomal) in presence of ATMi or not.

b. Intra-chromosomal (blue) or inter-chromosomal translocations (yellow) were quantified using multiplexed amplification followed by high-throughput sequencing (amplicon-seq) between 20 different DSBs induced in the DivA cell line, with ATM+ATR inhibitor or not. N=4 independent replicates.

Referee #3:

In this manuscript the authors report that upon induction of DSBs, damaged TADs cluster together in an ATM-dependent manner forming a DSB-induced compartment (“D”-

compartment). They report that these changes in genomic architecture caused by DSB induction targets DDR genes to D compartments, which augments their transcription. They also propose that DSB clustering could promote translocations and genome instability since many DDR genes are tumor suppressor genes. Overall, the data is of very high quality, presented in a logical flow, and potentially represents important new insights that would be of interest to both specialists and a general readership. I believe this manuscript could be considerably strengthened with additional validation (see below).

Major comments:

1. D-compartment formation shown by fluorescence in-situ hybridization before and after DSBs would significantly strengthen this manuscript. For example, in agreement with Hi-C and 4C, immune-FISH should show that “cluster-prone” DSBs cluster together within γ -H2AX foci, as opposed to control DSB loci, and the same loci in cells without DSBs.

We agree with these referees that visualizing clustering using immuno-FISH experiment would be a great asset to this study. Unfortunately, we did not succeed in developing a high-throughput assay with enough resolution (using HD-FISH, Gelati et al, 2019) in order to visualize these events. We also worked quite extensively to implement the recent Casilio approach (Clow et al, 2022, PMID: 35387989) to follow in real time individual loci, as this could allow to capture even rare events in the cell population. Unfortunately, our attempt has been yet unsuccessful (high background signal, multiple non-specific foci...). We apologize for not supplying such data yet, however, we now provide in the revised manuscript, microscopy analyses that confirm some of our findings:

- i) We followed, using Random Illumination Microscopy (RIM), DSB clustering in time lapse with a very high spatial resolution.
- ii) We analyzed the properties of the fused 53BP1 foci using Half-FRAP and showed that they agree with Polymer-Polymer Phase separation (PPPS) (relying on a chromatin substrate to form condensate) rather than Liquid-Liquid Phase Separation (LLPS) (**Fig. 2f and Extended data 2g-i**). We report that 53BP1 decorated D-compartment is molecularly distinct from sites of micro-irradiation at early stage (10min) where PAR chains accumulate and where 53BP1 diffusive properties agree with LLPS. (**Fig. 2g, Extended data 2h**)
- iii) We used RNA in situ hybridization to show that *GADD45A* mRNA (D-gene) colocalizes more often with γ H2AX foci than two other non-D genes (*PPIB* and *CCL2*) (**Fig. 3e and Extended data 5h**).

2. On p. 5 (line 97) the authors mention that some γ -H2AX domains are able to interact with more than one other γ -H2AX domain. They imply that their data indicates that multiple damaged TADs are capable of simultaneous clustering. However, it cannot be assumed to be the case since this is population Hi-C data. This would be an ideal set of TADS for a multi-label DNA FISH experiment to show simultaneous associations of multiple TADs in a single D-compartment following DSB.

We thank the reviewers for this comment. Here rather than multi-FISH experiments we have now imaged with high resolution both in space and time, DSB clustering in DlvA cells using the recently described RIM (Mangeat et al, 2020, PMID: 35474693). We could frequently observe multiple foci clustering (**Supplemental Movies S1-3, Fig. 2d**).

On a different note, our collaborator, Jop Kind (Hubrecht Institute, Utrecht) has performed single-cell Dam-ID in DlvA cells against various DSB repair factors. They actually found that multiple AsiSI induced DSBs show simultaneous repair protein occupancy within the same cell. Importantly, this observation was much stronger for DSBs that form contacts (based on our Hi-C data), than for non-contacting sites. This could be interpreted as “coordinated” repair center, which would also be in agreement with TAD cliques (Jop Kind, personal communication).

3. The authors suggest that increased clustering of DDR genes in D-compartments plays a role in their transcriptional upregulation. RNA FISH with an immuno-marker of D-compartments would greatly strengthen/validate this conclusion and rule out some downstream consequence or diffusible signal from D-compartment formation or DSBs, such as phosphorylation of a transcription factor (for example) in response to DSBs or D-compartment formation.

We have now performed RNA-FISH against *GADD45A* (D-compartment gene) and against two non-D genes (*PPIB* and *CCL2*). We found that *GADD45A* significantly colocalizes more with γ H2AX foci (D-comp) than the two other genes (**Fig. 3e and Extended data 5h**).

Minor comments:

4. It is not clear what the distribution of DSBs is across all chromosomes. The data indicates that 80 DSBs were induced by AsiSI in this study. What is the distribution of these breaks across all chromosomes? Was this similar to the DSB distribution obtained in Clouaire et al. (ref. 3 in manuscript)? A table such as “Table S1” included in Clouaire et al. (2018) will make this information more transparent to the reader.

Indeed, the DSBs analyzed in this study is the list published previously (Clouaire et al, 2018). We refer now specifically to Table S1 of Clouaire et al, in the methods section. They are distributed across all chromosomes, with the Chr 1 and Chr 17 being the most enriched in DSBs.

5. The methods section indicates that the authors “were able to extract the D-compartment on chromosomes 1, 17 and X”. Does this mean that D-compartment was not found on other chromosomes?

Actually, PCA were applied on each chromosome individually (Chromosomal Eigen Vector or CEV) since we did not manage to run PCA at the genome wide scale (not enough computational resources). Therefore, in order to be able to detect the D-compartment on a chromosome we need enough DSBs induced on that chromosome. This was the case for chr1, 17 and X in normal conditions. In presence of DNA-PK inhibition, that blocks DSB repair and enhances clustering, we could retrieve the D-compartment on Chr 1, 2, 6, 9, 13, 17, 18, 20, and X. Importantly, the features of D-genes were similar if we used our D-genes list on 1/17/X (identified without DNA-PKi) or the extended list (identified with DNA-PKi) (see **Extended Data 5e, 5g**).

6. It is suggested that the DSB clustering “entirely dependent on ATM, exacerbated upon DNA-PK inhibition...” More validation/data presentation is required to make this more obvious to the reader. Inhibition of DNA repair (ATM and DNA-PK inhibition) may be causing these cells to

undergo apoptosis/pyknosis. Do genome-wide Hi-C heat maps show other normal higher order features upon DNA-PK and ATM inhibition, for example TAD structure and A/B compartments? This is not possible to decipher from the plots included in Fig. 2.

We apologize about that. Indeed, we do not extensively present these data in the manuscript due to a lack of space. DNA-PK and ATM inhibition did not noticeably modified A/B compartmentalization (**Extended Data 4, and Fig. 8a for referee**), nor made strong modifications in *cis* contact probabilities (**Fig. 8b-c for referee**). We could add these data in the manuscript if the referees believe it is important for the reader.

Moreover, of note inhibitors are only added 1h before OHT addition (DSB induction) and then maintained during the 4h of DSB induction. We know from our past work in the lab that at this time point, we do not see any morphological changes, nor dapi-dense cells (Caron et al, 2015).

Fig.8 For referees:

a. Genomic tracks of A /B compartment (PC1) computed on Hi-C data -DSB, +DSB, +DSB+ATMi, and +DSB+DNA-PKi as indicated.

b. Contact probability as a function of distance computed on Chr13 for all the above conditions, as indicated

c. Example of Hi-C heatmaps on chr13. Left panel, +DSB vs +DSB+ATMi; Right panel +DSB vs +DSB+DNA-PKi

7. Fig. 2e shows some residual contact between DSBs in the ATMi panel. Although this would suggest that DSB clustering is somewhat compromised in the absence of ATM, it is not sufficient to suggest complete dependence of DSB clustering on ATM, unless this phenomenon can be shown in ATMi-depleted cells by microscopy. Complete dependence could be indicated if DSB loci in ATMi cells would show no evidence of clustering (similar proximity as the same loci in cells without DSBs).

We have now rephrased these sentences. We also show the γ H2AX foci distribution in ATMi +DNA-PKi treated cells using RIM high resolution microscopy (**Extended Data 3d**), as well as a Ripley analysis of the spatial distribution of these foci (**Extended Data 3e**). These analyses show that ATM inhibitor severely compromised DSB clustering.

8. “D-compartment” and “D-sub compartment” are both used in the text. This is potential cause for confusion. Please keep nomenclature the same.

Thanks for pointing this out. We have now used D-compartment all throughout the text.

9. The legend of Fig. 3f reads “Boxplot showing the quantification of DRIP-seq read count”; however, the y-axis in the figure does not show true read count (scale trends below 0).

Actually, Fig. 3f shows normalized readcount for DRIP-seq (no negative values).

10. In a number of instances in the text the induced interaction between TADs with DSBs is described as “exacerbated upon DNA-PK inhibition”. Exacerbated means to make worse, and it may not be clear to a non-specialist reader whether an increase or a decrease in clustering is beneficial or detrimental. I suggest replacing exacerbated with more quantitative language such as increased or decreased.

Thanks for pointing this, we did not know it since in french it does not convey the same meaning. We have replaced this word throughout the text.

11. It is questionable as to whether the intra-chromosomal “translocations” observed are in fact translocations. According to the NCI definition of cancer terms a translocation is defined as: a genetic change in which a piece of one chromosome breaks off and attaches to another chromosome. Sometimes pieces from two different chromosomes will trade places with each other. I suggest that the authors consider changing their text to “large deletion” rather than translocation, though they could obviously suggest a role in translocations in their discussion.

We agree and have now replaced “translocations” by “illegitimate DSB rejoining”

Divyaa Srinivasan and Peter Fraser

Reviewer Reports on the First Revision:

Referee #1:

The authors have done a thorough job of addressing my comments, and I think this is now a very appropriate for Nature and will be of interest to a broad readership.

Minor point - In the text the references to Fig. 5e, f, and g are mixed up.

Referee #2:

The authors have included additional data to strengthen their observations and provide novel directions. The revised manuscript, although it has been improved, still fails to provide definite answers on the underlying mechanisms and therefore at its current state it is rather premature for publication in Nature.

Addressing the comments below will further improve the manuscript.

- In the revised version the authors claim that LLPS contributes to the early recruitment of 53BP1 at sites of damage coinciding with PAR accumulation (also shown by previous studies), while at later time points, when 53BP1 foci have clustered, the clustering is driven by self-interactions among chromatin-bound 53BP1 molecules mediating polymer-polymer phase separation (PPPS). While the concept is interesting, the authors compare findings using a laser micro-irradiation system to assess properties of early 53BP1 recruitment at damaged sites and the AsiSI system for "late" ones (fusion of foci). Since the nature of DNA damage is expected to be very different between the two systems, the authors should use only the AsiSI system to address this, for example by using half-FRAP for newly-formed 53BP1 foci and those just undergone fusion. Moreover, the authors could further strengthen their conclusions (LLPS vs PPPS) by performing additional experiments using state-of-the-art assays used in the PS field.
- The authors used RNA FISH to follow the expression of D compartment and non-D genes and measured the colocalisation with γ H2AX foci upon formation of AsiSI-induced DSBs. They found that D compartment genes were more frequently colocalising with γ H2AX foci compared to non-D genes. Since the identified RNA loci by this technique, do not only mark the expressing gene loci, it is crucial to follow D compartment genes, non-D genes and AsiSI cut sites simultaneously by using DNA FISH. This is a crucial experiment to show whether indeed cut AsiSI sites and non-cut D genes coalesce at same centers, while non-D are excluded. BAC probe-based multicolored FISH, combined with high-throughput imaging and automated analysis to infer colocalization can be used as a simple alternative to more sophisticated high-throughput FISH probing techniques. Since only a fraction of AsiSI sites is expected to be cut in the population, immuno-FISH for γ H2AX would further help identifying the cut AsiSI sites.
- Any evidence that R-loops and not transcription in general are important for D-compartment targeting should be added to the main manuscript.
- The authors report that R-loop accumulation contributes to the formation of the D compartment. In one of the experiments overexpression of RNase H1 leads to an increase in the number of small γ H2AX foci and knockdown to the opposite effect. It would be essential to complement these data with experiments showing changes in the D compartment by Hi-C, and accompanied changes in R-loops and nascent RNA genome-wide. How does SETX deficiency influence the transcription levels in D and non-D compartment genes?
- The authors use treatment with etoposide to recapitulate their findings, showing that when DSB clustering is abrogated (e.g. SUN2 depletion), only the expression of genes identified to be

recruited at the D compartment were altered. However, those genes were identified to be recruited at the D compartment after AsiSI-induced DSBs and not etoposide. It is crucial for the validity of these results to recapitulate the existence of the D compartment, the role of R-loops, the influence on expression of DDR genes (at the D compartment) when an orthogonal way is used to induce DSBs, such as for example upon etoposide treatment or the use of CRISPR etc. In line, all further analyses e.g. of the location and frequency of DSBs (BLESS), clustering and D compartment analyses (Hi-C), R-loops (qDRIP-seq), and expression changes (TT-seq), should be performed in parallel in cells treated with the same genotoxic agent. It appears that the authors assume that the same genes are targeted to the D compartment, independently of the type and dose of the genotoxic agent, which is rather an open crucial question and not a fact.

Referee #3:

The authors have provided new data, but unfortunately these new data do not directly address our two major concerns. Both of these concerns focus on major findings/conclusions in the manuscript and are centered on the nucleic acids (DNA and nascent RNA) and not on the repair proteins themselves.

1) To show that the DNA (not repair proteins) in the immediate vicinity of a DNA DSB are found clustered in the nucleus after formation of the DNA breaks. A suitable experiment would be to use a simple DNA FISH protocol for a few (3) of the TADs that are said to form "cliques" upon DNA breaks. No super high resolution is needed. These DSB regions should show higher levels of clustering than non-DNA break regions. This type of data would indicate that the DNA of the DSB regions changed position in the nucleus relative to one another to form clusters. DSB clustering is THE major premise of this manuscript. Without direct observation of distal DSB loci changing positions leading to increased proximity one is left with some doubts. Various forms of this technique have been around for over two decades and should certainly be within the grasp of any laboratory studying nuclear organization of the genome. RIM movies showing repair protein clustering, and personal communications of multiple DSBs showing simultaneous repair protein occupancy within the same cell by Dam-ID do not address this.

2) Another important and potentially very exciting finding is the suggestion that DNA break repair genes are found in DSB foci, or D compartments, and that this may play a role in the transcription of the repair genes. This would suggest D compartments are multifunctional, involved not only in bringing broken DNA ends into proximity for eventual repair, but also playing a role in facilitating transcription of the repair machinery genes. This is very novel if it can be demonstrated. Unfortunately the authors used a kit (RNASCOPE) which, as far as I can tell from the catalog number provided, and from the number of GAD45 RNA foci per cell (Fig. S5h) that they are detecting the mRNA rather than nascent transcripts at the site of transcription. I may have been mistaken in asking for RNA FISH, which I thought would have been clear enough to suggest that intron (and not exon) probes should be used. The use of mRNA probes (complementary to exon sequences) cannot differentiate between an mRNA in the nucleus or on its way to cytoplasm, and a nascent transcript at the site of the GAD45 gene locus. The authors state, "RNAscope, which allows to localize transcription sites in nuclei^{40,41}...", but unfortunately did not apply the technique to specifically localize transcription sites.

The data in Fig. 3e shows a small difference in localization of GAD45 mRNAs compared to the control mRNAs. It is not clear, or at least I could not find, information on the number of cells counted which produced those results, making it extremely difficult to get an idea of the overlap between the GAD45 mRNA and gammaH2AX foci. This needs to be explicitly stated since there are multiple GAD45 mRNA foci and multiple gammaH2AX foci per cell. One could imagine cells with one or two GAD45 foci overlapping with gammaH2AX foci, or perhaps cells with 5 or 7 GAD45 foci overlapping with gammaH2AX foci. Knowing which, if any, of these GAD45 foci is the transcribing gene is not possible as presented, nor with the probes used. If the authors wish to conclude that

repair genes are transcribed in the D compartments they need to show nascent transcripts (inton containing).

Author Rebuttals to First Revision:

Point by point response to Referees' comments:

First, we would like to thank the referees and to acknowledge their thorough understanding of our work.

We were happy to see that both referee 1 and 2 appreciated the amount of additional data provided in the revision, that we believe really strengthened the original manuscript.

Referee #1:

The authors have done a thorough job of addressing my comments, and I think this is now a very appropriate for Nature and will be of interest to a broad readership.

Minor point - In the text the references to Fig. 5e, f, and g are mixed up.

We thank the referee for their comment and their work on our revised manuscript. We were happy to see that they found our revision satisfactory. Thank you for spotting this error, we have corrected the reference to these figures.

Referee #2:

The authors have included additional data to strengthen their observations and provide novel directions. The revised manuscript, although it has been improved, still fails to provide definite answers on the underlying mechanisms and therefore at its current state it is rather premature for publication in Nature.

We thank the referee for their assessment of our revised manuscript.

Addressing the comments below will further improve the manuscript.

- In the revised version the authors claim that LLPS contributes to the early recruitment of 53BP1 at sites of damage coinciding with PAR accumulation (also shown by previous studies), while at later time points, when 53BP1 foci have clustered, the clustering is driven by self-interactions among chromatin-bound 53BP1 molecules mediating polymer-polymer phase separation (PPPS). While the concept is interesting, the authors compare findings using a laser micro-irradiation system to assess properties of early 53BP1 recruitment at damaged sites and the AsiSI system for “late” ones (fusion of foci). Since the nature of DNA damage is expected to be very different between the two systems, the authors should use only the AsiSI system to address this, for example by using half-FRAP for newly-formed 53BP1 foci and those just undergone fusion. Moreover, the authors could further strengthen their conclusions (LLPS vs PPPS) by performing additional experiments using state-of-the-art assays used in the PS field.

We would like to emphasize here that using half-FRAP on laser-induced 53BP1 foci was more a mean to control our experimental system. Indeed, to validate that the half-FRAP assay can detect LLPS of 53BP1 in our hands, we used laser-induced damage sites as a positive control, since LLPS was already described in these conditions (e.g., PMID 31267591). We have rephrased this part in the manuscript to make this clearer to the reader.

However, given that 53BP1 foci laser-induced at early time point and AsiSI-induced at late time points, also differed in their content in term of PAR chains, we thought it was worth reporting this difference. We agree with this reviewer that performing half-FRAP on newly assembled 53BP1 foci right after AsiSI induction would have been ideal. Unfortunately, it is technically not possible since at such early time points, the AsiSI-induced 53BP1 foci are too small for half-FRAP. We could remove the PAR chain detection experiment, and just leave the half-FRAP analysis on micro irradiation sites as a mean to validate our ability to detect LLPS, if this referee thinks it is better.

Finally, we would like to emphasize here that we actually used the best validated method (half-FRAP) that allows to distinguish between PPPS and LLPS (PMID: 36526633; PMID: 32101700). Classical full-FRAP data cannot distinguish between PPPS and LLPS, and neither can 1,6-hexanediol treatment (PMID 36526633), which is in addition toxic to cells and induces chromatin condensation (e.g., PMID 33536240).

- The authors used RNA FISH to follow the expression of D compartment and non-D genes and measured the colocalisation with γ H2AX foci upon formation of AsiSI-induced DSBs. They found that D compartment genes were more frequently colocalising with γ H2AX foci compared to non-D genes. Since the identified RNA loci by this technique, do not only mark the expressing gene loci, it is crucial to follow D compartment genes, non-D genes and AsiSI cut sites simultaneously by using DNA FISH. This is a crucial experiment to show whether indeed cut AsiSI sites and non-cut D genes coalesce at same centers, while non-D are excluded. BAC probe-based multicolored FISH, combined with high-throughput imaging and automated analysis to infer colocalization can be used as a simple alternative to more sophisticated high-throughput FISH probing techniques. Since only a fraction of AsiSI sites is expected to be cut in the population, immuno-FISH for γ H2AX would further help identifying the cut AsiSI sites.

We thank the referee for this comment. In our revised manuscript, we now provide RNAscope data using probes targeting intronic sequences. The use of such probes indeed allowed to detect only 1-2 spots in the nuclei, thus corresponding to sites of transcription. We report that *PLK3* (D-gene, upregulated) significantly colocalizes more with γ H2AX foci than *CDC42* and *LPHN2* (two non-D genes) (Fig. 3e).

As for DNA-FISH, the validation of Hi-C data by DNA FISH, has been a controversial issue in the 3D field, and is far from being trivial. The nature of Hi-C (3C-based assay) and DNA FISH experiments are different, with Hi-C capturing only very close interactions (proximity ligation in the nm-scale) and DNA FISH detecting colocalization over longer ranges (imaging after deconvolution in the 100 nm-scale). Reconciliation of these data therefore, in the best situation, requires the acquisition of a very large number of single cells and thus high-throughput microscopy, whereas in the worst situation, findings are not compatible. This “paradox” has been commented in detail by others (PMID: 27760553; PMID: 28604723). Combined with the heterogeneity of DSB induction in our cells, validating our Hi-C data (*i.e* the identity of clustered loci) by FISH would thus require high throughput immuno-DNA FISH. This is technically quite challenging and would require without doubt several months of optimization.

We also would like to mention here that we made some attempt to use the recently reported Casilio system (PMID: 35387989) to follow the dynamics of individual loci in the nucleus, but we have not been successful so far.

- Any evidence that R-loops and not transcription in general are important for D-compartment targeting should be added to the main manuscript.

We agree with this comment and have now added an additional panel to show that upregulated D-genes are not in average more transcribed than up-regulated non-D genes (Fig. 3i)

- The authors report that R-loop accumulation contributes to the formation of the D compartment. In one of the experiments overexpression of RNase H1 leads to an increase in the number of small gH2AX foci and knockdown to the opposite effect. It would be essential to complement these data with experiments showing changes in the D compartment by Hi-C, and accompanied changes in R-loops and nascent RNA genome-wide.

While we agree that in an environment of unlimited time and resources all these high throughputs assays would add additional information to our study, we believe it is not necessary, nor reasonable to add this additional work to validate our current conclusions.

Indeed, we already provide convincing evidences that R-loops regulate D-compartment formation. In our manuscript we present a comprehensive set of data upon Senataxin depletion with qDRIP-seq, RNA-seq and Hi-C experiments. Our data indicated that SETX depletion triggered an increased D-compartment formation, together with a specific increase of R-loop at D-genes compared to non-D genes, as well as a specific increased transcription post DNA damage on D-genes compared to non-D genes (Fig. 4g). We further validated the influence of R-loop accumulation on clustering using IF by both performing overexpression of RNaseH1 and siRNA against RNaseH1, which behaved as expected (*i.e.* OE RNaseH1 decreased clustering while siRNaseH1 behaved as siSETX and increased clustering). Therefore, we do not see the added value of doing a validation using all genome-wide assays upon OE RNaseH1 and RNaseH1 knock down. Moreover, on a technical note, overexpression of RNaseH1 is challenging and performing these large-scale experiments require large numbers of cells. Therefore, we have no guarantee that such large-scale experiments are in fact technically feasible (for instance to our knowledge nobody has performed Hi-C in such condition).

How does SETX deficiency influence the transcription levels in D and non-D compartment genes?

We show by RNA-seq in SETX proficient and deficient cells that D-comp genes are upregulated upon SETX depletion as compared to non D-genes (Fig. 4g), showing a selective effect of SETX deficiency on D-compartment genes.

- The authors use treatment with etoposide to recapitulate their findings, showing that when DSB clustering is abrogated (e.g. SUN2 depletion), only the expression of genes identified to be recruited at the D compartment were altered. However, those genes were identified to be recruited at the D compartment after AsiSI-induced DSBs and not etoposide. It is crucial for the validity of these results to recapitulate the existence of the D compartment, the role of R-loops, the influence on expression of DDR genes (at the D compartment) when an orthogonal way is used to induce DSBs, such as for example upon etoposide treatment or the use of CRISPR etc.

In line, all further analyses e.g. of the location and frequency of DSBs (BLESS), clustering and D compartment analyses (Hi-C), R-loops (qDRIP-seq), and expression changes (TT-seq), should be performed in parallel in cells treated with the same genotoxic agent. It appears that the authors assume that the same genes are targeted to the D compartment, independently of the type and dose of the genotoxic agent, which is rather an open crucial question and not a fact.

While we agree that such experiments could add interesting insights, performing Hi-C, BLESS, qDRIP-seq as well as TT-seq upon etoposide would require a considerable amount of work and resources that we believe is not necessary to validate our current conclusions.

Indeed, in order to validate our findings using another type of DSB induction, we first confirmed that, as expected from our data acquired with the DlvA system, DSBs induced by etoposide do indeed cluster in a manner that is regulated by OE RNaseH1 (Fig.S7e) and by Sun2 and SETX (initial rebuttal letter Fig. 2 for referees; *NB: this could be added in the manuscript*).

From there, we further showed that the expression of two genes, identified in DlvA cells as upregulated by the DDR and targeted to the D-comp, and also found upregulated upon etoposide (*PLK3* and *RNF19B*) are also selectively regulated by the depletion of Sun2 upon etoposide treatment, compared to upregulated non-D genes, or non-regulated genes (Fig. S7g. *NB: we have now added on this figure additional non-D, non-regulated genes*). Of note, our goal was not to state that the exact same genes are all targeted to the D-compartment upon etoposide, and we agree that it is likely that the DNA Damage Response (and hence the genes targeted to the D-compartment) may be substantially different between AsiSI and etoposide induced DSBs. Rather, from our analyses performed upon enzymatically induced DSB, we made predictions which indeed revealed to hold true for other types of DSBs.

Of note, as additional evidence, we further showed using cancer genome databases (and hence a set of data completely different from our data generated in AsiSI expressing U2OS cells) that, as predicted, D-genes display a higher translocation frequency compared to non-D, yet DSB-induced, genes.

Altogether we believe that our additional work validate that the D-compartment can form upon other DSB inducing agents.

Referee #3:

The authors have provided new data, but unfortunately these new data do not directly address our two major concerns. Both of these concerns focus on major findings/conclusions in the manuscript and are centered on the nucleic acids (DNA and nascent RNA) and not on the repair proteins themselves.

1) To show that the DNA (not repair proteins) in the immediate vicinity of a DNA DSB are found clustered in the nucleus after formation of the DNA breaks. A suitable experiment would be to use a simple DNA FISH protocol for a few (3) of the TADs that are said to form “cliques” upon DNA breaks. No super high resolution is needed. These DSB regions should show higher levels of clustering than non-DNA break regions. This type of data would indicate that the DNA of the DSB regions changed position in the nucleus relative to one another to form clusters. DSB clustering is THE major premise of this manuscript. Without direct observation of distal DSB loci changing positions leading to increased proximity one is

left with some doubts. Various forms of this technique have been around for over two decades and should certainly be within the grasp of any laboratory studying nuclear organization of the genome. RIM movies showing repair protein clustering and personal communications of multiple DSBs showing simultaneous repair protein occupancy within the same cell by Dam-ID do not address this.

We believe that super-resolution live imaging using a state-of-the-art microscopy set up (RIM) unequivocally allowed us to show the clustering of multiple DSBs. We disagree with this comment which implies that by imaging 53BP1 we may just image unbound proteins and not chromatin. The use of 53BP1 as a marker of DSB has been used for nearly a decade, and in the repair field it is univocally accepted that 53BP1 foci in the nucleus are chromatin-bound. Our FRAP data on 53BP1 foci also agree with a typical chromatin-bound protein behavior (presence of a mobile and an immobile fraction). Thus, following the behavior of 53BP1 foci in living cells is an excellent alternative to DNA FISH in fixed cells to demonstrate that several DSB-induced foci can cluster. Furthermore, such super resolution RIM imaging of DDR foci (the volumetric resolution in live is 9.3 time better than a confocal or 5 time higher than AiryScan) has never been performed before and provided unprecedented information on the clustering process such as the morphology at a high resolution and the dynamics of the fusion that cannot be achieved using DNA-FISH. Altogether, we believe that we do provide a “direct observation of distal DSB loci changing positions leading to increased proximity”.

Moreover, the validation of Hi-C data by DNA FISH, has been a controversial issue in the 3D field (commented in detail by others (PMID: 27760553; PMID: 28604723)). It is far from being trivial, given the different nature of 3C-based assay and DNA FISH experiments, and combined with the heterogeneity of DSB induction in our cells, validating our Hi-C data (*i.e* the identity of clustered loci) by FISH would require high throughput immuno-DNA FISH, that would require without doubt several months of optimization.

We believe that our super resolution live imaging sufficiently shows multiple DSB clustering, thus validating the quality of our Hi-C data, which on their side provide the identity of the DSBs able to coalesce together.

2) Another important and potentially very exciting finding is the suggestion that DNA break repair genes are found in DSB foci, or D compartments, and that this may play a role in the transcription of the repair genes. This would suggest D compartments are multifunctional, involved not only in bringing broken DNA ends into proximity for eventual repair, but also playing a role in facilitating transcription of the repair machinery genes. This is very novel if it can be demonstrated. Unfortunately, the authors used a kit (RNASCOPE) which, as far as I can tell from the catalog number provided, and from the number of GAD45 RNA foci per cell (Fig. S5h) that they are detecting the mRNA rather than nascent transcripts at the site of transcription. I may have been mistaken in asking for RNA FISH, which I thought would have been clear enough to suggest that intron (and not exon) probes should be used. The use of mRNA probes (complementary to exon sequences) cannot differentiate between an mRNA in the nucleus or on its way to cytoplasm, and a nascent transcript at the site of the GAD45 gene locus. The authors state, “RNAscope, which allows to localize transcription sites in nuclei^{40,41...}”, but unfortunately did not apply the technique to specifically localize transcription sites.

The data in Fig. 3e shows a small difference in localization of GAD45 mRNAs compared to the control mRNAs. It is not clear, or at least I could not find, information on the number of cells counted which produced those results, making it extremely difficult to get an idea of the

overlap between the GAD45 mRNA and gammaH2AX foci. This needs to be explicitly stated since there are multiple GAD45 mRNA foci and multiple gammaH2AX foci per cell. One could imagine cells with one or two GAD45 foci overlapping with gammaH2AX foci, or perhaps cells with 5 or 7 GAD45 foci overlapping with gammaH2AX foci. Knowing which, if any, of these GAD45 foci is the transcribing gene is not possible as presented, nor with the probes used. If the authors wish to conclude that repair genes are transcribed in the D compartments, they need to show nascent transcripts (intron containing).

We thank the referee for their comments. Indeed, in the previous revised version of our manuscript, we used probes targeting mRNA for *GADD45A*, *CCL2* and *PPIB*, because the probe targeting pre mRNA for *GADD45A* did not work properly. Using these probes, we indeed detected, in addition to the transcription sites, additional foci corresponding to mRNA (on their way to be exported, under processing etc..).

In our new revised manuscript, in order to further strengthen our findings using RNAscope we now show data using intronic probes (thus staining transcription sites) against *PLK3* (Upregulated, D-gene) and two other genes neither upregulated nor targeted to the D-compartment. (Fig. 3e). The use of such probes indeed allowed to detect only 1-2 spots in the nuclei, thus corresponding to sites of transcription. We report that *PLK3* significantly colocalize more with γ H2AX foci than *CDC42* and *LPHN2*.

Reviewer Reports on the second Revision:

Referees' comments:

Referee #2:

The authors have included additional data to strengthen their observations and the new data provided are indeed technically sound. However, two important control experiments are required to strengthen the authors' proposed model.

1) The authors provide evidence that SETX depletion increases DSB clustering and enhances transcription of DDR upregulated genes and selectively triggers enhanced expression of DDR upregulated genes identified as targeted to the D compartment compared to those not targeted to the D compartment. Isn't it possible that enhanced transcription of those genes in SETX-depleted cells contributes to their clustering rather than clustering contributes to the activation of transcription? Can the authors clarify this? They could perform RNA-seq (or better nascent RNA-seq) in SETX-depleted cells before and after DSB induction. This experiment will rule out the possibility that SETX depletion leads to transcriptional changes of these genes that facilitates their clustering upon DSB formation.

2) Performing Hi-C upon overexpression of RNase H1 cannot be technically unfeasible as the authors claim, as they have performed already similar large-scale experiments (e.g. siRNA for SETX and Hi-C). Therefore, I believe it is an important experiment to show changes in the D compartment by Hi-C, and accompanied changes in R-loops and transcription levels genome-wide, upon RNase H1 overexpression.

Referee #3:

I do not want to hold up this manuscript. I am convinced that it contains important new information that will be of interest to the DNA repair community and to Nature readers in general. As I am not an expert in the DNA repair field I will have to accept the authors ascertain that all 53BP1 foci in the nucleus are in fact DSB regions and that imaging their dynamics is the same as measuring DSB locations or dynamics.

I thank the authors for the inclusion of RNA FISH analyses of DNA damage-upregulated genes in association with D compartments. Although the differences are small the results are statistically significant.

**Author Rebuttals to Second Revision:
Point by point response to reviewers**

1) The authors provide evidence that SETX depletion increases DSB clustering and enhances transcription of DDR upregulated genes and selectively triggers enhanced expression of DDR upregulated genes identified as targeted to the D compartment compared to those not targeted to the D compartment. Isn't it possible that enhanced transcription of those genes in SETX-depleted cells contributes to their clustering rather than clustering contributes to the activation of transcription? Can the authors clarify this? They could perform RNA-seq (or better nascent RNA-seq) in SETX-depleted cells before and after DSB induction. This experiment will rule out the possibility that SETX depletion leads to transcriptional changes of these genes that facilitates their clustering upon DSB formation.

To answer this question, we performed RNA-seq in CTRL and SETX-depleted cells before DSB induction. We retrieved the genes that are upregulated upon SETX depletion in absence of DSB induction. These genes do not display more D-compartment signal in SETX-depleted cells (using our HiC data in SETX siRNA) than the genes that are not upregulated upon SETX depletion. This shows that an enhanced expression is not a determinant of gene targeting to the D-compartment. These data have been added as Extended Data Fig. 7j.

Additionally, upon SETX depletion before any DNA damage induction, the DNA damage upregulated genes that are targeted in the D-compartment are not more expressed than their counterpart not targeted to the D-compartment (see below Fig. 1 for reference), in contrast to what happens after DSB induction (Fig. 4g of the manuscript).

As Fig. 4g, but before DSB induction (-DSB)

2) Performing Hi-C upon overexpression of RNase H1 cannot be technically unfeasible as the authors claim, as they have performed already similar large-scale experiments (e.g. siRNA for SETX and Hi-C). Therefore, I believe it is an important experiment to show changes in the D compartment by Hi-C, and accompanied changes in R-loops and transcription levels genome-wide, upon RNase H1 overexpression.

Hi-C experiment upon RNaseH1 overexpression is not similar to an Hi-C experiment upon SETX depletion. Indeed, overexpression of RNaseH1 displays toxicity and has a lot of drawbacks (see for instance the review of Chedin and Vanhoosthuyse (PMID: 33411340)). This experiment will be challenging, with no guarantee it will work. Since we already performed similar experiment upon SETX depletion to manipulate R-loops level, we believe these experiments are not truly essential to strengthen our conclusions.